# A Topological Perspective on Demystifying GNN-Based Link Prediction Performance

**Yu Wang**[1], **Tong Zhao**[2], **Yuying Zhao**[1], **Yunchao Liu**[1], **Xueqi Cheng**[1], **Neil Shah**[2], **Tyler Derr**[1]

[1]Vanderbilt University    [2]Snap Inc.

{yu.wang.1,yuying.zhao,yunchao.liu,xueqi.cheng,tyler.derr}@vanderbilt.edu
{tzhao,nshah}@snap.com

## Abstract

Graph Neural Networks (GNNs) have shown great promise in learning node embeddings for link prediction (LP). While numerous studies improve the overall GNNs' LP performance, none have explored their varying performance across different nodes and the underlying reasons. To this end, we demystify which nodes perform better from the perspective of their local topology. Despite the widespread belief that low-degree nodes exhibit worse LP performance, we surprisingly observe an inconsistent performance trend. This prompts us to propose a node-level metric, Topological Concentration (TC), based on the intersection of the local subgraph of each node with the ones of its neighbors. We empirically demonstrate that TC correlates with LP performance more than other node-level topological metrics, better identifying low-performing nodes than using degree. With TC, we discover a novel topological distribution shift issue in which nodes' newly joined neighbors tend to become less interactive with their existing neighbors, compromising the generalizability of node embeddings for LP at testing time. To make the computation of TC scalable, We further propose Approximated Topological Concentration (ATC) and justify its efficacy in approximating TC with reduced computation complexity. Given the positive correlation between node TC and its LP performance, we explore the potential of boosting LP performance via enhancing TC by re-weighting edges in the message-passing and discuss its effectiveness with limitations. Our code is publicly available at https://github.com/YuWVandy/Topo_LP_GNN.

## 1 Introduction

Recent years have witnessed unprecedented success in applying link prediction (LP) in real-world applications (Ying et al., 2018; Rozemberczki et al., 2022). Compared with heuristic-based (Brin & Page, 1998; Liben-Nowell & Kleinberg, 2003) and shallow embedding-based LP approaches (Perozzi et al., 2014; Grover & Leskovec, 2016), GNN-based ones (Zhang & Chen, 2018; Chamberlain et al., 2022) have achieved state-of-the-art (SOTA) performance; these methods first learn node/subgraph embeddings by applying linear transformations with message-passing and a decoder/pooling layer to predict link scores/subgraph class. While existing works are dedicated to boosting overall LP performance (Zhao et al., 2022; Chen et al., 2021b) by more expressive message-passing or data augmentation, it is heavily under-explored whether different nodes within a graph would obtain embeddings of different quality and have varying LP performance.

Previous works have explored GNNs' varying performance on nodes within a graph, considering factors like local topology (e.g., degree and homophily/heterophily) (Tang et al., 2020; Mao et al., 2023), feature quality (Taguchi et al., 2021), and class quantity (Zhao et al., 2021a). While these studies have provided significant insights, their focus has primarily remained on node/graph-level tasks, leaving the realm of LP unexplored. A more profound examination of the node-varying LP performance can enhance our comprehension of network dynamics (Liben-Nowell & Kleinberg, 2003), facilitate timely detection of nodes with ill-topology (Lika et al., 2014), and inspire customized data augmentation for different nodes (Zhao et al., 2021b). Recognizing the criticality of studying the node-varying LP performance and the apparent gap in the existing literature, we ask:

***Can we propose a metric that measures GNNs' varying LP performance across different nodes?***

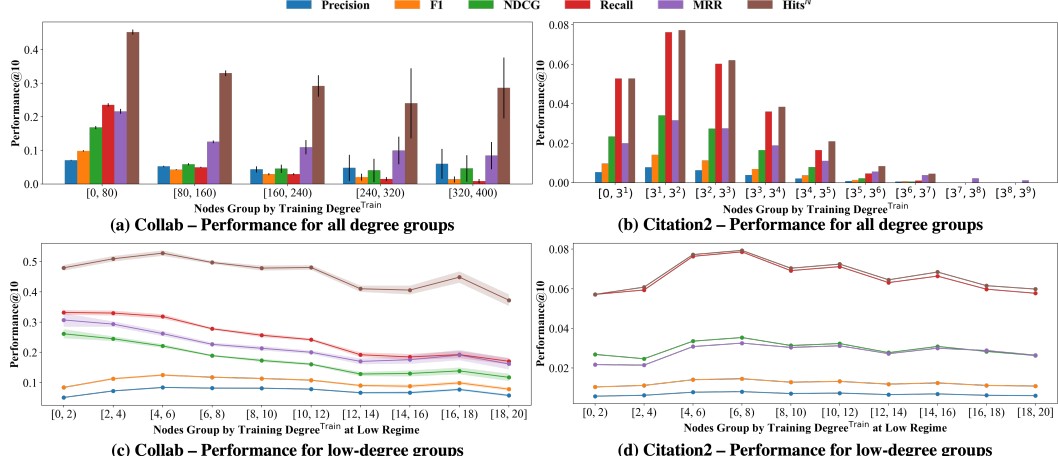

Figure 1: Average LP performance of nodes across different degree groups based on Degree$^{\text{Train}}$(i.e., node degree by training edges) on Collab/Citation2. In **(a)-(b)**, Performance@10 does not increase as the node degree increases. In **(c)-(d)**, few/lower-degree nodes do not perform worse than their higher-degree counterparts. Detailed experimental setting is included in Appendix F

To answer an analogous question in node classification, prior works observed that GNNs perform better on high-degree nodes than low-degree nodes (Tang et al., 2020; Liu et al., 2021). Similarly, the persistent sparse topology issue in the general LP domain and recommender systems (Hao et al., 2021; Li et al., 2021) indicates that nodes with zero-to-low degrees lag behind their high-degree counterparts. However, as surprisingly shown in Figure 1(a)-(b), GNN-based LP on these two large-scale social networks does not exhibit a consistent performance trend as the node degree increases. For example, the performance@10 on Collab under all evaluation metrics decreases as the node degree increases, while on Citation2, performance@10 first increases and then decreases. This counter-intuitive observation indicates the weak correlation between the node degree and LP performance, which motivates us to design a more correlated metric to answer the above question.

Following (Zhang & Chen, 2018) that the link formation between each pair of nodes depends on the interaction between their local subgraphs, we probe the relation between the local subgraphs around each node and its GNN-based LP performance. Specifically, we propose Topological Concentration (TC) and its scalable version, Approximated Topological Concentration (ATC), to measure the topological interaction between the local subgraph of each node and the local subgraphs of the neighbors of that node. Our empirical observations show that TC offers a superior characterization of node LP performance in GNNs, leading to 82.10% more correlation with LP performance and roughly 200% increase in the performance gap between the identified under-performed nodes and their counterparts than degree. Moreover, with TC, we discover a novel topological distribution shift (TDS) in which newly joined neighbors of a node tend to become less interactive with that node's existing neighbors. This TDS would compromise the generalizability of the learned node embeddings in LP at the testing time. Given the closer correlation between TC and LP performance, we reweigh the edges in message-passing to enhance TC and discuss its efficacy/limitations in boosting LP performance. Our contributions are summarized as follows:

- We propose Topological Concentration (TC) and demonstrate it leads to 82.10% more correlation with LP performance and roughly 200% increase in the performance gap between the identified under-performed nodes and their counterparts than node degree, shedding new insights on degree-related issues in LP. We further propose Approximated Topological Concentration (ATC) and demonstrate it maintains high correlations to the LP performance similar to TC while significantly reducing the computation complexity.

- We uncover a novel Topological Distribution Shift (TDS) issue according to TC and demonstrate its negative impact at the node/graph level for link prediction at the testing time. Moreover, we discover that different nodes within the same graph can have varying amounts of TDS.

- We design a TC-inspired message-passing where a node aggregates more from neighbors that have more connections to the local subgraph of that node, which can enhance the node's weighted TC. We observe this empirically boosts LP performance and lastly discuss its noncausal limitations.

## 2 RELATED WORK

**Varying Performance of GNNs on Node/Graph Classification.** GNNs' efficacy in classification differs across nodes/graphs with varying label quantity (e.g., imbalanced node/graph classification (Zhao et al., 2021a; Wang et al., 2022b)) and varying topology quality (e.g., long-tailed (Tang et al., 2020; Liu et al., 2021)/heterophily node classification (Zhu et al., 2020; Mao et al., 2023)). To enhance GNNs' performance for the disadvantaged nodes/graphs in these two varying conditions, previous works either apply data augmentations to derive additional supervision (Wang et al., 2021; Park et al., 2021) or design expressive graph convolutions to mitigate structural bias (Zhu et al., 2021; Han et al., 2023). However, none of them tackle the varying performance of nodes in LP. We fill this gap by studying the relationship between node LP performance and its local topology.

**GNN-based LP and Node-Centric Evaluation.** GNN-based LP works by first learning node embeddings/subgraph embeddings through linear transformation and message-passing, and then applying the scoring function to predict link probability/subgraph class (Zhang & Chen, 2018; Yin et al., 2022; Wang et al., 2022c; Tan et al., 2023). It has achieved new SOTA performance owing to using the neural network to extract task-related information and the message-passing to encode the topological properties (e.g., common neighbors) (Yun et al., 2021; Chamberlain et al., 2022). Existing GNN-based LP baselines evaluate performance by computing the average rank of each link against the randomly sampled negative links (Hu et al., 2020). However, because these sampled negative links only count a tiny portion of the quadratic node pairs, this evaluation contains positional bias (Li et al., 2023). In view of this issue, we leverage the node-centric evaluation metrics (Precision/F1/NDCG/Recall/Hits$^N$@K) that are frequently used in recommender systems (Gori et al., 2007; He et al., 2020) and rank each node against all other nodes in predicting the incoming neighbors. Detailed definitions of these evaluation metrics are provided in Appendix C.

**Varying Performance of GNNs on LP.** As LP nowadays has been heavily used to enhance user experience in social/e-commerce recommendations (Fan et al., 2019; Zhao et al., 2023), studying its varying performance across different users has real-world applications Zhao et al. (2024) such as identifying users with ill-topology. Although no efforts have been investigated into the node-varying performance in GNN-based LP, prior work (Li et al., 2021; Rahmani et al., 2022; Guo et al., 2024) have investigated the relation of node-varying LP performance with its degree, and both claimed that users/nodes with higher activity levels/degrees tend to possess better recommendation performance than their less active counterparts, which also aligns with observations in GNN-based node classification (Tang et al., 2020; Liu et al., 2021). However, Figure 1(c)-(d) has already raised concern over the validity of this claim in LP. We follow (Wang & Derr, 2022) and theoretically discover that some node-centric evaluation metrics have degree-related bias in Appendix D.2, implying that the GNNs' varying LP performance could be partially attributed to the choice of evaluation metrics. To mitigate this bias, we employ a full spectrum of evaluation metrics and find that degree is not so correlated with the node LP performance. This motivates us to devise a better topological metric than the degree. Note that although some prior works also define cold-start nodes to be the ones with few degrees, we systematically review their connections and differences in Appendix B.

## 3 TOPOLOGICAL CONCENTRATION

### 3.1 NOTATIONS

Let $G = (\mathcal{V}, \mathcal{E}, \mathbf{X})$ be an attributed graph, where $\mathcal{V} = \{v_i\}_{i=1}^n$ is the set of $n$ nodes (i.e., $n = |\mathcal{V}|$) and $\mathcal{E} \subseteq \mathcal{V} \times \mathcal{V}$ is the set of $m$ observed training edges (i.e., $m = |\mathcal{E}|$) with $e_{ij}$ denoting the edge between the node $v_i$ and $v_j$, and $\mathbf{X} \in \mathbb{R}^{n \times d}$ represents the node feature matrix. The observed adjacency matrix of the graph is denoted as $\mathbf{A} \in \{0, 1\}^{n \times n}$ with $\mathbf{A}_{ij} = 1$ if an observed edge exists between node $v_i$ and $v_j$ and $\mathbf{A}_{ij} = 0$ otherwise. The diagonal matrix of node degree is notated as $\mathbf{D} \in \mathbb{Z}^{n \times n}$ with the degree of node $v_i$ being $d_i = \mathbf{D}_{ii} = \sum_{j=1}^n \mathbf{A}_{ij}$. For the LP task, edges are usually divided into three groups notated as $\mathcal{T} = \{\text{Train}, \text{Val}, \text{Test}\}$, i.e., training, validation, and testing sets, respectively. We denote $\mathcal{N}_i^t, t \in \mathcal{T}$ as node $v_i$'s 1-hop neighbors according to edge group $t$. Furthermore, we denote the set of nodes that have at least one path of length $k$ to node $i$ based on observed training edges as $\mathcal{H}_i^k$ and naturally $\mathcal{H}_i^1 = \mathcal{N}_i^{\text{Train}}$. Note that $\mathcal{H}_i^{k_1} \cap \mathcal{H}_i^{k_2}$ is not necessarily empty since neighbors that are $k_1$-hops away from $v_i$ could also have paths of length $k_2$ reaching $v_i$. We collect $v_i$'s neighbors at all different hops until $K$ to form the $K$-hop computation tree centered on $v_i$ as $\mathcal{S}_i^K = \{\mathcal{H}_i^k\}_{k=1}^K$. We summarize all notations in Table 2 in Appendix A.

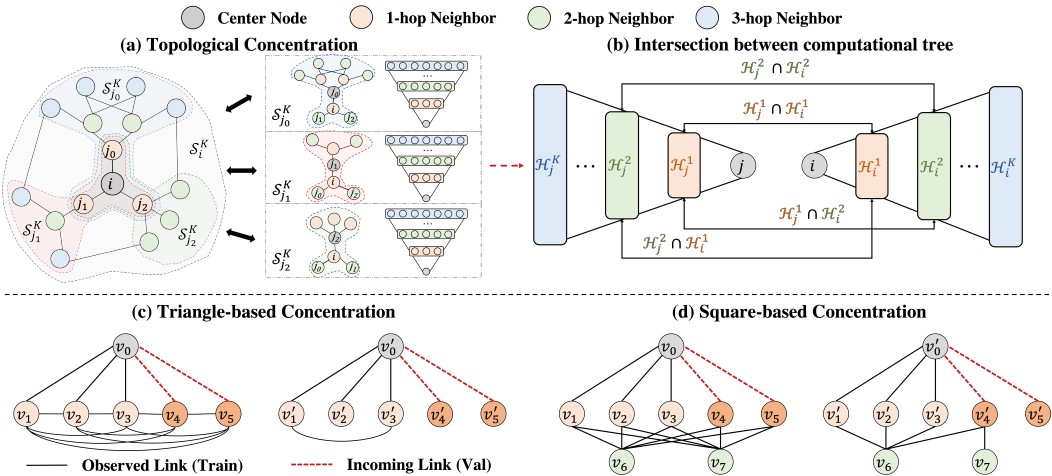

Figure 2: **(a)-(b)**: $v_i$'s Topological Concentration: we calculate the average intersection between $v_i$'s computation tree and each of $v_i$'s neighbor's computation tree. The intersection between two computation trees is the ratio of the observed intersections to all possible intersections. **(c)-(d)**: two specifications of TC, corresponding to social and e-commerce networks. A higher triangle/square-based concentration indicates more triangles/squares are formed among $v_0$'s local subgraph.

## 3.2 TOPOLOGICAL CONCENTRATION: INTUITION AND FORMALIZATION

As the link formation between a node pair heavily depends on the intersection between their local subgraphs (Zhang & Chen, 2018; Chamberlain et al., 2022), we similarly hypothesize that the predictability of a node's neighbors relates to the intersection between this node's subgraph and the subgraphs of that node's neighbors, e.g., the prediction of the links $\{(i, j_k)\}_{k=0}^{2}$ in Figure 2(a) depends on the intersection between $\mathcal{S}_i^K$ and $\{\mathcal{S}_{j_k}^K\}_{k=0}^{2}$. A higher intersection leads to higher LP performance. For example, in Figure 2(c)-(d), $v_0$ neighbors closely interact with themselves while $v_0'$ neighbors do not, posing different topological conditions for the LP on $v_0$ and $v_0'$. From graph heuristics perspective, $v_0$ shares common neighbors $v_1, v_2, v_3$ with its incoming validation neighbors $v_4, v_5$ while $v_0'$ shares no neighbors with $v_4', v_5'$. From the message-passing perspective, the propagated embeddings of $v_0$ and $v_4, v_5$ share common components since they all aggregate $\{v_k\}_{k=1}^{3}$ embeddings while $v_0'$ and $v_4', v_5'$ do not share any common embeddings among $\{v_k'\}_{k=1}^{3}$. When the subgraph (i.e., computation tree) surrounding a node increasingly overlaps with the subgraphs of its neighbors, more paths originating from that node are likely to loop nearby and eventually return to it, resulting in a more dense/concentrated local topology for that node. Inspired by this observation, we introduce *Topological Concentration* to measure the average level of intersection among these local subgraphs as follows:

**Definition 1.** *Topological Concentration (TC):* *The Topological Concentration $C_i^{K,t}$ for node $v_i \in \mathcal{V}$ is defined as the average intersection between $v_i$'s $K$-hop computation tree ($\mathcal{S}_i^K$) and the computation trees of each of $v_i$'s type $t$ neighbors:*

$$C_i^{K,t} = \mathbb{E}_{v_j \sim \mathcal{N}_i^t} I(\mathcal{S}_i^K, \mathcal{S}_j^K) = \mathbb{E}_{v_j \sim \mathcal{N}_i^t} \frac{\sum_{k_1=1}^{K} \sum_{k_2=1}^{K} \beta^{k_1+k_2-2} |\mathcal{H}_i^{k_1} \cap \mathcal{H}_j^{k_2}|}{\sum_{k_1=1}^{K} \sum_{k_2=1}^{K} \beta^{k_1+k_2-2} g(|\mathcal{H}_i^{k_1}|, |\mathcal{H}_j^{k_2}|)} \tag{1}$$

$\forall v_i \in \mathcal{V}, \forall t \in \mathcal{T}$, where $I(\mathcal{S}_i^K, \mathcal{S}_j^K)$ quantifies the intersection between the $K$-hop computation trees around $v_i$ and $v_j$, and is decomposed into the ratio of the observed intersections $|\mathcal{H}_i^{k_1} \cap \mathcal{H}_j^{k_2}|$ to the total possible intersections $g(\mathcal{H}_i^{k_1}, \mathcal{H}_j^{k_2})$ between neighbors that are $k_1$ and $k_2$ hops away as shown in Figure 2(b). $\beta^{k_1+k_2-2}$ accounts for the exponential discounting effect as the hop increases. The normalization term $g$ is a function of the size of the computation trees of node $v_i, v_j$ (Fu et al., 2022). Although computation trees only consist of edges from the training set, $v_i$'s neighbors $\mathcal{N}_i^t$ in Eq. (1) could come from training/validation/testing sets, and we term the corresponding TC as $\text{TC}^{\text{Train}}, \text{TC}^{\text{Val}}, \text{TC}^{\text{Test}}$ and their values as $C_i^{K,\text{Train}}, C_i^{K,\text{Val}}, C_i^{K,\text{Test}}$. We verify the correlation between TC and the node LP performance in Section 3.3.

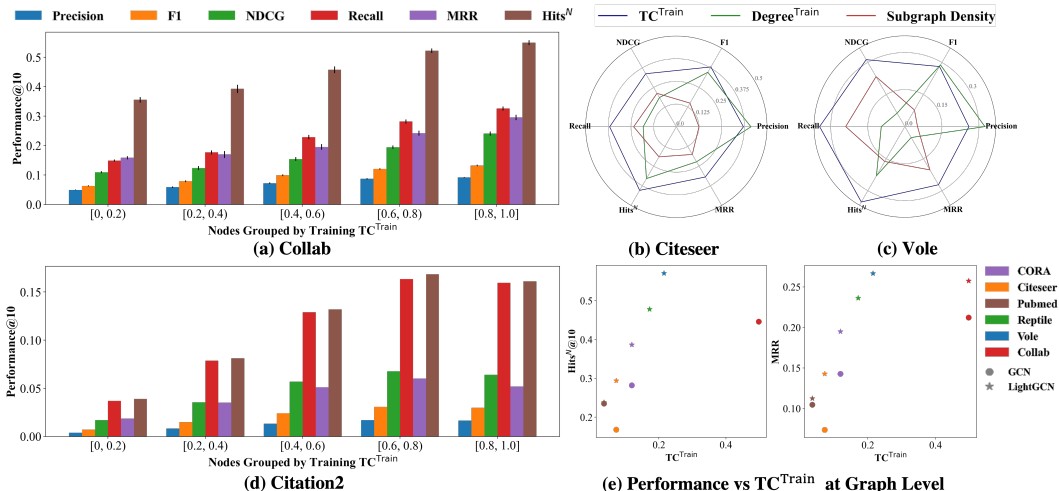

Figure 3: **(a)/(d)**: The average LP Performance of nodes on Collab/Citation2 monotonically increases as the $\text{TC}^{\text{Train}}$ increases. **(b)/(c)**: $\text{TC}^{\text{Train}}$ mostly achieves the highest Pearson Correlation with LP performance on Citeseer/Vole than $\text{Degree}^{\text{Train}}$ and Subgraph Density metrics. **(e)**: LP performance is positively correlated to $\text{TC}^{\text{Train}}$ across different network datasets. The deviation of Collab under both GCN and LightGCN baselines from the primary linear trend might be attributed to the duplicated edges in the network that create the illusion of a higher $\text{TC}^{\text{Train}}$.

## 3.3 TOPOLOGICAL CONCENTRATION: OBSERVATION AND ANALYSIS

In this section, we draw three empirical observations to delve into the role of TC in GNN-based LP. For all experiments, we evaluate datasets with only the topology information using LightGCN and those also with node features using GCN/SAGE (Kipf & Welling, 2016; Hamilton et al., 2017; He et al., 2020)[1]. Detailed experimental settings are described in Appendix F. Please note that while the findings illustrated in this section are limited to the presented datasets due to page limitation, more comprehensive results are included in Appendix G.

**Obs. 1. TC correlates to LP performance more than other node topological properties.** In Figure 3 (a)/(d), we group nodes in Collab/Citation2 based on their $\text{TC}^{\text{Train}}$ and visualize the average performance of each group. Unlike Figure 1(a)/(b), where there is no apparent relationship between the performance and the node degree, the performance almost monotonically increases as the node $\text{TC}^{\text{Train}}$ increases regardless of the evaluation metrics. This demonstrates the capability of $\text{TC}^{\text{Train}}$ in characterizing the quality of nodes' local topology for their LP performance. Moreover, we quantitatively compare the Pearson Correlation of the node LP performance with $\text{TC}^{\text{Train}}$ and other commonly used node local topological properties, $\text{Degree}^{\text{Train}}$ (i.e., the number of training edges incident to a node) and SubGraph Density (i.e., the density of the 1-hop training subgraph centering around a node). As shown in Figure 3(b)/(c), $\text{TC}^{\text{Train}}$ almost achieves the highest Pearson Correlation with the node LP performance across every evaluation metric than the other two topological properties except for the precision metric. This is due to the degree-related evaluation bias implicitly encoded in the precision metric, i.e., even for the untrained link predictor, the precision of a node still increases linearly as its degree increases, as proved in Theorem 3. Note that the node's 1-hop Subgraph Density equals its local clustering coefficient (LCC), and one previous work (Pan et al., 2022) has observed its correlation with node LP performance. To justify the advantages of $\text{TC}^{\text{Train}}$ over LCC, we provide a concrete example in Appendix E. Additionally, Figure 3(e) shows that $\text{TC}^{\text{Train}}$ also positively correlated with LP performance across various networks, depicting a preliminary benchmark for GNNs' LP performance at the graph level (Palowitch et al., 2022). The LightGCN architecture exhibits a steeper slope than GCN, as it relies exclusively on network topology without leveraging node features and thus is more sensitive to changes in the purely topological metric, TC. The deviation of Collab under both GCN and LightGCN baselines from the primary linear trend might be attributed to the duplicated edges in the network that create the illusion of a higher $\text{TC}^{\text{Train}}$ (Hu et al., 2020).

---

[1]Due to GPU memory limitation, we choose SAGE for Citation2.

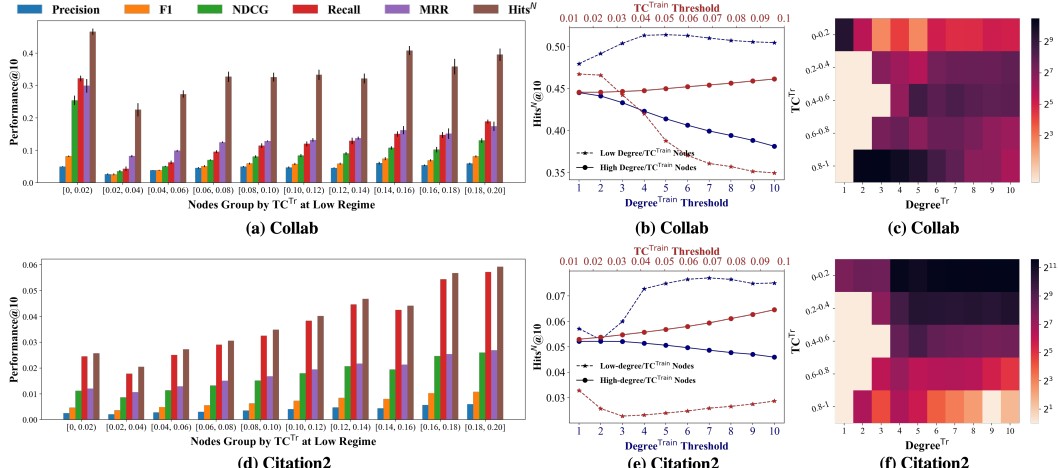

Figure 4: **(a)/(d)**: The average LP performance of nodes with extremely low $TC^{Train}$ on Collab/Citation2 almost monotonically increases as $TC^{Train}$ increases. **(b)/(e)**: Nodes with lower $Degree^{Train}$ surprisingly perform better than their higher degree counterparts (**Blue** curves). In contrast, non-concentrated nodes identified by owning lower $TC^{Train}$ in most cases perform worse than their concentrated counterparts (**Red** curves). **(c)/(f)**: As node $Degree^{Train}$ increases, the ratio of nodes owning higher $TC^{Train}$ increases first and then decreases, corresponding to the observed first-increasing-and-then-decreasing performance trend in Figure 1(c)/(d).

**Obs. 2. TC better identifies low-performing nodes than degree, and lower-degree nodes may not necessarily have lower LP performance.**

As previously shown in Figure 1(c)/(d), when the node degree is at the very low regime, we do not observe a strict positive relationship between node $Degree^{Train}$ and its LP performance. For example, the node Recall/MRR/NDCG@10 in Collab decreases as $Degree^{Train}$ increases and $Hits^N$/F1/Precision@10 first increases and then decreases. These contradicting observations facilitate our hypothesis that the degree might not fully capture the local topology in characterizing the underperforming nodes. Conversely, in Figure 4(a)/(d) on Collab/Citation2, nodes with lower $TC^{Train}$ almost always have worse LP performance under all evaluation metrics except when $TC^{Train}$ is between $[0, 0.02]$. For this extreme case, we ascribe it to the distribution shift as nodes with extremely low $TC^{Train}$ generally have a decent $TC^{Test}$ (shown in Figure 20) and sustain a reasonable LP performance. We thoroughly investigate this distribution shift issue in **Obs. 3**. Furthermore, we select nodes with $Degree^{Train}$ from 1 to 10 and group them into 'Lower-degree/Higher-degree'. Similarly, we select nodes with $TC^{Train}$ from 0.01 to 0.1 and group them into 'Concentrated/Non-Concentrated'. We compare their average LP performance on Collab/Citation2 in Figure 4(b)/(e). Intriguingly, Lower-degree nodes always perform better than their Higher-degree counterparts across all $Degree^{Train}$ thresholds. This brings nuances into the conventional understanding that nodes with a weaker topology (lower degree) would yield inferior performance (Li et al., 2021; Tang et al., 2020; Liu et al., 2021). In contrast, with our defined $TC^{Train}$, non-concentrated nodes (lower $TC^{Train}$) generally underperform by a noticeable margin than their concentrated counterparts (higher $TC^{Train}$).

We further visualize the relation between $Degree^{Train}$ and $TC^{Train}$ in Figure 4(c)/(f). When node $Degree^{Train}$ increases from 1 to 4, the ratio of nodes owning higher $TC^{Train}$ also increases because these newly landed nodes start forming interactions and create their initial topological context. Since we have already observed the positive correlation of $TC^{Train}$ to nodes' LP performance previously, the LP performance under some evaluation metrics also increases as the $Degree^{Train}$ initially increases from 0 to 4 observed in Figure 1(c)/(d). When $Degree^{Train}$ increases further beyond 5, the ratio of nodes owning higher $TC^{Train}$ gradually decreases, leading to the decreasing performance observed in the later stage of Figure 1(c)/(d). This decreasing $TC^{Train}$ is because, for high $Degree^{Train}$ nodes, their neighbors are likely to lie in different communities and share fewer connections among themselves. For example, in social networks, high-activity users usually possess diverse relations in different online communities, and their interacted people are likely from significantly different domains and hence share less common social relations themselves Zhao et al. (2021c; 2024).

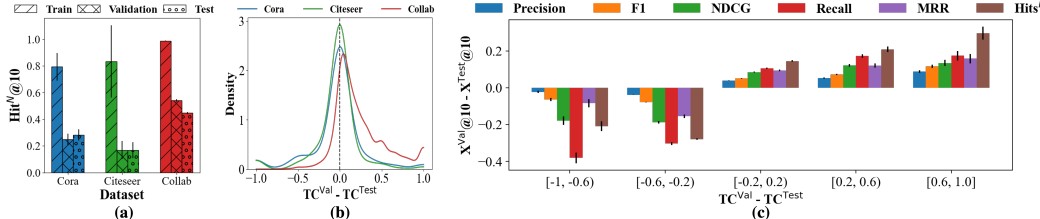

Figure 5: **(a)** Hits$^N$@10 of predicting training/validation/testing edges on Cora/Citeseer/Collab. The gap between validation and testing performance is much bigger on Collab than on Cora/Citeseer. **(b)** Compared with Cora/Citeseer where edges are randomly split, the distribution of the difference between TC$^{\text{Val}}$ and TC$^{\text{Test}}$ shifts slightly right on Collab where edges are split according to time, indicating the interaction between training and testing neighbors become less than the one between training and validation neighbors. **(c)** As the gap between TC$^{\text{Val}}$ and TC$^{\text{Test}}$ increases for different nodes, their corresponding performance gap also increases, demonstrating TDS varies among different nodes even within the same graph.

**Obs. 3. Topological Distribution Shift compromises the LP performance at testing time, and TC can measure its negative impact at both graph and node level.** In real-world LP scenarios, new nodes continuously join the network and form new links with existing nodes, making the whole network evolve dynamically (Ma et al., 2020; Rossi et al., 2020). *Here, we discover a new Topological Distribution Shift (TDS) issue, i.e., as time goes on, the newly joined neighbors of a node become less interactive with that node's old neighbors.* Since the edges serving message-passing and providing supervision only come from the training set, TDS would compromise the capability of the learned node embeddings for predicting links in the testing set. As verified in Figure 5(a), the performance gap between validation and testing sets on Collab where edges are split according to time is much more significant than the one on Cora/Citeseer where edges are split randomly. Note that the significantly higher performance on predicting training edges among all these three datasets is because they have already been used in the training phase (Wang et al., 2023), and this distribution shift is different from TDS. As TC essentially measures the interaction level among neighbors of a particular node, we further visualize the distribution of the difference between TC$^{\text{Val}}$ and TC$^{\text{Test}}$ in Figure 5(b). We observe a slight shift towards the right on Collab rather than on Cora/Citeseer, demonstrating nodes' testing neighbors become less interactive with their training neighbors than their validation neighbors. Figure 4(c) further demonstrates the influence of this shift at the node level by visualizing the relationship between TDS and the performance gap. We can see that as the strength of such shift increases (evidenced by the larger difference between TC$^{\text{Val}}$ and TC$^{\text{Test}}$), the performance gap also increases. This suggests that nodes within the same graph display varying levels of TDS. As one potential application, we can devise adaptive data valuation techniques or message-passing mechanisms to selectively use neighborhood information (i.e., emphasize less on stale edges in LP as Chamberlain et al. (2022) did). We demonstrate one use-case in Appendix K.

### 3.4 TOPOLOGICAL CONCENTRATION: COMPUTATIONAL COMPLEXITY AND OPTIMIZATION

Calculating TC following Eq. (1) involves counting the intersection between two neighboring sets that are different hops away from the centering nodes in two computation trees. Assuming the average degree of the network is $\widehat{d}$, the time complexity of computing $C_i^{K,t}$ for all nodes in the network is $\mathcal{O}(|\mathcal{E}| \sum_{k=1}^{K} \sum_{k=1}^{K} \min(\widehat{d}^{k_1}, \widehat{d}^{k_2})) = \mathcal{O}(K^2|\mathcal{E}||\mathcal{V}|) \approx \mathcal{O}(K^2|\mathcal{V}|^2)$ for sparse networks, which increases quadratically as the size of the network increases and is hence challenging for large-scale networks. To handle this issue, we propagate the randomly initialized Gaussian embeddings in the latent space to approximate TC in the topological space and propose *Approximated Topological Concentration* as follows:

**Definition 2.** *Approximated Topological Concentration (ATC): Approximated topological concentration $\widetilde{C}_i^{K,t}$ for $v_i \in \mathcal{V}$ is the average similarity between $v_i$ and its neighbors' embeddings initialized from Gaussian Random Projection (Chen et al., 2019) followed by row-normalized graph diffusion $\widetilde{\mathbf{A}}^k$ (Gasteiger et al., 2019), with $\phi$ as the similarity metric function:*

$$\widetilde{C}_i^{K,t} = \mathbb{E}_{v_j \sim \mathcal{N}_i^t} \phi(\mathbf{N}_i, \mathbf{N}_j), \quad \mathbf{N} = \sum_{k=1}^{K} \alpha_k \widetilde{\mathbf{A}}^k \mathbf{R}, \quad \mathbf{R} \sim \mathcal{N}(\mathbf{0}^d, \mathbf{\Sigma}^d) \tag{2}$$

**Theorem 1.** *Assuming* $g(|\mathcal{H}_i^{k_1}|, |\mathcal{H}_j^{k_2}|) = |\mathcal{H}_i^{k_1}||\mathcal{H}_j^{k_2}|$ *in Eq. (1) and let* $\phi$ *be the dot-product based similarity metric (He et al., 2020), then node* $v_i$*'s 1-layer Topological Concentration* $C_i^{1,t}$ *is linear correlated with the mean value of the 1-layer Approximated Topological Concentration* $\mu_{\widetilde{C}_i^{K,t}}$ *as:*

$$C_i^{1,t} \approx d^{-1} \mu_{\mathbb{E}_{v_j \sim \mathcal{N}_i^t}(\mathbf{E}_j^1)^\top \mathbf{E}_i^1} = d^{-1} \mu_{\widetilde{C}_i^{1,t}}, \tag{3}$$

where $\mathbf{E}^1 \in \mathbb{R}^{n \times d}$ denotes the node embeddings after 1-layer SAGE-style message-passing and $d$ is the embedding dimension. The full proof is in Appendix D. *This theorem bridges the gap between TC defined in the topological space and ATC defined in the latent space, which theoretically justifies the effectiveness of this approximation.* Computationally, obtaining node embeddings $\mathbf{N}$ in Eq. (2) is free from optimization, and the graph diffusion can be efficiently executed via power iteration, which reduces the complexity to $\mathcal{O}(Kd(|\mathcal{E}| + |\mathcal{V}|))$. Note that although we only demonstrate the approximation power for the case of 1-layer message-passing, we empirically verify the efficacy for higher-layer message-passing in the following.

We compare TC$^{\text{Train}}$ and ATC$^{\text{Train}}$ under various number of hops in terms of their computational time and their correlation with LP performance in Figure 6. As the number of hops increases, the running time for computing TC increases exponentially (especially for large-scale datasets like Collab, we are only affordable to compute its TC$^{\text{Train}}$ up to 3 hops) while ATC stays roughly the same. This aligns with the quadratic/linear time complexity $\mathcal{O}(K^2|\mathcal{V}|^2)/\mathcal{O}(Kd(|\mathcal{E}|+|\mathcal{V}|))$ we derived earlier

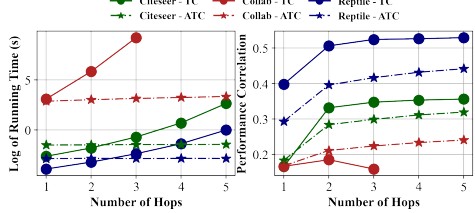

Figure 6: ATC$^{\text{Train}}$ maintains a similar level of correlation to TC$^{\text{Train}}$ while significantly reducing the computational time.

for TC/ATC. Moreover, ATC achieves a similar level of correlation to TC at all different hops. For both TC and ATC, their correlations to LP performance increase as the number of hops $K$ used in Eq. (1)-Eq. (2) increases. This is because larger $K$ enables us to capture intersections among larger subgraphs and hence accounts for more common neighbor signals (Chamberlain et al., 2022). We include comprehensive analysis including (A)TC$^{\text{Val}}$/(A)TC$^{\text{Test}}$ in Appendix J.

## 4 TOPOLOGICAL CONCENTRATION: BOOSTING GNNS' LP PERFORMANCE

From the aforementioned observations, TC consistently exhibits a stronger correlation with GNNs' LP performance than other commonly used node topological metrics. This insight motivates us to explore the potential of boosting GNNs' LP performance via enhancing TC$^{\text{Train}}$. Specifically, we propose to vary the edge weight used in message-passing by aggregating more information from neighbors that contribute more to TC$^{\text{Train}}$. We theoretically prove this way could enhance 1-layer TC$^{\text{Train}}$ in Theorem 4 and empirically verify it in Figure 7(a). Since neighbors owning more connections with the whole neighborhood have higher LP scores with the average neighborhood embeddings, we update the adjacency matrix used for message-passing as:

$$\widetilde{\mathbf{A}}_{ij}^\tau = \begin{cases} \widetilde{\mathbf{A}}_{ij}^{\tau-1} + \gamma \frac{\exp(g_{\mathbf{\Theta}_g}(\mathbf{N}_i^{\tau-1}, \mathbf{H}_j^{\tau-1}))}{\sum_{j=1}^n \exp(g_{\mathbf{\Theta}_g}(\mathbf{N}_i^{\tau-1}, \mathbf{H}_j^{\tau-1}))}, & \text{if} \mathbf{A}_{ij} = 1 \\ 0, & \text{if} \mathbf{A}_{ij} = 0 \end{cases}, \forall v_i, v_j \in \mathcal{V}, \tag{4}$$

where $\mathbf{H}^{\tau-1} = f_{\mathbf{\Theta}_f}(\widetilde{\mathbf{A}}^{\tau-1}, \mathbf{X})$ is the node embeddings obtained from GNN model $f_{\mathbf{\Theta}_f}$, $\mathbf{N}^{\tau-1} = \widetilde{\mathbf{A}}\mathbf{H}^{\tau-1}$ is the average neighborhood embeddings by performing one more SAGE-style propagation, $\gamma$ is the weight coefficient, $\widetilde{\mathbf{A}}^0 = \widetilde{\mathbf{A}}$, and $g_{\mathbf{\Theta}_g}$ is the link predictor. The detailed algorithm is presented in Appendix H. Note that this edge reweighting strategy directly operates on the original adjacency matrix and hence shares the same time/space complexity as the conventional message-passing $\mathcal{O}(Ld(|\mathcal{E} + \mathcal{V}|))$ with $L$ being the number of message-passing layers. For baselines obtaining node embeddings without using message-passing such as BUDDY Chamberlain et al. (2022), we reweigh edges in computing the binary cross entropy loss in the training stage. More details are attached Appendix I. Specifically, we equip four baselines, GCN/SAGE/NCN/BUDDY, with our designed edge reweighting strategy and present their LP performance in Table 11. Appendix F thoroughly describes the experimental settings. For GCN, SAGE and BUDDY, equipping with our proposed edge reweighting strategy enhances their LP performance in most cases. This demonstrates that by pushing up the TC$^{\text{Train}}$ of the whole graph, the LP performance can be boosted to a certain level, which is also verified by the increasing TC shown in Figure 7(a). We hypothesize that neighbors

Table 1: Results on LP benchmarks. $X_{rw}$ denotes weighted message-passing added to baseline X.

| Baseline | Cora Hits@100 | Citeseer Hits@100 | Pubmed Hits@100 | Collab Hits@50 | Collab* Hits@50 | Citation2 MRR | Reptile Hits@100 | Vole Hits@100 |
|---|---|---|---|---|---|---|---|---|
| GCN | 70.63±0.67 | 65.96±2.12 | 69.35±1.02 | 49.52±0.52 | 58.00±0.27 | 84.42±0.05 | 65.52±2.73 | 73.84±0.98 |
| GCN$_{rw}$ | 75.98±1.28 | 74.40±1.13 | 68.87±0.99 | 52.85±0.14 | 60.57±0.38 | 85.34±0.30 | 70.79±2.00 | 74.50±0.84 |
| SAGE | 74.27±2.08 | 61.57±3.28 | 66.25±1.08 | 50.01±0.50 | 57.06±0.06 | 80.44±0.10 | 72.59±3.19 | 80.55±1.59 |
| SAGE$_{rw}$ | 74.62±2.30 | 69.89±1.66 | 66.77±0.69 | 52.59±0.37 | 59.26±0.08 | 80.61±0.10 | 74.35±3.20 | 81.27±0.96 |
| NCN | 87.73±1.41 | 90.93±0.83 | 76.20±1.55 | 54.43±0.17 | 65.34±0.03 | 88.64±0.14 | 68.37±3.57 | 66.10±1.13 |
| NCN$_{rw}$ | 87.95±1.30 | 91.86±0.82 | 76.51±1.41 | 54.16±0.24 | 65.41±0.03 | OOM | 73.81±4.71 | 67.32±0.81 |
| BUDDY | 84.82±1.93 | 91.44±1.14 | 74.46±3.32 | 55.73±0.16 | 66.25±0.28 | 88.41±0.10 | 74.69±7.55 | 83.37±1.05 |
| BUDDY$_{rw}$ | 85.06±1.81 | 91.86±1.00 | 75.74±2.03 | 56.17±0.42 | 66.31±0.50 | 88.48±0.09 | 77.20±4.10 | 84.64±1.29 |

Note that for Citation2, due to memory limitation, we directly take the result from the original NCN paper (Wang et al., 2023). Different from Collab only using training edges in message-passing for evaluating testing performance, we also allow validation edges in Collab*.

connecting more to the overall neighborhood likely have greater interactions with incoming neighbors. Thus, aggregating more information from them inherently captures the common neighbor signals of these incoming neighbors. Meanwhile, as NCN already explicitly accounts for this common neighbor signal in its decoder, the performance gains from our strategy are relatively modest. Furthermore, the positive trend observed in Figure 7(b) verifies that the larger enhancement in node $TC^{Train}$ leads to a larger performance boost in its $Hits^N@10$. However, we note that LP performance is evaluated on the incoming testing neighbors rather than training neighbors, so $TC^{Train}$ is more of a correlated metric than causal. To illustrate, consider an extreme case where a node $v_i$ has training neighbors forming a complete graph with no connection to its incoming neighbors. In such a scenario, even if we achieve the maximum $TC^{Train} = 1$ and a significantly high training performance, it still hardly generalizes to predicting testing links. This could be reflected in the inconsistent trend in Figure 21/22 in Appendix G.7. In addition, Figure 7(c) verifies the assumption made in Theorem 4 and in the reweighting model design that nodes with larger $TC^{Train}$ have higher average embedding similarity to their neighbors.

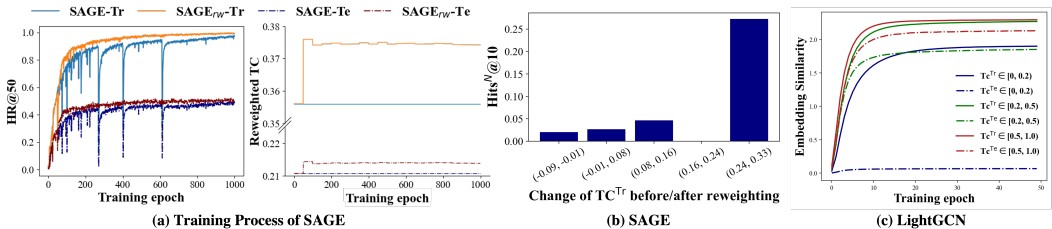

Figure 7: **(a)** Change of Hits@50 and Reweighted TC along the training process in SAGE. **(b)** Nodes with enhanced $TC^{Train}$ exhibit higher performance. **(c)** Embedding similarity during LightGCN training increases faster and higher for nodes with higher $TC^{Train}$.

## 5 CONCLUSION

Although many recent works have achieved unprecedented success in enhancing link prediction (LP) performance with GNNs, demystifying the varying levels of embedding quality and LP performance across different nodes within the graph is heavily under-explored yet fundamental. In this work, we take the lead in understanding the nodes' varying performance from the perspective of their local topology. Given the connection between link formation and the subgraph interaction, we propose Topological Concentration (TC) and demonstrate its superiority in characterizing node LP performance and identifying low-performing nodes by specifically verifying its higher performance correlation than other node-centric topological metrics. Moreover, we discover a novel topological distribution shift (TDS) issue by observing the changing LP performance over time and demonstrate the capability of using TC to measure this distribution shift. Our work offers the community strong insights into which local topology enables nodes to have better LP performance with GNNs. In the future, we plan to investigate the causal relationship between TC and LP. Additionally, we aim to utilize TC for data valuation to select consistently crucial edges in dynamic link prediction.

## ACKNOWLEDGEMENTS

This research is supported by the National Science Foundation (NSF) under grant number IIS2239881 and by Snap Inc.

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

# Appendix

## Table of Contents

# A NOTATIONS

This section summarizes notations used throughout this paper.

Table 2: Notations used throughout this paper.

| Notations | Definitions or Descriptions |
|---|---|
| $G = (\mathcal{V}, \mathcal{E}, \mathbf{X})$ | Graph with node set $\mathcal{V}$, edge set $\mathcal{E}$ and node feature $\mathbf{X}$ |
| $m, n$ | Number of nodes $m = |\mathcal{V}|$ and number of edges $n = |\mathcal{E}|$ |
| $v_i, e_{ij}$ | Node $v_i$ and the edge $e_{ij}$ between node $v_i$ and $v_j$ |
| $\mathbf{A}$ | Adjacency matrix $\mathbf{A}_{ij} = 1$ indicates an edge $e_{ij}$ between $v_i, v_j$ |
| $\widetilde{\mathbf{A}}$ | Row-based normalized graph adjacency matrix $\widetilde{\mathbf{A}} = \mathbf{D}^{-1}\mathbf{A}$ |
| $\widehat{\mathbf{A}}$ | GCN-based normalized graph adjacency matrix $\widehat{\mathbf{A}} = \mathbf{D}^{-0.5}\mathbf{A}\mathbf{D}^{-0.5}$ |
| $\widetilde{\mathbf{A}}^t$ | Updated adjacency matrix at iteration $t$ |
| $\mathbf{D}$ | Diagonal degree matrix $\mathbf{D}_{ii} = \sum_{j=1}^{n} \mathbf{A}_{ij}$ |
| $\widehat{d}$ | Average degree of the network |
| $\mathcal{T} = \{\text{Train, Val, Test}\}$ | Set of Training/Validation/Testing edge groups |
| $\text{Degree}^{\text{Train/Val/Test}}$ | Degree based on Training/Validation/Testing Edges |
| $\text{TC}^{\text{Train/Val/Test}}$ | Topological Concentrations that quantify intersections with Training/Validation/Testing neighbors |
| $\mathcal{N}_i^t$ | Node $v_i$'s 1-hop neighbors of type $t, t \in \mathcal{T}$ |
| $\mathcal{H}_i^k$ | Nodes having at least one path of length $k$ to $v_i$ based on training edges $\mathcal{E}^{\text{Train}}$ |
| $\mathcal{S}_i^K = \{\mathcal{H}_i^k\}_{k=1}^K$ | K-hop computational tree centered on the node $v_i$ |
| $C_i^{K,t} \backslash \widetilde{C}_i^{K,t}$ | (Approximated) Topological concentration for node $v_i$ considering the intersection among $K$-hop computational trees among its type $t$ neighbors. |
| $\mathbf{E}_i^k$ | Embedding of the node $v_i$ after $k^{\text{th}}$-layer message-passing |
| $\mathbf{R}_{ij}$ | Sample from gaussian random variable $\mathcal{N}(0, 1/d)$ |
| $g_{\mathbf{\Theta}_g}$ | Link predictor parameterized by $\mathbf{\Theta}_g$ |
| $\widetilde{\mathcal{E}}_i, \mathcal{E}_i$ | Predicted and ground-truth neighbors of node $v_i$ |
| $\mathcal{HG}$ | Hypergeometric distribution |
| LP | Link Prediction |
| (A)TC | (Approxminated) Topological Concentration |
| TDS | Topological Distribution Shift |
| $\beta$ | Exponential discounting effect as the hop increases |
| $\alpha_k$ | Weighted coefficient of layer $k$ in computing ATC |
| $\mu$ | Mean of the distribution |
| $L$ | Number of message-passing layers |
| $\gamma$ | Coefficients measuring the contribution of updating adjacency matrix |

## B  REVIEW OF THE COLD-START ISSUE IN LINK PREDICTION AND RECOMMENDER SYSTEMS

One line of the research (Leroy et al., 2010; Ge & Zhang, 2012; Yan et al., 2013; Han et al., 2015; Wang et al., 2016; Xu et al., 2017) defines the cold-start nodes as the ones with little to no topological information (isolated user) and augment these disadvantaged groups with auxiliary information, e.g., user profile/rich text information, community information, and group membership. Specifically, (Yan et al., 2013) derive the auxiliary information based on the interactions of these disadvantageous nodes/users from their cross-platform behaviors. (Leroy et al., 2010) constructs the probabilistic graph and then refines it by considering the transitivity of the contract relationship. (Ge & Zhang, 2012) incorporates feature selection and regularization to avoid overfitting. The other line of research (Wang et al., 2019; Dong et al., 2020; Li et al., 2021; Hao et al., 2021; Rahmani et al., 2022; Wei & He, 2022) studies the cold-start issue from the user perspective in recommender systems. They usually define cold-start nodes/users as the ones with no-to-sparse/low activities. (Li et al., 2021; Rahmani et al., 2022) devises a re-ranking strategy by optimizing the performance gap between low-activity and high-activity users. (Dong et al., 2020; Wei & He, 2022) design multiple meta-learning frameworks to learn user preferences based on his/her few past interactions. (Wang et al., 2019) uses knowledge graph embedding to assist with recommendation tasks for low-activity users while (Hao et al., 2021) trains GNNs to adapt to cold-start nodes by mimicking the cold-start scenario for warm users.

Following the above second line of research, we study the cold-start link prediction at the node level since our paper targets demystifying the varying link prediction performance across different nodes. Therefore, we follow some conventional literature (Wang et al., 2019; Dong et al., 2020; Li et al., 2021; Wei & He, 2022) and deem the nodes with generally few degrees as cold-start ones. Particularly, in Figure 4(b)/(e), we change the degree threshold from 1 to 10, divide nodes into two groups at each degree threshold, and further visualize the average performance for each group. We can see that nodes in the lower-degree groups generally have higher performance than nodes in the higher-degree groups. The above observation has two promising insights compared with conventional literature:

- **(1)** Many existing recommendation-based papers (Wang et al., 2019; Dong et al., 2020; Li et al., 2021; Newman, 2001) define cold-start users/nodes as the ones with few/little interactions/topological signals. However, our paper empirically demonstrates that nodes with lower degrees even exhibit higher LP performance.

- **(2)** Many existing node classification papers (Tang et al., 2020; Chen et al., 2021a; Wang et al., 2022a) find nodes with low degrees have lower performance. However, our work sheds new insights into the degree-related bias in link prediction where nodes with lower degrees can actually possess higher performance.

We justify the above $1^{st}$ insight by relating to real-world scenarios where users with high degrees usually tend to possess diverse interests (nodes with higher degrees may tend to belong to diverse communities) and therefore, using the equal capacity of embedding cannot equally characterize all of their interests (Zhao et al., 2021d).

We justify the above $2^{nd}$ insight by relating to the inherent difference between the mechanism of node classification and the mechanism of link prediction. For node classification, high-degree nodes are more likely to obtain the supervised signals from labeled nodes in the same class (Chen et al., 2021a). For link prediction, the ground-truth class for each node is actually its testing neighbors and hence when performing message-passing, beneficial supervision signals are not guaranteed to be captured more by high-degree nodes.

In our paper, we focus on the performance difference between low-degree nodes and high-degree nodes rather than the cold-start issue. However, if we also consider cold-start nodes as the ones with sparse interactions as some previous work did (Li et al., 2021; Rahmani et al., 2022), then our analysis and observation can also apply there.

# C LINK-CENTRIC AND NODE-CENTRIC EVALUATION METRICS

In addition to the conventional link-centric evaluation metrics used in this work, node-centric evaluation metrics are also used to mitigate the positional bias caused by the tiny portion of the sampled negative links. We introduce their mathematical definition respectively as follows:

## C.1 LINK-CENTRIC EVALUATION

Following (Hu et al., 2020), we rank the prediction score of each link among a set of randomly sampled negative node pairs and calculate the link-centric evaluation metric Hits@$K$ as the ratio of positive edges that are ranked at $K^{\text{th}}$-place or above. Note that this evaluation may cause bias as the sampled negative links only count a tiny portion of the quadratic node pairs (Li et al., 2023). Hereafter, we introduce the node-centric evaluation metrics and specifically denote the node-level Hit ratio as Hits$^N$@$K$ to differentiate it from the link-centric evaluation metric Hits@$K$.

## C.2 NODE-CENTRIC EVALUATION

For each node $v_i \in \mathcal{V}$, the model predicts the link formation score between $v_i$ and *every other node*, and selects the top-$K$ nodes to form the potential candidates $\widetilde{\mathcal{E}}_i$. Since the ground-truth candidates for node $v_i$ is $\mathcal{N}_i^{\text{Test}}$ (hereafter, we notate as $\widehat{\mathcal{E}}_i$), we can compute the Recall (R), Precision (P), F1, NDCG (N), MRR and Hits$^N$ of $v_i$ as follows:

$$\text{R@}K_i = \frac{|\widetilde{\mathcal{E}}_i \cap \widehat{\mathcal{E}}_i|}{|\widehat{\mathcal{E}}_i|}, \qquad \text{P@}K_i = \frac{|\widetilde{\mathcal{E}}_i \cap \widehat{\mathcal{E}}_i|}{K} \tag{5}$$

$$\text{F1@}K_i = \frac{2|\widetilde{\mathcal{E}}_i \cap \widehat{\mathcal{E}}_i|}{K + |\widehat{\mathcal{E}}_i|}, \qquad \text{N@}K_i = \frac{\sum_{k=1}^{K} \frac{\mathbb{1}[v_{\phi_i^k} \in (\widetilde{\mathcal{E}}_i \cap \widehat{\mathcal{E}}_i)]}{\log_2(k+1)}}{\sum_{k=1}^{K} \frac{1}{\log_2(k+1)}} \tag{6}$$

$$\text{MRR@}K_i = \frac{1}{\min_{v \in (\widetilde{\mathcal{E}}_i \cap \widehat{\mathcal{E}}_i)} \text{Rank}_v}, \qquad \text{Hits}^N\text{@}K_i = \mathbb{1}[|\widehat{\mathcal{E}}_i \cap \widetilde{\mathcal{E}}_i| > 0], \tag{7}$$

where $\phi_i^k$ denotes $v_i$'s $k^{\text{th}}$ preferred node according to the ranking of the link prediction score, $\text{Rank}_v$ is the ranking of the node $v$ and $\mathbb{1}$ is the indicator function equating 0 if the intersection between $\widehat{\mathcal{E}}^i \cap \widetilde{\mathcal{E}}_i$ is empty otherwise 1. The final performance of each dataset is averaged across each node:

$$\text{X@}K = \mathbb{E}_{v_i \in \mathcal{V}} \text{X@}K_i, \text{X} \in \{\text{R}, \text{P}, \text{F1}, \text{N}, \text{MRR}, \text{Hits}^N\} \tag{8}$$

Because for each node, the predicted neighbors will be compared against all the other nodes, there is no evaluation bias compared with the link-centric evaluation where only a set of randomly selected negative node pairs are used.

# D PROOF OF THEOREMS

## D.1 APPROXIMATION POWER OF ATC FOR TC

**Theorem 1.** *Assuming $g(|\mathcal{H}_i^{k_1}|, |\mathcal{H}_j^{k_2}|) = |\mathcal{H}_i^{k_1}||\mathcal{H}_j^{k_2}|$ in Eq. (1) and let $\phi$ be the dot-product based similarity metric (He et al., 2020), then node $v_i$'s 1-layer Topological Concentration $C_i^{1,t}$ is linear correlated with the mean value of the 1-layer Approximated Topological Concentration $\mu_{\widetilde{C}_i^{K,t}}$ as:*

$$C_i^{1,t} \approx d^{-1}\mu_{\mathbb{E}_{v_j \sim \mathcal{N}_i^t}(\mathbf{E}_j^1)^\top \mathbf{E}_i^1} = d^{-1}\mu_{\widetilde{C}_i^{1,t}}, \tag{9}$$

*where $\mathbf{E}^1 \in \mathbb{R}^{n \times d}$ denotes the node embeddings after 1-layer SAGE-style message-passing over the node embeddings $\mathbf{R} \sim \mathcal{N}(\mathbf{0}^d, \boldsymbol{\Sigma}^d)$.*

*Proof.* Assuming without loss of generalizability that the row-normalized adjacency matrix $\widetilde{\mathbf{A}} = \mathbf{D}^{-1}\mathbf{A}$ is used in aggregating neighborhood embeddings. We focus on a randomly selected node $\mathbf{E}_i \in \mathbb{R}^d, \forall v_i \in \mathcal{V}$ and its 1-layer ATC given by Eq. (2) is:

$$
\begin{aligned}
\widetilde{C}_i^{1,t} = \mathbb{E}_{v_j \sim \mathcal{N}_i^t}(\mathbf{E}_j^1)^\top \mathbf{E}_i^1 &= \mathbb{E}_{v_j \sim \mathcal{N}_i^t}(\widetilde{\mathbf{A}}\mathbf{R})_j^\top(\widetilde{\mathbf{A}}\mathbf{R})_i \\
&= \mathbb{E}_{v_j \sim \mathcal{N}_i^t}\frac{1}{|\mathcal{N}_j^{\text{Train}}||\mathcal{N}_i^{\text{Train}}|}\Big(\sum_{v_m \in \mathcal{N}_j^{\text{Train}}} \mathbf{R}_m\Big)^\top\Big(\sum_{v_n \in \mathcal{N}_i^{\text{Train}}} \mathbf{R}_n\Big) \\
&= \mathbb{E}_{v_j \sim \mathcal{N}_i^t}\frac{1}{|\mathcal{N}_j^{\text{Train}}||\mathcal{N}_i^{\text{Train}}|}\sum_{(v_m, v_n) \in \mathcal{N}_j^{\text{Train}} \times \mathcal{N}_i^{\text{Train}}}(\mathbf{R}_m)^\top\mathbf{R}_n \\
&= \mathbb{E}_{v_j \sim \mathcal{N}_i^t}\frac{1}{|\mathcal{H}_i^1||\mathcal{H}_j^1|}\Big(\underbrace{\sum_{\substack{(v_m,v_n) \in \mathcal{N}_j^{\text{Train}} \times \mathcal{N}_i^{\text{Train}}, \\ v_m \neq v_n}}(\mathbf{R}_m)^\top\mathbf{R}_n}_{\text{Non-common neighbor embedding pairs}} + \underbrace{\sum_{v_k \in \mathcal{N}_j^{\text{Train}} \cap \mathcal{N}_i^{\text{Train}}}(\mathbf{R}_k)^\top\mathbf{R}_k}_{\text{Common neighbor embedding pairs}}\Big),
\end{aligned}
\tag{10}
$$

Note that the first term is the dot product between any pair of two non-common neighbor embeddings, which is essentially the dot product between two independent samples from the same multivariate Gaussian distribution (*note that here we do not perform any training optimization, so the embeddings of different nodes are completely independent*). By central limit theorem (Kwak & Kim, 2017), the first term approaches the standard Gaussian distribution with 0 as the mean, *i.e.*, $\mu_{(\mathbf{R}_m)^\top\mathbf{R}_n} = 0$. In contrast, the second term is the dot product between any Gaussian-distributed sample and itself, which can be essentially characterized as the sum of squares of $d$ independent standard normal random variables and hence follows the chi-squared distribution with $d$ degrees of freedom, i.e., $(\mathbf{R}_k)^\top\mathbf{R}_k \sim \chi_d^2$ (Sanders, 2009). By Central Limit Theorem, $\lim_{d \to \infty} P(\frac{\chi_d^2 - d}{\sqrt{2d}} \leq z) = P_{\mathcal{N}(0,1)}(z)$ and hence $\lim_{d \to \infty} \chi_d^2 = \mathcal{N}(d, 2d), i.e., \mu_{(\mathbf{R}_k)^\top\mathbf{R}_k} = d$. Then we obtain the mean value of $\mathbb{E}_{v_j \sim \mathcal{N}_i^t}(\mathbf{E}_j^1)^\top\mathbf{E}_i^1$:

$$
\begin{aligned}
\mu_{\widetilde{C}_i^{1,t}} = \mu_{\mathbb{E}_{v_j \sim \mathcal{N}_i^t}(\mathbf{E}_j^1)^\top\mathbf{E}_i^1} &\approx \mathbb{E}_{v_j \sim \mathcal{N}_i^t}\frac{1}{|\mathcal{H}_i^1||\mathcal{H}_j^1|}\Big(\mu_{\sum_{\substack{(v_m,v_n) \in \mathcal{N}_j^{\text{Train}} \times \mathcal{N}_i^{\text{Train}}, \\ v_m \neq v_n}}(\mathbf{R}_m)^\top\mathbf{R}_n} + \mu_{\sum_{v_k \in \mathcal{N}_j^{\text{Train}} \cap \mathcal{N}_i^{\text{Train}}}(\mathbf{R}_k)^\top\mathbf{R}_k}\Big) \\
&\approx \mathbb{E}_{v_j \in \mathcal{N}_i^t}\frac{d|\mathcal{N}_i^{\text{Train}} \cap \mathcal{N}_j^{\text{Train}}|}{|\mathcal{H}_i^1||\mathcal{H}_j^1|} = \mathbb{E}_{v_j \in \mathcal{N}_i^t}\frac{d|\mathcal{H}_i^1 \cap \mathcal{H}_j^1|}{|\mathcal{H}_i^1||\mathcal{H}_j^1|} = dC_i^{1,t}.
\end{aligned}
\tag{11}
$$

The first approximation holds if assuming all nodes share the same degree. The second approximation holds since we set $d$ to be at least $64$ for all experiments in this paper. We next perform Monte-Carlo Simulation to verify that by setting $d = 64$, the obtained distribution is very similar to the Gaussian distribution. Assuming without loss of generality that the embedding dimension is 64 with the mean vector $\boldsymbol{\mu} = \mathbf{0}^{64} \in \mathbb{R}^{64}$ and the identity covariance matrix $\boldsymbol{\Sigma}^{64} = \mathbf{I} \in \mathbb{R}^{64 \times 64}$, we randomly sample 1000 embeddings from $\mathcal{N}(\boldsymbol{\mu}, \boldsymbol{\Sigma})$.

We visualize the distributions of the inner product between the pair of non-common neighbor embeddings, i.e., the first term in Eq. (10) $(\mathbf{R}_m)^\top \mathbf{R}_n, v_m \neq v_n$, and the pair of common neighbor embeddings, i.e., the second term in Eq. (10) $(\mathbf{R}_k)^\top \mathbf{R}_k, v_k \in \mathcal{N}_j^{\text{Train}} \cap \mathcal{N}_i^{\text{Train}}$ in Figure 8. We can see that the distribution of the dot product between the pair of non-common neighbor embeddings behaves like a Gaussian distribution centering around 0. In contrast, the distribution of the dot product between the pair of common neighbor embeddings behaves like a chi-square distribution of degree 64, which also centers around 64, and this in turn verifies the Gaussian approximation. Note that the correctness of the first approximation in Eq. (11) relies on the assumption that the average of the inverse of the node's neighbors should be the same across all nodes. Although it cannot be theoretically satisfied, we still empirically verify the positive correlation between TC and the link prediction performance shown in Figure 3.

*The above derivation bridges the gap between the Topological Concentration (TC) defined in the topological space and the Approximated Topological Concentration (ATC) defined in the latent space, which theoretically justifies the approximation efficacy of ATC.* $\square$

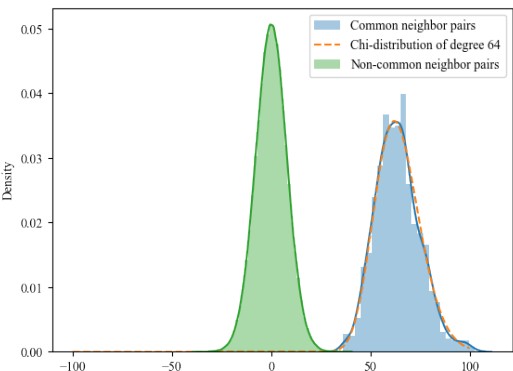

Figure 8: The distribution of the inner product between common neighbor pairs is statistically higher than that between non-common neighbor pairs.

## D.2 DEGREE-RELATED BIAS OF EVALUATION METRICS

One previous work (Wang & Derr, 2022) has empirically shown the degree-related bias of evaluation metrics used in link prediction models. Following that, we go one step further and theoretically derive the concrete format of the evaluation bias in this section. We leverage an untrained link prediction model to study the bias. This avoids any potential supervision signal from training over observed links and enables us to study the evaluation bias exclusively. Since two nodes with the same degree may end up with different performances, i.e., $\text{X@}K_i \neq \text{X@}K_j, d_i = d_j$, we model $\text{X@}K|d$ as a random variable and expect to find the relationship between its expectation and the node degree $d$, i.e., $f : E(\text{X@}K|d) = f(d)$.

Following many existing ranking works (He et al., 2020; Chen et al., 2021b), we assume without loss of generalizability that the link predictor $\mathscr{P}$ ranking the predicted neighbors based on their embedding similarity with embeddings noted as $\mathbf{E}$, then we have:

**Lemma 1.** *For any untrained embedding-based link predictor $\mathscr{P}$, given the existing $k-1$ predicted neighbors for the node $v_i \in \mathcal{V}$, the $k^{th}$ predicted neighbor is generated by randomly selecting a node without replacement from the remaining nodes with equal opportunities, i.e., $P(v_{\phi_i^k} = v | \{v_{\phi_i^1}, v_{\phi_i^2}, ..., v_{\phi_i^{k-1}}\}) = \frac{1}{N-(k-1)}$.*

Without any training, Lemma 1 trivially holds since embeddings of all nodes are the same, which trivially leads to the following theorem:

**Theorem 2.** *Given the untrained embedding-based link predictor $\mathscr{P}$, the size of the intersection between any node's predicted list $\widetilde{\mathcal{E}}_i$ and its ground-truth list $\widehat{\mathcal{E}}_i$ follows a hypergeometric distribution: $|\widetilde{\mathcal{E}}_i \cap \widehat{\mathcal{E}}_i| \sim \mathcal{HG}(|\mathcal{V}|, K, |\widehat{\mathcal{E}}_i|)$ where $|\mathcal{V}|$ is the population size (the whole node space), $K$ is*

the number of trials and $|\widehat{\mathcal{E}}_i|$ is the number of successful states (the number of node's ground-truth neighbors).

*Proof.* Given the ground-truth node neighbors $\widehat{\mathcal{E}}_i$, the predicted neighbors $\widetilde{\mathcal{E}}_i = \{v_{\phi_i^k}\}_{k=1}^K$ is formed by selecting one node at a time without replacement $K$ times from the whole node space $\mathcal{V}$. Since any selected node $v_{\phi_i^k}$ can be classified into one of two mutually exclusive categories $\widehat{\mathcal{E}}_i$ or $\mathcal{V}\backslash\widehat{\mathcal{E}}_i$ and by Lemma 1, we know that for any untrained link predictor, each unselected node has an equal opportunity to be selected in every new trial, we conclude that $|\widetilde{\mathcal{E}}_i \cap \widehat{\mathcal{E}}_i| \sim \mathcal{HG}(|\mathcal{V}|, K, |\widehat{\mathcal{E}}_i|)$ and by default $E(|\widetilde{\mathcal{E}}_i \cap \widehat{\mathcal{E}}_i|) = |\widetilde{\mathcal{E}}_i|\frac{|\widehat{\mathcal{E}}_i|}{|\mathcal{V}|} = K\frac{|\widehat{\mathcal{E}}_i|}{|\mathcal{V}|}$. □

Furthermore, we present Theorem 3 to state the relationships between the LP performance under each evaluation metric and the node degree:

**Theorem 3.** *Given that $|\widetilde{\mathcal{E}}_i \cap \widehat{\mathcal{E}}_i|$ follows hyper-geometric distribution, we have:*

$$E(\text{R@}K_i|d) = \frac{K}{N}, \frac{\partial E(\text{R@}K|d)}{\partial d} = 0, \tag{12}$$

$$E(\text{P@}K|d_i) = \frac{\alpha d}{N}, \frac{\partial E(\text{P@}K|d)}{\partial d} = \frac{\alpha}{N}, \tag{13}$$

$$E(\text{F1@}K|d) = \frac{2K}{N}\frac{\alpha d}{K+\alpha d}, \frac{\partial E(\text{F1@}K|d)}{\partial d} = \frac{2\alpha K^2}{N}\frac{1}{(K+\alpha d)^2}, \tag{14}$$

$$E(\text{N@}K|d) = \frac{\alpha d}{N}, \frac{\partial E(\text{N@}K|d)}{\partial d} = \frac{\alpha}{N}. \tag{15}$$

*Proof.*

$$E(\text{R@}K_i|d) = E(\frac{|\widetilde{\mathcal{E}}_i \cap \widehat{\mathcal{E}}_i|}{|\widehat{\mathcal{E}}_i|}) = \frac{E(|\widetilde{\mathcal{E}}_i \cap \widehat{\mathcal{E}}_i|)}{|\widehat{\mathcal{E}}_i|} = \frac{\frac{|\widehat{\mathcal{E}}_i|}{|\mathcal{V}|}K}{|\widehat{\mathcal{E}}_i|} = \frac{K}{N} \tag{16}$$

$$E(\text{P@}K_i|d) = E(\frac{|\widetilde{\mathcal{E}}_i \cap \widehat{\mathcal{E}}_i|}{K}) = \frac{E(|\widetilde{\mathcal{E}}_i \cap \widehat{\mathcal{E}}_i|)}{K} = \frac{\frac{|\widehat{\mathcal{E}}_i|}{|\mathcal{V}|}K}{K} = \frac{\alpha d}{N} \tag{17}$$

$$E(\text{F1@}K_i|d) = E(\frac{2|\widetilde{\mathcal{E}}_i \cap \widehat{\mathcal{E}}_i|}{K + |\widehat{\mathcal{E}}_i|}) = \frac{2E(|\widetilde{\mathcal{E}}_i \cap \widehat{\mathcal{E}}_i|)}{K + \alpha d} = \frac{2K}{N}\frac{\alpha d}{K + \alpha d} \tag{18}$$

$$E(\text{N@}K_i|d) = E(\frac{\sum_{k=1}^K \frac{\mathbb{1}[v_{\phi^k} \in (\widetilde{\mathcal{E}}_i \cap \widehat{\mathcal{E}}_i)]}{\log_2(k+1)}}{\sum_{k=1}^K \log_2(k+1)}) = \frac{E(\sum_{k=1}^K \frac{\mathbb{1}[v_{\phi^k} \in (\widetilde{\mathcal{E}}_i \cap \widehat{\mathcal{E}}_i)]}{\log_2(k+1)})}{\sum_{k=1}^K \frac{1}{\log_2(k+1)}} \tag{19}$$

To calculate the numerator DCG, i.e., $E(\sum_{k=1}^K \frac{\mathbb{1}[v_{\phi^k} \in (\widetilde{\mathcal{E}}_i \cap \widehat{\mathcal{E}}_i)]}{\log_2(k+1)})$ in Eq. (19), we model the link prediction procedure as 1) randomly select $K$ nodes from the whole node space $\mathcal{V}$; 2) calculate $|\widetilde{\mathcal{E}}_i \cap \widehat{\mathcal{E}}_i|$, i.e., how many nodes among the selected nodes $\widetilde{\mathcal{E}}_i$ are in the ground-truth neighborhood list $\widehat{\mathcal{E}}_i$; 3) randomly select $|\widetilde{\mathcal{E}}_i \cap \widehat{\mathcal{E}}_i|$ slots to position nodes in $\widetilde{\mathcal{E}}_i \cap \widehat{\mathcal{E}}_i$ and calculate DCG. The above steps can be mathematically formulated as:

$$\sum_{i=0}^K \frac{C(N-\alpha d, K-i)C(\alpha d, i)}{C(N, K)} \sum_{j=1}^{C(K,i)} p(\mathbf{O}_j^{(K,i)}) \sum_{k=1}^K \frac{\mathbb{1}[\mathbf{O}_{jk}^{(K,i)} = 1]}{\log_2(k+1)}, \tag{20}$$

where $\mathbf{O}^{(K,i)} \in \{0,1\}^{C(K,i) \times K}$ represents all $C(K,i)$ possible positional indices of putting $i$ nodes into $K$ candidate slots. Specifically $\mathbf{O}_j^{(K,i)} \in \{0,1\}^K$ indicates the $j^{\text{th}}$ positional configuration of $i$ nodes where $\mathbf{O}_{jk}^{(K,i)} = 1$ if an node is positioned at $k^{\text{th}}$ slot and $\mathbf{O}_{jk}^{(K,i)} = 0$ otherwise. Since our link predictor has no bias in positioning nodes in the K slots by Lemma 1, we have $p(\mathbf{O}_j^{(K,i)}) = \frac{1}{C(K,i)}$ and Eq. (20) can be transformed as:

$$\sum_{i=0}^{K} \frac{C(N-\alpha d, K-i)C(\alpha d, i)}{C(N,K)} \frac{1}{C(K,i)} \sum_{j=1}^{C(K,i)} \sum_{k=1}^{K} \frac{\mathbb{1}[\mathbf{O}_{jk}^{(K,i)} = 1]}{\log_2(k+1)}. \tag{21}$$

We know that only when the $k^{\text{th}}$ slot is positioned a node can we have $\mathbf{O}_{jk}^{(K,i)} = 1$ and among the total $C(K,i)$ selections, every candidate slot $k \in \{1, 2, ..., K\}$ would be selected $C(K-1, i-1)$ times to position a node, which hence leads to:

$$\sum_{j=1}^{C(K,i)} \sum_{k=1}^{K} \frac{\mathbb{1}[\mathbf{O}_{jk}^{(K,i)} = 1]}{\log_2(k+1)} = \sum_{k=1}^{K} \frac{C(K-1, i-1)}{\log_2(k+1)}. \tag{22}$$

We then substitute Eq. (22) into Eq. (21) as:

$$\begin{aligned}
&\sum_{i=0}^{K} \frac{C(N-\alpha d, K-i)C(\alpha d, i)}{C(N,K)} \frac{1}{C(K,i)} \sum_{k=1}^{K} \frac{C(K-1, i-1)}{\log_2(k+1)} \\
&= \sum_{i=0}^{K} \frac{C(N-\alpha d, K-i)C(\alpha d, i)}{C(N,K)} \frac{C(K-1, i-1)}{C(K,i)} \sum_{k=1}^{K} \frac{1}{\log_2(k+1)}.
\end{aligned} \tag{23}$$

Further substituting Eq. (23) into Eq. (19), we finally get:

$$\begin{aligned}
E(\text{N@}K|d_i) &= \sum_{i=0}^{K} \frac{C(N-\alpha d, K-i)C(\alpha d, i)}{C(N,K)} \frac{C(K-1, i-1)}{C(K,i)} \\
&= \sum_{i=0}^{K} \frac{C(N-\alpha d, K-i)C(\alpha d, i)}{C(N,K)} \frac{\frac{(K-1)!}{(i-1)!(K-i)!}}{\frac{K!}{i!(K-i)!}} \\
&= \frac{1}{K} \underbrace{\sum_{i=0}^{K} i \frac{C(N-\alpha d, K-i)C(\alpha d, i)}{C(N,K)}}_{E(|\widetilde{\mathcal{E}}_i \cap \widehat{\mathcal{E}}_i|)} = \frac{1}{K} \frac{\alpha d}{N} * K = \frac{\alpha d}{N}
\end{aligned} \tag{24}$$

$\square$

Based on Theorem 3, Precision, F1, and NDCG increase as node degree increases even when no observed links are used to train the link predictor, which informs the degree-related evaluation bias and causes the illusion that high-degree nodes are more advantageous than low-degree ones observed in some previous works (Li et al., 2021; Rahmani et al., 2022).

### D.3 REWEIGHTING BY LP SCORE ENHANCE 1-LAYER TC

**Theorem 4.** *Taking the normalization term $g(|\mathcal{H}_i^1|, |\mathcal{H}_j^1|) = |\mathcal{H}_i^1|$ and also assume that higher link prediction score $\mathbf{S}_{ij}$ between $v_i$ and its neighbor $v_j$ corresponds to more number of connections between $v_j$ and the neighborhood $\mathcal{N}_i^{\text{Train}}$, i.e., $\mathbf{S}_{ij} > \mathbf{S}_{ik} \to |\mathcal{N}_j^{1,\text{Train}} \cap \mathcal{N}_i^{1,\text{Train}}| > |\mathcal{N}_k^{1,\text{Train}} \cap \mathcal{N}_i^{1,\text{Train}}|, \forall v_j, v_k \in \mathcal{N}_i^{\text{Train},1}$, then we have:*

$$\widehat{C}_i^{1,\text{Train}} = \sum_{v_j \sim \mathcal{N}_i^{\text{Train}}} \frac{\mathbf{S}_{ij}|\mathcal{H}_i^1 \cap \mathcal{H}_j^1|}{|\mathcal{H}_i^1|} \geq \mathbb{E}_{v_j \sim \mathcal{N}_i^{\text{Train}}} \frac{|\mathcal{H}_i^1 \cap \mathcal{H}_j^1|}{|\mathcal{H}_i^1|} = C_i^{1,\text{Train}} \tag{25}$$

*Proof.* By definition, we have $\mathcal{H}_i^1 = \mathcal{N}_i^{1,\text{Train}}$, then the computation of 1-layer $\text{TC}^{\text{Train}}$ is transformed as:

$$C_i^{1,\text{Train}} = \mathbb{E}_{v_j \sim \mathcal{N}_i^{\text{Train}}} I(\mathcal{S}_i^1, \mathcal{S}_j^1) = \mathbb{E}_{v_j \sim \mathcal{N}_i^{\text{Train}}} \frac{|\mathcal{N}_i^{\text{Train}} \cap \mathcal{N}_j^{\text{Train}}|}{|\mathcal{N}_i^{\text{Train}}|} = \frac{1}{|\mathcal{N}_i^{\text{Train}}|} \mathbb{E}_{v_j \sim \mathcal{N}_i^{\text{Train}}} (|\mathcal{N}_i^{\text{Train}} \cap \mathcal{N}_j^{\text{Train}}|). \tag{26}$$

On the other hand, we similarly transform weighted TC as:

$$\widehat{C}_i^{1,\text{Train}} = \frac{1}{|\mathcal{N}_i^{\text{Train}}|} \sum_{v_j \sim \mathcal{N}_i^{\text{Train}}} (\mathbf{S}_{ij} |\mathcal{N}_i^{\text{Train}} \cap \mathcal{N}_j^{\text{Train}}|). \tag{27}$$

By the relation that:

$$\mathbf{S}_{ij} > \mathbf{S}_{ik} \rightarrow |\mathcal{N}_j^{1,\text{Train}} \cap \mathcal{N}_i^{1,\text{Train}}| > |\mathcal{N}_k^{1,\text{Train}} \cap \mathcal{N}_i^{1,\text{Train}}|, \forall v_j, v_k \in \mathcal{N}_i^{\text{Train},1}, \tag{28}$$

Then we have:

$$\widehat{C}_i^{1,\text{Train}} \geq C_i^{1,\text{Train}} \tag{29}$$

$\square$

Moreover, we include Figure 9 to illustrate the idea of enhancing TC via assigning higher weights to edges connecting neighbors that have higher connections to the whole neighborhoods. We can see in this case, weighted TC in Figure 9(a) is naturally higher than the one in Figure 9(b)

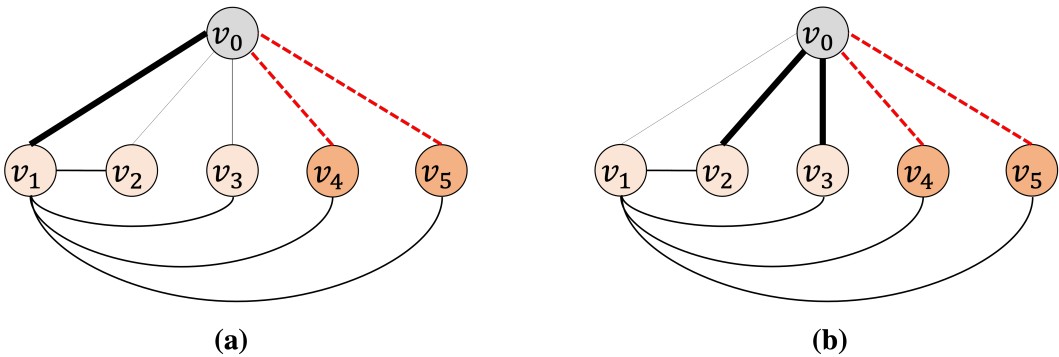

**(a)**        **(b)**

Figure 9: (a) Increasing the weight of neighbors that have more connections with the whole neighborhood while (b) increasing the weight of neighbors that have fewer connections with the whole neighborhood. (a) would increase the weighted TC while (b) would not

## E  EXAMPLE DEMONSTRATING THE ADVANTAGES OF TC OVER LCC

According to the definition of local clustering coefficient (LCC) and TC, we respectively calculate their values for node $v_1$ in Figure 10. $v_2, v_3, v_4$ do not have any connection among themselves, indicating node $v_1$ prefer interacting with nodes coming from significantly different domain/community. Subsequently, the incoming neighbors $v_5, v_6$ of $v_1$ are likely to also come from other communities and hence share no connections with $v_2, v_3, v_4$, which leads to the ill topological condition for predicting links of $v_1$. However, in this case, the clustering coefficient still maintains 0.5 because of the connections between $v_1$ and $v_2/v_3/v_4$, which cannot precisely capture the ill-topology of $v_1$ in this case. Conversely, our $\text{TC}^{\text{Train}}$ equals 0, reflecting the ill topological condition of $v_1$.

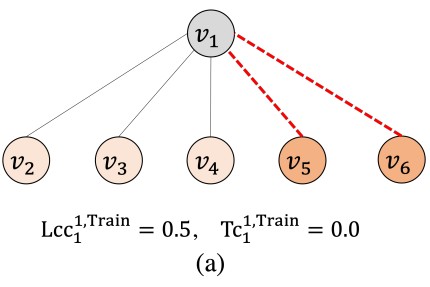

$$\text{Lcc}_1^{1,\text{Train}} = 0.5, \quad \text{Tc}_1^{1,\text{Train}} = 0.0$$

(a)

Figure 10: Comparison of TC and LCC

# F    DATASETS AND EXPERIMENTAL SETTINGS

This section introduces datasets and experimental settings used in this paper.

## F.1    DATASET INTRODUCTION AND STATISTICS

We use five widely employed datasets for evaluating the link prediction task, including four citation networks: Cora, Citeseer, Pubmed, and Citation2, and 1 human social network Collab. We further introduce two real-world animal social networks, Reptile and Vole, based on animal interactions.

- **Cora/Citeseer/Pubmed**: Following (Zhao et al., 2022; Chamberlain et al., 2022; Wang et al., 2023), we randomly split edges into 70%/10%/20% so that there is no topological distribution shift in these datasets. We use Hits@100 to evaluate the final performance.

- **Collab/Citation2**: We leverage the default edge splitting from OGBL (Hu et al., 2020). These two datasets mimic the real-life link prediction scenario where testing edges later joined in the network than validation edges and further than training edges. This would cause the topological distribution shift observed in the **Obs.3** of Section 3.3. For Collab, different from (Chamberlain et al., 2022; Wang et al., 2023), our setting does not allow validation edges to join the network for message-passing when evaluating link prediction performance. Therefore, the edges used for message-passing and supervision come from edges in the training set. In addition, we also consider a widely used setting in prior work where the validation edges would be allowed in message-passing when evaluating in the testing stage and we term this one on Collab as Collab* (Wang et al., 2023; Chamberlain et al., 2022).

- **Reptile/Vole**: we obtain the dataset from Network Repository (Rossi & Ahmed, 2015). To construct this network, a bipartite network was first constructed based on burrow use - an edge connecting a tortoise node to a burrow node indicated a burrow used by the individual. Social networks of desert tortoises were then constructed by the bipartite network into a single-mode projection of tortoise nodes. Node features are initialized by a trainable embedding layer, and we leverage the same edge splitting 70%/10%/20% for training/validation/testing.

Table 3: Statistic of datasets used for evaluating link prediction.

| Network Domain | Dataset | # Nodes | # Edges | Split Type | Metric | Split Ratio |
|---|---|---|---|---|---|---|
| Citation Network | Cora | 2,708 | 5,278 | Random | Hits@100 | 70/10/20% |
| | Citeseer | 3,327 | 4,676 | Random | Hits@100 | 70/10/20% |
| | Pubmed | 18,717 | 44,327 | Random | Hits@100 | 70/10/20% |
| | Citation2 | 2,927,963 | 30,561,187 | Time | MRR | Default |
| Social Network | Collab | 235,868 | 1,285,465 | Time | Hits@50 | Default |
| Animal Network | Reptile | 787 | 1232 | Random | Hits@100 | 70/10/20% |
| | Vole | 1480 | 3935 | Random | Hits@100 | 70/10/20% |

## F.2    HYPERPARAMETER DETAILS

For all experiments, we select the best configuration on validation edges and report the model performance on testing edges. The search space for the hyperparameters of the GCN/SAGE/LightGCN baselines and their augmented variants $GCN_{rw}$/$SAGE_{rw}$ are: graph convolutional layer $\{1, 2, 3\}$, hidden dimension of graph encoder $\{64, 128, 256\}$, the learning rate of the encoder and predictor $\{0.001, 0.005, 0.01\}$, dropout $\{0.2, 0.5, 0.8\}$, training epoch $\{50, 100, 500, 1000\}$, batch size $\{256, 1152, 64 * 1024\}$ (Hu et al., 2020; Chamberlain et al., 2022; Wang et al., 2023), weights $\alpha \in \{0.5, 1, 2, 3, 4\}$, the update interval $\tau \in \{1, 2, 10, 20, 50\}$, warm up epochs $T^{warm} \in \{1, 2, 5, 10, 30, 50\}$. For baseline NCN[2], we directly run their code using their default best-performing configurations on Cora/Citeseer/Pubmed/Collab but for Citation2, due to memory limitation, we directly take the result from the original paper. We use cosine similarity metric as the similarity function $\phi$ in computing ATC.

---

[2]https://github.com/GraphPKU/NeuralCommonNeighbor

# G ADDITIONAL RESULTS

To demonstrate that the observations made previously in Section 3 can also generalize to other datasets, here we present the comprehensive results on all datasets we study in this paper as follows.

## G.1 LINK PREDICTION PERFORMANCE GROUPED BY $TC^{\text{Test}}$

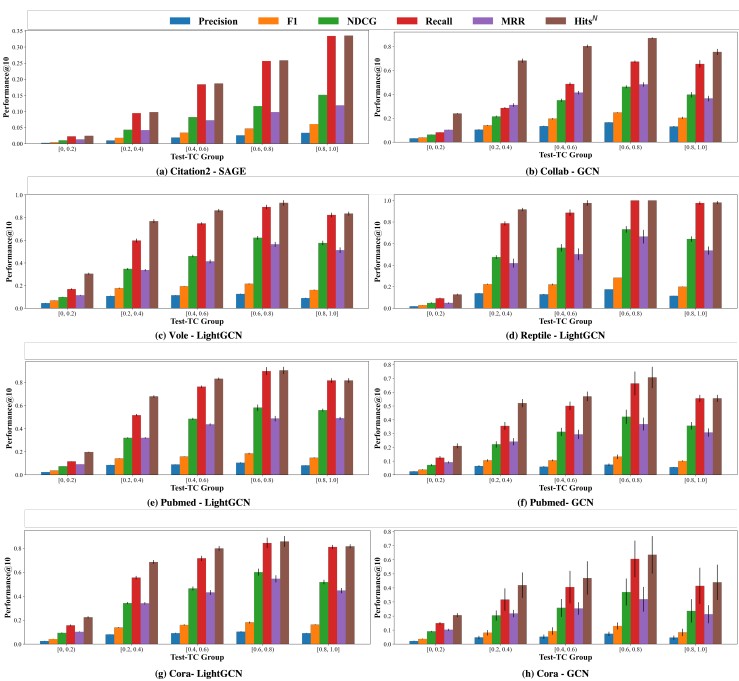

Figure 11: LP performance grouped by $TC^{\text{Test}}$ for all nodes

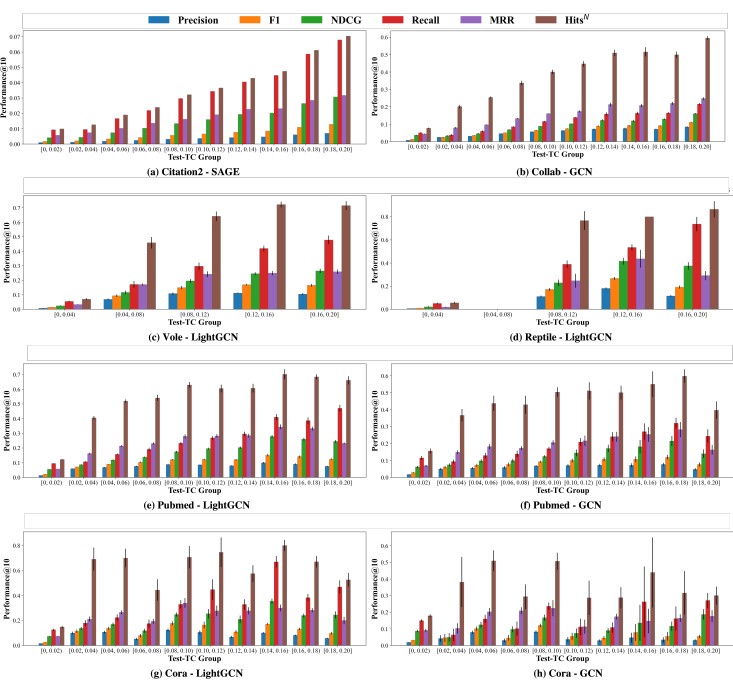

Figure 12: LP performance grouped by $TC^{\text{Test}}$ for low $TC^{\text{Test}}$ nodes

## G.2 LINK PREDICTION PERFORMANCE GROUPED BY TC$^{\text{Train}}$

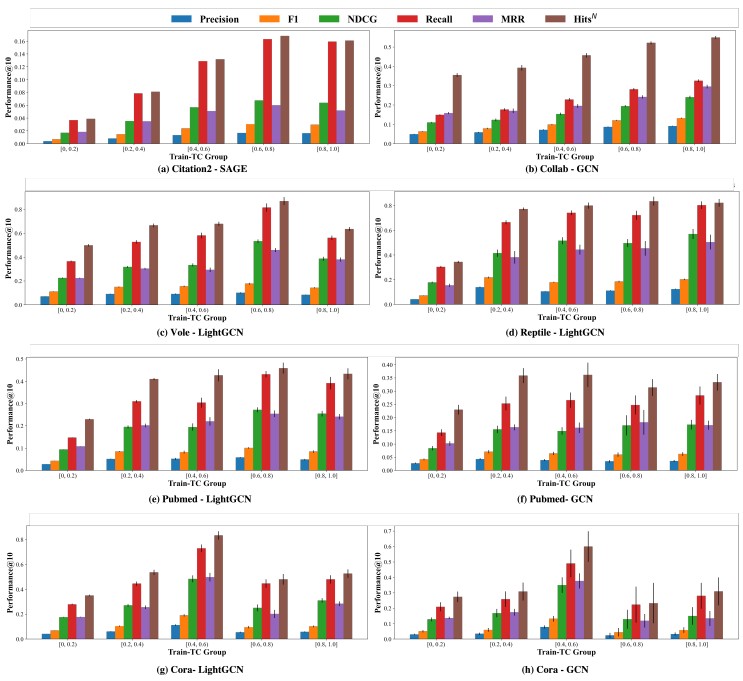

Figure 13: LP performance grouped by TC$^{\text{Train}}$ for all nodes

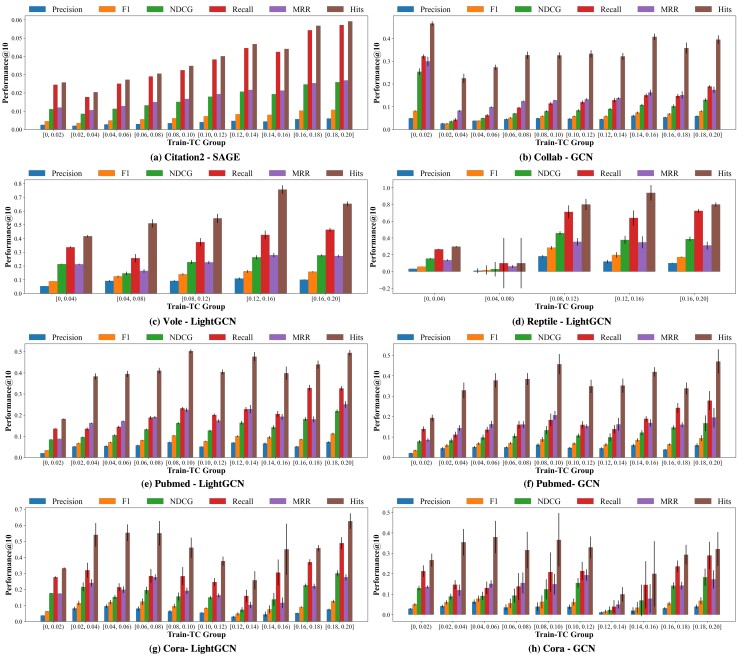

Figure 14: LP performance grouped by TC$^{\text{Train}}$ for low TC$^{\text{Train}}$ nodes

## G.3 LINK PREDICTION PERFORMANCE GROUPED BY DEGREE[Test]

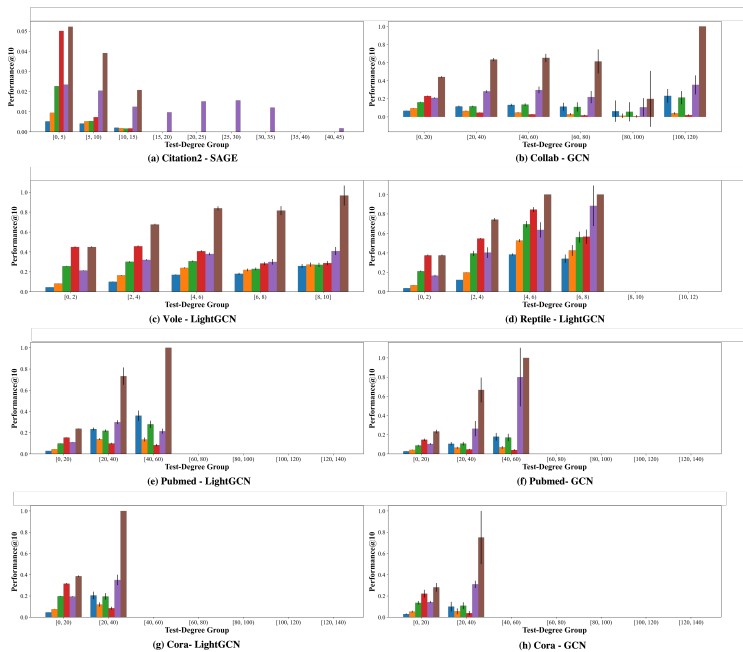

Figure 15: LP performance grouped by Degree[Test] for all nodes

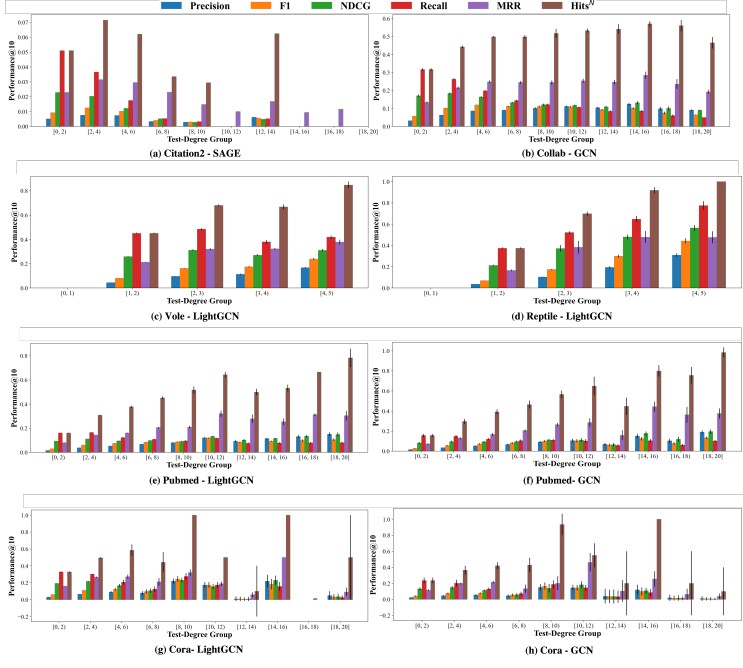

Figure 16: LP performance grouped by Degree[Test] for low Test-Degree nodes

## G.4 LINK PREDICTION PERFORMANCE GROUPED BY DEGREE$^{\text{Train}}$

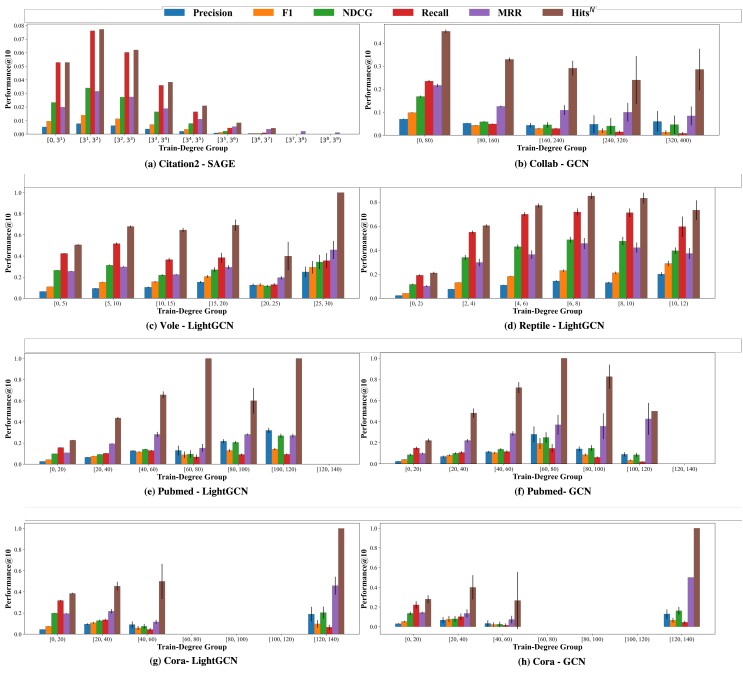

Figure 17: LP performance grouped by Degree$^{\text{Train}}$ for all nodes

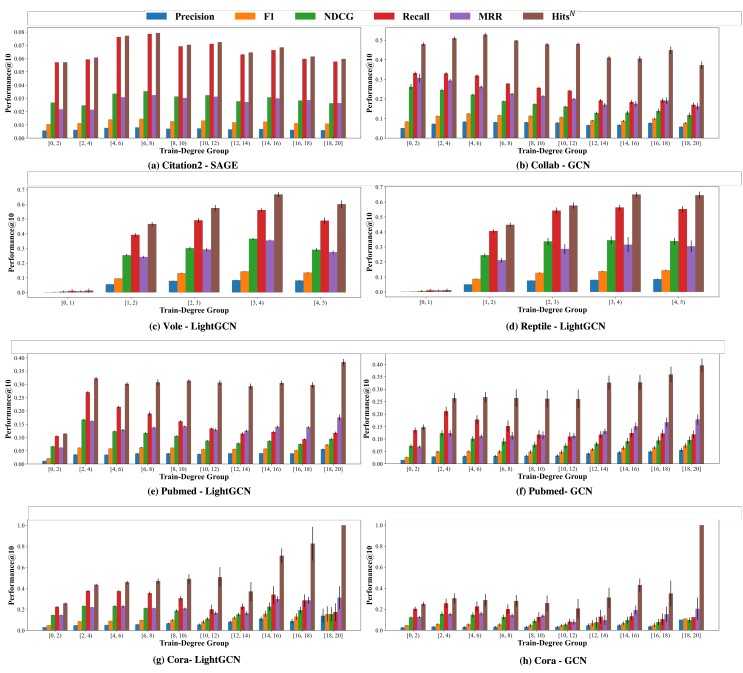

Figure 18: LP performance grouped by Degree$^{\text{Train}}$ for low Degree$^{\text{Train}}$ nodes

## G.5 RELATION BETWEEN LP PERFORMANCE AND TC AT GRAPH-LEVEL

Figure 19: Relation between LP performance and TC at Graph-level

## G.6 RELATION BETWEEN $TC^{Train}$ AND $TC^{Test}$

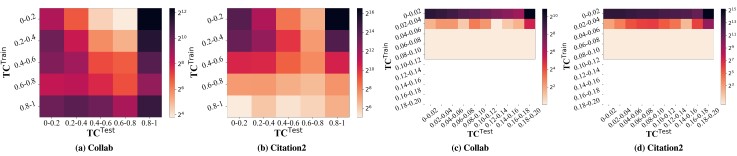

Figure 20: Relation between $TC^{Train}$ and $TC^{Test}$ on Collab/Citation2

## G.7 DIFFERENCE IN TC VS DIFFERENCE IN PERFORMANCE BEFORE/AFTER APPLYING REWEIGHTING

## G.8 CORRELATION OF THE PERFORMANCE WITH TC AND DEGREE

Here we present the comprehensive correlation of the performance with $TC^{Train}/TC^{Val}/TC^{Test}$ and $Degree^{Train}$. As the performance is evaluated under different K, we further define the absolute average/the typical average correlation across different K values to reflect the absolute correlation strength/the consistency of the correlation average:

$$\text{Absolute Avg.}_{X@K} = \frac{1}{4} \sum_{k \in \{5,10,20,50\}} |X@k|, \quad \text{Basic Avg.}_{X@K} = \frac{1}{4} \sum_{k \in \{5,10,20,50\}} X@k$$

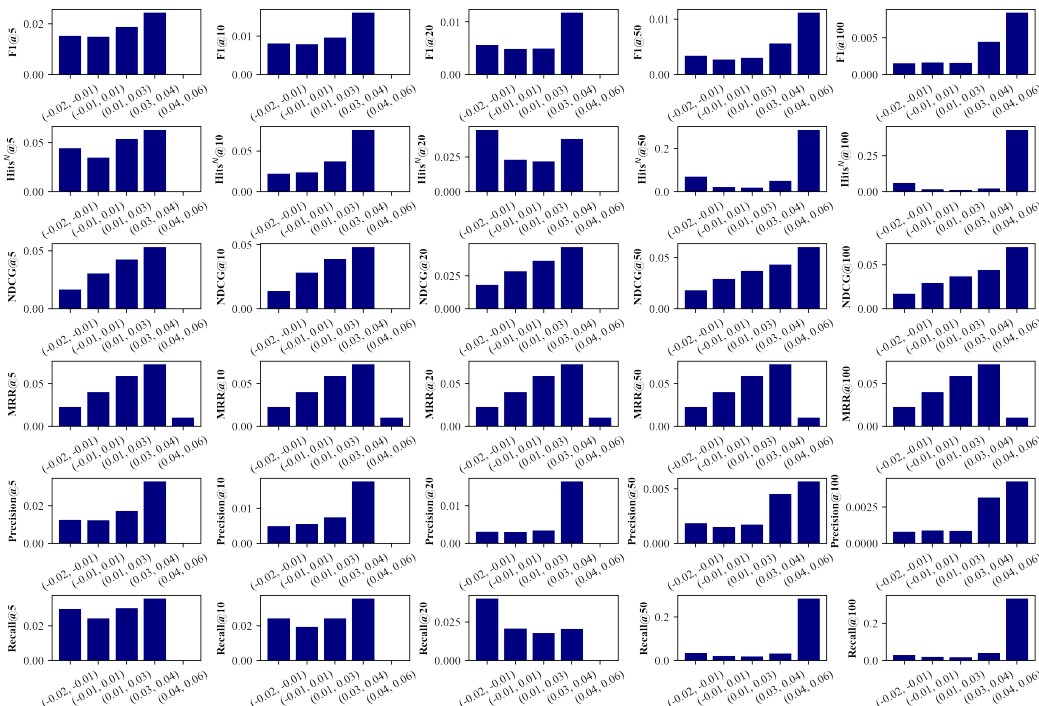

Figure 21: Relation between TC$^{\text{Train}}$ and TC$^{\text{Test}}$ on Collab by running GCN

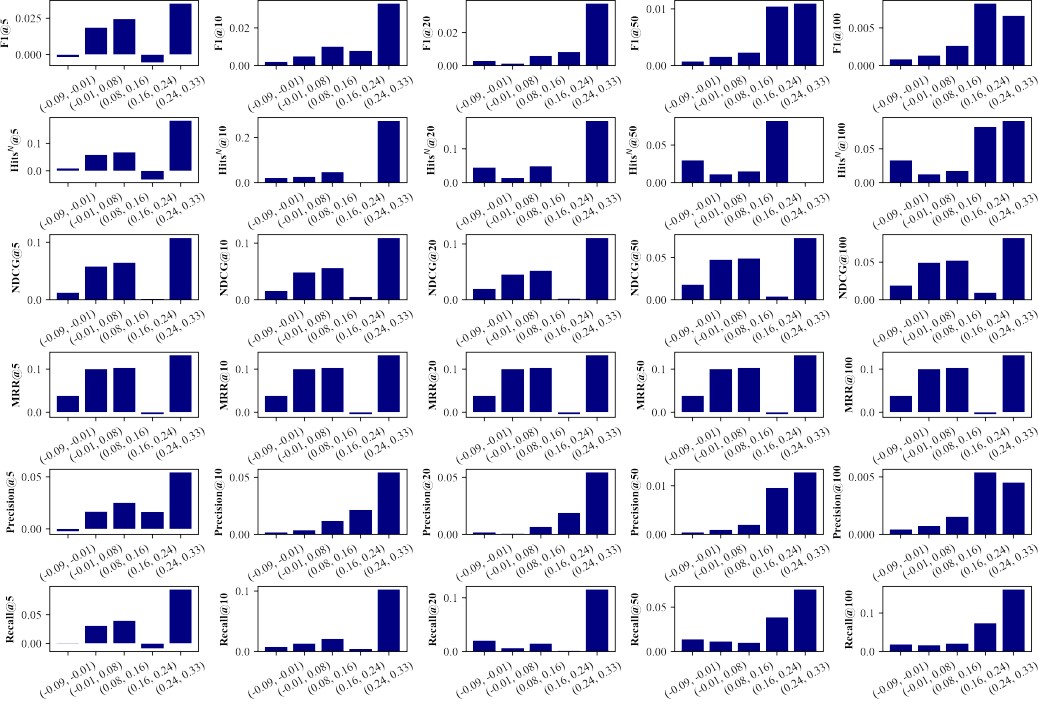

Figure 22: Relation between TC$^{\text{Train}}$ and TC$^{\text{Test}}$ on Collab by running SAGE

Table 4: The correlation between $TC^{Train}/TC^{Val}/TC^{Test}/Degree^{Train}$ and the GCN's LP performance on **Collab**. We note that the formal definitions of Absolute Avg. and Basic Avg. are provided in Section G.8 and they represent the average absolute and simple average correlation, respectively, across the range of @K for the given metric; these are also then calculated overall.

| | Metric | @5 | @10 | @20 | @50 | Absolute Avg. | Basic Avg. |
|---|---|---|---|---|---|---|---|
| $TC^{Train}$ | Precision | 0.2252 | 0.1925 | 0.1353 | 0.0578 | 0.1527 | 0.1527 |
| | F1 | 0.2601 | 0.2364 | 0.1733 | 0.0790 | 0.1872 | 0.1872 |
| | NDCG | 0.2279 | 0.2427 | 0.2375 | 0.2206 | 0.2322 | 0.2322 |
| | Recall | 0.2296 | 0.2358 | 0.2156 | 0.1754 | 0.2141 | 0.2141 |
| | $Hits^N$ | 0.2057 | 0.1800 | 0.1328 | 0.0717 | 0.1476 | 0.1476 |
| | MRR | | 0.2044 | | | 0.2044 | 0.2044 |
| | | | | | | 0.1867 | 0.1867 |
| $TC^{Val}$ | Precision | 0.2573 | 0.2832 | 0.2788 | 0.2387 | 0.2645 | 0.2645 |
| | F1 | 0.2425 | 0.2901 | 0.2991 | 0.2641 | 0.2740 | 0.2740 |
| | NDCG | 0.2066 | 0.2330 | 0.2521 | 0.2624 | 0.2385 | 0.2385 |
| | Recall | 0.1742 | 0.2179 | 0.2428 | 0.2514 | 0.2216 | 0.2216 |
| | $Hits^N$ | 0.2445 | 0.2674 | 0.2720 | 0.2620 | 0.2615 | 0.2615 |
| | MRR | | 0.2350 | | | 0.2350 | 0.2350 |
| | | | | | | 0.2520 | 0.2520 |
| $TC^{Test}$ | Precision | 0.5184 | 0.5437 | 0.5107 | 0.4127 | 0.4964 | 0.4964 |
| | F1 | 0.5858 | 0.6311 | 0.5964 | 0.4799 | 0.5733 | 0.5733 |
| | NDCG | 0.5443 | 0.6282 | 0.6706 | 0.6902 | 0.6333 | 0.6333 |
| | Recall | 0.5644 | 0.6753 | 0.7324 | 0.7533 | 0.6814 | 0.6814 |
| | $Hits^N$ | 0.5272 | 0.5816 | 0.5924 | 0.5720 | 0.5683 | 0.5683 |
| | MRR | | 0.5085 | | | 0.5085 | 0.5085 |
| | | | | | | 0.5905 | 0.5905 |
| $Degree^{Train}$ | Precision | -0.1261 | -0.0829 | 0.0006 | 0.1440 | 0.0884 | -0.0161 |
| | F1 | -0.1997 | -0.1663 | -0.0813 | 0.0812 | 0.1321 | -0.0915 |
| | NDCG | -0.1822 | -0.2017 | -0.1985 | -0.1750 | 0.1894 | -0.1894 |
| | Recall | -0.2183 | -0.2288 | -0.2118 | -0.1681 | 0.2068 | -0.2068 |
| | $Hits^N$ | -0.1395 | -0.1164 | -0.0658 | 0.0055 | 0.0818 | -0.0791 |
| | MRR | | -0.1349 | | | -0.1349 | -0.1349 |
| | | | | | | 0.1397 | -0.1166 |
| $Degree^{Val}$ | Precision | 0.0047 | 0.0472 | 0.1117 | 0.2141 | 0.0944 | 0.0944 |
| | F1 | -0.0823 | -0.0469 | 0.0200 | 0.1416 | 0.0727 | 0.0081 |
| | NDCG | -0.0608 | -0.0803 | -0.0838 | -0.0736 | 0.0746 | -0.0746 |
| | Recall | -0.1203 | -0.1296 | -0.1269 | -0.1100 | 0.1217 | -0.1217 |
| | $Hits^N$ | -0.0063 | 0.0171 | 0.0481 | 0.0848 | 0.0391 | 0.0359 |
| | MRR | | -0.0108 | | | -0.0108 | -0.0108 |
| | | | | | | 0.0805 | -0.0116 |
| $Degree^{Test}$ | Precision | 0.1075 | 0.1833 | 0.2924 | 0.4617 | 0.2612 | 0.2612 |
| | F1 | -0.0669 | 0.0043 | 0.1249 | 0.3375 | 0.1334 | 0.1000 |
| | NDCG | -0.034 | -0.0723 | -0.0814 | -0.0668 | 0.0636 | -0.0636 |
| | Recall | -0.1678 | -0.1856 | -0.187 | -0.1724 | 0.1782 | -0.1782 |
| | $Hits^N$ | 0.0785 | 0.1103 | 0.1407 | 0.1718 | 0.1253 | 0.1253 |
| | MRR | | 0.0727 | | | 0.0727 | 0.0727 |
| | | | | | | 0.1524 | 0.0489 |
| Subgraph Density | Precision | 0.2199 | 0.1646 | 0.0875 | -0.0073 | 0.1198 | 0.1162 |
| | F1 | 0.2806 | 0.2259 | 0.1353 | 0.0161 | 0.1645 | 0.1645 |
| | NDCG | 0.2811 | 0.2891 | 0.2748 | 0.2491 | 0.2735 | 0.2735 |
| | Recall | 0.2911 | 0.2783 | 0.2399 | 0.1834 | 0.2482 | 0.2482 |
| | $Hits^N$ | 0.2265 | 0.1842 | 0.1196 | 0.0423 | 0.1432 | 0.1432 |
| | MRR | | 0.2331 | | | 0.2331 | 0.2331 |
| | | | | | | 0.1898 | 0.1891 |

Table 5: The correlation between $TC^{Train}/TC^{Val}/TC^{Test}/Degree^{Train}$ and the GCN's LP performance on **Citation2**. We note that the formal definitions of Absolute Avg. and Basic Avg. are provided in Section G.8 and they represent the average absolute and simple average correlation, respectively, across the range of @K for the given metric; these are also then calculated overall.

| | Metric | @5 | @10 | @20 | @50 | Absolute Avg. | Basic Avg. |
|---|---|---|---|---|---|---|---|
| $TC^{Train}$ | Precision | 0.0839 | 0.1312 | 0.1784 | 0.2157 | 0.1523 | 0.1523 |
| | F1 | 0.0849 | 0.1323 | 0.1795 | 0.2165 | 0.1533 | 0.1533 |
| | NDCG | 0.0773 | 0.1164 | 0.1585 | 0.2012 | 0.1384 | 0.1384 |
| | Recall | 0.0860 | 0.1346 | 0.1845 | 0.2265 | 0.1579 | 0.1579 |
| | $Hits^N$ | 0.0840 | 0.1314 | 0.1791 | 0.2182 | 0.1532 | 0.1532 |
| | MRR | | 0.1229 | | | 0.1229 | 0.1229 |
| | | | | | | 0.1510 | 0.1510 |
| $TC^{Val}$ | Precision | 0.0575 | 0.0868 | 0.1200 | 0.1479 | 0.1031 | 0.1031 |
| | F1 | 0.0581 | 0.0874 | 0.1206 | 0.1484 | 0.1036 | 0.1036 |
| | NDCG | 0.0545 | 0.0790 | 0.1078 | 0.1377 | 0.0948 | 0.0948 |
| | Recall | 0.0586 | 0.0884 | 0.1231 | 0.1540 | 0.1060 | 0.1060 |
| | $Hits^N$ | 0.0574 | 0.0870 | 0.1206 | 0.1500 | 0.1038 | 0.1038 |
| | MRR | | 0.0846 | | | 0.0846 | 0.0846 |
| | | | | | | 0.1022 | 0.1022 |
| $TC^{Test}$ | Precision | 0.1797 | 0.2541 | 0.3313 | 0.3996 | 0.2912 | 0.2912 |
| | F1 | 0.1812 | 0.2558 | 0.3328 | 0.4008 | 0.2927 | 0.2927 |
| | NDCG | 0.1706 | 0.2365 | 0.3071 | 0.3825 | 0.2742 | 0.2742 |
| | Recall | 0.1829 | 0.2599 | 0.3401 | 0.4141 | 0.2993 | 0.2993 |
| | $Hits^N$ | 0.1797 | 0.2550 | 0.3331 | 0.4048 | 0.2932 | 0.2932 |
| | MRR | | 0.2512 | | | 0.2512 | 0.2512 |
| | | | | | | 0.2901 | 0.2901 |
| $Degree^{Train}$ | Precision | -0.0288 | -0.0406 | -0.0536 | -0.0689 | 0.0480 | -0.0480 |
| | F1 | -0.0295 | -0.0415 | -0.0546 | -0.0699 | 0.0489 | -0.0489 |
| | NDCG | -0.0285 | -0.0394 | -0.0522 | -0.0692 | 0.0473 | -0.0473 |
| | Recall | -0.0305 | -0.0436 | -0.0589 | -0.0791 | 0.0530 | -0.0530 |
| | $Hits^N$ | -0.0289 | -0.0408 | -0.0540 | -0.0708 | 0.0486 | -0.0486 |
| | MRR | | -0.0421 | | | -0.0421 | -0.0421 |
| | | | | | | 0.0492 | 0.0492 |
| $Degree^{Val}$ | Precision | 0.0161 | 0.0229 | 0.0300 | 0.0393 | 0.0271 | 0.0271 |
| | F1 | 0.0156 | 0.0220 | 0.0289 | 0.0381 | 0.0262 | 0.0262 |
| | NDCG | 0.0150 | 0.0199 | 0.0248 | 0.0305 | 0.0226 | 0.0226 |
| | Recall | 0.0150 | 0.0203 | 0.0252 | 0.0301 | 0.0227 | 0.0227 |
| | $Hits^N$ | 0.0161 | 0.0232 | 0.0300 | 0.0384 | 0.0269 | 0.0269 |
| | MRR | | 0.0234 | | | 0.0234 | 0.0234 |
| | | | | | | 0.0251 | 0.0251 |
| $Degree^{Test}$ | Precision | 0.0060 | 0.0113 | 0.0190 | 0.0364 | 0.0182 | 0.0182 |
| | F1 | -0.0009 | 0.0047 | 0.0128 | 0.0314 | 0.0125 | 0.0120 |
| | NDCG | -0.0086 | -0.0113 | -0.0147 | -0.0185 | 0.0133 | -0.0133 |
| | Recall | -0.0135 | -0.0185 | -0.0251 | -0.0344 | 0.0229 | -0.0229 |
| | $Hits^N$ | 0.0051 | 0.0081 | 0.0120 | 0.0159 | 0.0103 | 0.0103 |
| | MRR | | 0.0104 | | | 0.0104 | 0.0104 |
| | | | | | | 0.0154 | 0.0009 |
| Subgraph Density | Precision | 0.0809 | 0.1217 | 0.1607 | 0.1916 | 0.1387 | 0.1387 |
| | F1 | 0.0823 | 0.1231 | 0.1621 | 0.1926 | 0.1400 | 0.1400 |
| | NDCG | 0.0761 | 0.1111 | 0.1476 | 0.1853 | 0.1300 | 0.1300 |
| | Recall | 0.0842 | 0.1268 | 0.1691 | 0.2063 | 0.1466 | 0.1466 |
| | $Hits^N$ | 0.0811 | 0.1219 | 0.1618 | 0.1956 | 0.1401 | 0.1401 |
| | MRR | | 0.1144 | | | 0.1144 | 0.1144 |
| | | | | | | 0.1391 | 0.1391 |

Table 6: The correlation between $TC^{Train}/TC^{Val}/TC^{Test}/Degree^{Train}$ and the GCN's LP performance on **Cora**. We note that the formal definitions of Absolute Avg. and Basic Avg. are provided in Section G.8 and they represent the average absolute and simple average correlation, respectively, across the range of @K for the given metric; these are also then calculated overall.

| | Metric | @5 | @10 | @20 | @50 | Absolute Avg. | Basic Avg. |
|---|---|---|---|---|---|---|---|
| $TC^{Train}$ | Precision | 0.0985 | 0.1046 | 0.1238 | 0.1571 | 0.1210 | 0.1210 |
| | F1 | 0.0989 | 0.1042 | 0.1239 | 0.1597 | 0.1217 | 0.1217 |
| | NDCG | 0.0933 | 0.0990 | 0.1088 | 0.1306 | 0.1079 | 0.1079 |
| | Recall | 0.1020 | 0.1042 | 0.1162 | 0.1568 | 0.1198 | 0.1198 |
| | $Hits^{N}$ | 0.0961 | 0.1000 | 0.1226 | 0.1617 | 0.1201 | 0.1201 |
| | MRR | | 0.0869 | | | 0.0869 | 0.0869 |
| | | | | | | 0.1181 | 0.1181 |
| $TC^{Val}$ | Precision | 0.0342 | 0.0456 | 0.0840 | 0.0903 | 0.0635 | 0.0635 |
| | F1 | 0.0296 | 0.0406 | 0.0820 | 0.0907 | 0.0607 | 0.0607 |
| | NDCG | 0.0215 | 0.0259 | 0.0446 | 0.0526 | 0.0362 | 0.0362 |
| | Recall | 0.0257 | 0.0322 | 0.0724 | 0.0841 | 0.0536 | 0.0536 |
| | $Hits^{N}$ | 0.0331 | 0.0413 | 0.0742 | 0.0932 | 0.0605 | 0.0605 |
| | MRR | | 0.0291 | | | 0.0291 | 0.0291 |
| | | | | | | 0.0549 | 0.0549 |
| $TC^{Test}$ | Precision | 0.4694 | 0.4702 | 0.4667 | 0.3977 | 0.4510 | 0.4510 |
| | F1 | 0.4952 | 0.4964 | 0.4948 | 0.4216 | 0.4770 | 0.4770 |
| | NDCG | 0.4970 | 0.5239 | 0.5551 | 0.5759 | 0.5380 | 0.5380 |
| | Recall | 0.4941 | 0.5109 | 0.5448 | 0.5347 | 0.5211 | 0.5211 |
| | $Hits^{N}$ | 0.4749 | 0.4909 | 0.5130 | 0.4866 | 0.4914 | 0.4914 |
| | MRR | | 0.4920 | | | 0.4920 | 0.4920 |
| | | | | | | 0.4957 | 0.4957 |
| $Degree^{Train}$ | Precision | 0.0751 | 0.0970 | 0.1701 | 0.3268 | 0.1673 | 0.1673 |
| | F1 | -0.0039 | 0.0237 | 0.0938 | 0.2549 | 0.0941 | 0.0921 |
| | NDCG | -0.0156 | -0.0276 | -0.0283 | -0.0191 | 0.0227 | -0.0227 |
| | Recall | -0.0432 | -0.0547 | -0.0568 | -0.0529 | 0.0519 | -0.0519 |
| | $Hits^{N}$ | 0.0656 | 0.0650 | 0.0862 | 0.1135 | 0.0826 | 0.0826 |
| | MRR | | 0.0307 | | | 0.0307 | 0.0307 |
| | | | | | | 0.0837 | 0.0837 |
| $Degree^{Val}$ | Precision | 0.0433 | 0.0623 | 0.1138 | 0.2248 | 0.1111 | 0.1111 |
| | F1 | -0.0230 | 0.0012 | 0.0508 | 0.1634 | 0.0596 | 0.0481 |
| | NDCG | -0.0235 | -0.0336 | -0.0369 | -0.0361 | 0.0325 | -0.0325 |
| | Recall | -0.0570 | -0.0648 | -0.0689 | -0.0784 | 0.0673 | -0.0673 |
| | $Hits^{N}$ | 0.0253 | 0.0222 | 0.0308 | 0.0431 | 0.0304 | 0.0304 |
| | MRR | | 0.0144 | | | 0.0144 | 0.0144 |
| | | | | | | 0.0602 | 0.0179 |
| $Degree^{Test}$ | Precision | 0.1669 | 0.2104 | 0.3046 | 0.4890 | 0.2927 | 0.2927 |
| | F1 | 0.0537 | 0.1111 | 0.2127 | 0.4149 | 0.1981 | 0.1981 |
| | NDCG | 0.0004 | -0.0104 | -0.0082 | 0.0060 | 0.0063 | -0.0031 |
| | Recall | -0.0599 | -0.0702 | -0.0760 | -0.0781 | 0.0711 | -0.0711 |
| | $Hits^{N}$ | 0.1406 | 0.1487 | 0.1624 | 0.1865 | 0.1596 | 0.1596 |
| | MRR | | 0.1116 | | | 0.1116 | 0.1116 |
| | | | | | | 0.1455 | 0.1153 |
| Subgraph Density | Precision | 0.0794 | 0.0900 | 0.0796 | 0.0381 | 0.0718 | 0.0718 |
| | F1 | 0.1088 | 0.1189 | 0.1066 | 0.0580 | 0.0981 | 0.0981 |
| | NDCG | 0.1157 | 0.1378 | 0.1543 | 0.1674 | 0.1438 | 0.1438 |
| | Recall | 0.1330 | 0.1690 | 0.2015 | 0.2272 | 0.1827 | 0.1827 |
| | $Hits^{N}$ | 0.0851 | 0.1109 | 0.1257 | 0.1385 | 0.1151 | 0.1151 |
| | MRR | | 0.0976 | | | 0.0976 | 0.0976 |
| | | | | | | 0.1223 | 0.1223 |

Table 7: The correlation between TC$^{\text{Train}}$/TC$^{\text{Val}}$/TC$^{\text{Test}}$/Degree$^{\text{Train}}$ and the GCN's LP performance on **Citeseer**. We note that the formal definitions of Absolute Avg. and Basic Avg. are provided in Section G.8 and they represent the average absolute and simple average correlation, respectively, across the range of @K for the given metric; these are also then calculated overall.

|  | Metric | @5 | @10 | @20 | @50 | Absolute Avg. | Basic Avg. |
|---|---|---|---|---|---|---|---|
| TC$^{\text{Train}}$ | Precision | 0.3330 | 0.3735 | 0.3898 | 0.3830 | 0.3698 | 0.3698 |
|  | F1 | 0.3324 | 0.3803 | 0.4056 | 0.4049 | 0.3808 | 0.3808 |
|  | NDCG | 0.2831 | 0.3226 | 0.3570 | 0.3879 | 0.3377 | 0.3377 |
|  | Recall | 0.3001 | 0.3481 | 0.3920 | 0.4295 | 0.3674 | 0.3674 |
|  | Hits$^N$ | 0.3386 | 0.3901 | 0.4287 | 0.4603 | 0.4044 | 0.4044 |
|  | MRR | | 0.3194 | | | 0.3194 | 0.3194 |
|  | | | | | | 0.3720 | 0.3720 |
| TC$^{\text{Val}}$ | Precision | 0.2796 | 0.2962 | 0.3224 | 0.3229 | 0.3053 | 0.3053 |
|  | F1 | 0.2756 | 0.2947 | 0.3291 | 0.3365 | 0.3090 | 0.3090 |
|  | NDCG | 0.2508 | 0.2662 | 0.2929 | 0.3118 | 0.2804 | 0.2804 |
|  | Recall | 0.2491 | 0.2585 | 0.2928 | 0.3086 | 0.2773 | 0.2773 |
|  | Hits$^N$ | 0.2801 | 0.3049 | 0.338 | 0.3496 | 0.3182 | 0.3182 |
|  | MRR | | 0.2763 | | | 0.2763 | 0.2763 |
|  | | | | | | 0.2980 | 0.2980 |
| TC$^{\text{Test}}$ | Precision | 0.6786 | 0.698 | 0.6745 | 0.6220 | 0.6683 | 0.6683 |
|  | F1 | 0.7157 | 0.7385 | 0.7207 | 0.6678 | 0.7107 | 0.7107 |
|  | NDCG | 0.7037 | 0.7540 | 0.7946 | 0.8300 | 0.7706 | 0.7706 |
|  | Recall | 0.7299 | 0.7797 | 0.8258 | 0.8588 | 0.7986 | 0.7986 |
|  | Hits$^N$ | 0.7127 | 0.7595 | 0.7979 | 0.8216 | 0.7729 | 0.7729 |
|  | MRR | | 0.7070 | | | 0.7070 | 0.7070 |
|  | | | | | | 0.7442 | 0.7442 |
| Degree$^{\text{Train}}$ | Precision | 0.2472 | 0.3523 | 0.4591 | 0.5861 | 0.4112 | 0.4112 |
|  | F1 | 0.1867 | 0.2727 | 0.3872 | 0.5408 | 0.3469 | 0.3469 |
|  | NDCG | 0.1303 | 0.1645 | 0.2022 | 0.2475 | 0.1861 | 0.1861 |
|  | Recall | 0.1144 | 0.1532 | 0.2047 | 0.2591 | 0.1829 | 0.1829 |
|  | Hits$^N$ | 0.2538 | 0.3181 | 0.3581 | 0.3886 | 0.3297 | 0.3297 |
|  | MRR | | 0.2227 | | | 0.2227 | 0.2227 |
|  | | | | | | 0.2913 | 0.2913 |
| Degree$^{\text{Val}}$ | Precision | 0.1431 | 0.1866 | 0.2255 | 0.277 | 0.2081 | 0.2081 |
|  | F1 | 0.1147 | 0.1582 | 0.2053 | 0.2693 | 0.1869 | 0.1869 |
|  | NDCG | 0.0845 | 0.1014 | 0.1194 | 0.1429 | 0.1121 | 0.1121 |
|  | Recall | 0.0693 | 0.0880 | 0.1113 | 0.1411 | 0.1024 | 0.1024 |
|  | Hits$^N$ | 0.1438 | 0.1683 | 0.1857 | 0.2148 | 0.1782 | 0.1782 |
|  | MRR | | 0.1366 | | | 0.1366 | 0.1366 |
|  | | | | | | 0.1575 | 0.1575 |
| Degree$^{\text{Test}}$ | Precision | 0.3052 | 0.4412 | 0.5704 | 0.7223 | 0.5098 | 0.5098 |
|  | F1 | 0.1919 | 0.3133 | 0.4639 | 0.6597 | 0.4072 | 0.4072 |
|  | NDCG | 0.0949 | 0.1220 | 0.1548 | 0.1975 | 0.1423 | 0.1423 |
|  | Recall | 0.0323 | 0.0562 | 0.0909 | 0.1314 | 0.0777 | 0.0777 |
|  | Hits$^N$ | 0.2745 | 0.3258 | 0.3378 | 0.3369 | 0.3188 | 0.3188 |
|  | MRR | | 0.2444 | | | 0.2444 | 0.2444 |
|  | | | | | | 0.2911 | 0.2911 |
| Subgraph Density | Precision | 0.1559 | 0.1412 | 0.1168 | 0.0858 | 0.1249 | 0.1249 |
|  | F1 | 0.1867 | 0.1699 | 0.1420 | 0.1035 | 0.1505 | 0.1505 |
|  | NDCG | 0.2006 | 0.2097 | 0.2176 | 0.2235 | 0.2129 | 0.2129 |
|  | Recall | 0.2218 | 0.2289 | 0.2411 | 0.2491 | 0.2352 | 0.2352 |
|  | Hits$^N$ | 0.1768 | 0.1799 | 0.1982 | 0.2097 | 0.1912 | 0.1912 |
|  | MRR | | 0.1759 | | | 0.1759 | 0.1759 |
|  | | | | | | 0.1829 | 0.1829 |

Table 8: The correlation between $TC^{Train}$/$TC^{Val}$/$TC^{Test}$/$Degree^{Train}$ and the GCN's LP performance on **Pubmed**. We note that the formal definitions of Absolute Avg. and Basic Avg. are provided in Section G.8 and they represent the average absolute and simple average correlation, respectively, across the range of @K for the given metric; these are also then calculated overall.

| | Metric | @5 | @10 | @20 | @50 | Absolute Avg. | Basic Avg. |
|---|---|---|---|---|---|---|---|
| $TC^{Train}$ | Precision | 0.1981 | 0.2358 | 0.2681 | 0.2924 | 0.2486 | 0.2486 |
| | F1 | 0.1690 | 0.2216 | 0.2652 | 0.2961 | 0.2380 | 0.2380 |
| | NDCG | 0.1195 | 0.1379 | 0.1600 | 0.1831 | 0.1501 | 0.1501 |
| | Recall | 0.0917 | 0.1142 | 0.1336 | 0.1397 | 0.1198 | 0.1198 |
| | $Hits^N$ | 0.1932 | 0.2267 | 0.2485 | 0.2513 | 0.2299 | 0.2299 |
| | MRR | | 0.1920 | | | 0.1920 | 0.1920 |
| | | | | | | 0.1973 | 0.1973 |
| $TC^{Val}$ | Precision | 0.1769 | 0.2180 | 0.2653 | 0.3134 | 0.2434 | 0.2434 |
| | F1 | 0.1253 | 0.1815 | 0.2462 | 0.3092 | 0.2156 | 0.2156 |
| | NDCG | 0.0780 | 0.0846 | 0.1046 | 0.1303 | 0.0994 | 0.0994 |
| | Recall | 0.0417 | 0.0503 | 0.0672 | 0.0804 | 0.0599 | 0.0599 |
| | $Hits^N$ | 0.1627 | 0.1882 | 0.2068 | 0.2077 | 0.1914 | 0.1914 |
| | MRR | | 0.1607 | | | 0.1607 | 0.1607 |
| | | | | | | 0.1619 | 0.1619 |
| $TC^{Test}$ | Precision | 0.3769 | 0.3989 | 0.4078 | 0.3909 | 0.3936 | 0.3936 |
| | F1 | 0.4011 | 0.4258 | 0.4329 | 0.4088 | 0.4172 | 0.4172 |
| | NDCG | 0.3902 | 0.4231 | 0.4547 | 0.4870 | 0.4388 | 0.4388 |
| | Recall | 0.3809 | 0.4080 | 0.4286 | 0.4335 | 0.4128 | 0.4128 |
| | $Hits^N$ | 0.3923 | 0.4247 | 0.4463 | 0.4436 | 0.4267 | 0.4267 |
| | MRR | | 0.4097 | | | 0.4097 | 0.4097 |
| | | | | | | 0.4178 | 0.4178 |
| $Degree^{Train}$ | Precision | 0.2433 | 0.3108 | 0.3761 | 0.4849 | 0.3538 | 0.3538 |
| | F1 | 0.1019 | 0.1970 | 0.2987 | 0.4456 | 0.2608 | 0.2608 |
| | NDCG | 0.0477 | 0.0366 | 0.0441 | 0.0715 | 0.0500 | 0.0500 |
| | Recall | -0.0402 | -0.0386 | -0.0385 | -0.0357 | 0.0383 | -0.0383 |
| | $Hits^N$ | 0.2080 | 0.2404 | 0.2504 | 0.2612 | 0.2400 | 0.2400 |
| | MRR | | 0.2051 | | | 0.2051 | 0.2051 |
| | | | | | | 0.1886 | 0.1733 |
| $Degree^{Val}$ | Precision | 0.1823 | 0.2290 | 0.2849 | 0.3681 | 0.2661 | 0.2661 |
| | F1 | 0.0676 | 0.1368 | 0.2220 | 0.3359 | 0.1906 | 0.1906 |
| | NDCG | 0.0293 | 0.0164 | 0.0221 | 0.0407 | 0.0271 | 0.0271 |
| | Recall | -0.0429 | -0.0466 | -0.0459 | -0.0476 | 0.0458 | -0.0458 |
| | $Hits^N$ | 0.1536 | 0.1749 | 0.1831 | 0.1872 | 0.1747 | 0.1747 |
| | MRR | | 0.1573 | | | 0.1573 | 0.1573 |
| | | | | | | 0.1408 | 0.1225 |
| $Degree^{Test}$ | Precision | 0.3073 | 0.3898 | 0.4719 | 0.6133 | 0.4456 | 0.4456 |
| | F1 | 0.1251 | 0.2423 | 0.3716 | 0.5624 | 0.3254 | 0.3254 |
| | NDCG | 0.0588 | 0.0406 | 0.0480 | 0.0821 | 0.0574 | 0.0574 |
| | Recall | -0.0537 | -0.0565 | -0.0605 | -0.0575 | 0.0571 | -0.0571 |
| | $Hits^N$ | 0.2615 | 0.2966 | 0.3030 | 0.3099 | 0.2928 | 0.2928 |
| | MRR | | 0.2556 | | | 0.2556 | 0.2556 |
| | | | | | | 0.2356 | 0.2128 |
| Subgraph Density | Precision | 0.1002 | 0.0746 | 0.0414 | -0.0146 | 0.0577 | 0.0504 |
| | F1 | 0.1732 | 0.1319 | 0.0792 | 0.0030 | 0.0968 | 0.0968 |
| | NDCG | 0.2146 | 0.2307 | 0.2357 | 0.2344 | 0.2289 | 0.2289 |
| | Recall | 0.2475 | 0.2547 | 0.2540 | 0.2428 | 0.2498 | 0.2498 |
| | $Hits^N$ | 0.1343 | 0.1330 | 0.1338 | 0.1288 | 0.1325 | 0.1325 |
| | MRR | | 0.1430 | | | 0.1430 | 0.1430 |
| | | | | | | 0.1531 | 0.1517 |

Table 9: The correlation between $TC^{Train}$/$TC^{Val}$/$TC^{Test}$/$Degree^{Train}$ and the GCN's LP performance on **Vole**. We note that the formal definitions of Absolute Avg. and Basic Avg. are provided in Section G.8 and they represent the average absolute and simple average correlation, respectively, across the range of @K for the given metric; these are also then calculated overall.

| | Metric | @5 | @10 | @20 | @50 | Absolute Avg. | Basic Avg. |
|---|---|---|---|---|---|---|---|
| $TC^{Train}$ | Precision | 0.2725 | 0.2710 | 0.2648 | 0.2287 | 0.2593 | 0.2593 |
| | F1 | 0.2985 | 0.2981 | 0.2869 | 0.2401 | 0.2809 | 0.2809 |
| | NDCG | 0.2714 | 0.3012 | 0.3300 | 0.3497 | 0.3131 | 0.3131 |
| | Recall | 0.2946 | 0.3267 | 0.3677 | 0.3917 | 0.3452 | 0.3452 |
| | $Hits^N$ | 0.3113 | 0.3307 | 0.3694 | 0.3988 | 0.3526 | 0.3526 |
| | MRR | | 0.2721 | | | 0.2721 | 0.2721 |
| | | | | | | 0.3102 | 0.3102 |
| $TC^{Val}$ | Precision | 0.1375 | 0.1717 | 0.1847 | 0.1721 | 0.1665 | 0.1665 |
| | F1 | 0.1233 | 0.1690 | 0.1871 | 0.1739 | 0.1633 | 0.1633 |
| | NDCG | 0.0931 | 0.1201 | 0.1403 | 0.1479 | 0.1254 | 0.1254 |
| | Recall | 0.0825 | 0.1251 | 0.1570 | 0.1548 | 0.1299 | 0.1299 |
| | $Hits^N$ | 0.1347 | 0.1558 | 0.1815 | 0.1814 | 0.1634 | 0.1634 |
| | MRR | | 0.1219 | | | 0.1219 | 0.1219 |
| | | | | | | 0.1497 | 0.1497 |
| $TC^{Test}$ | Precision | 0.5547 | 0.4822 | 0.3937 | 0.2527 | 0.4208 | 0.4208 |
| | F1 | 0.6498 | 0.5597 | 0.4449 | 0.2742 | 0.4822 | 0.4822 |
| | NDCG | 0.7395 | 0.7712 | 0.7954 | 0.8030 | 0.7773 | 0.7773 |
| | Recall | 0.7325 | 0.7384 | 0.7367 | 0.6812 | 0.7222 | 0.7222 |
| | $Hits^N$ | 0.6470 | 0.6529 | 0.6452 | 0.6016 | 0.6367 | 0.6367 |
| | MRR | | 0.6950 | | | 0.6950 | 0.6950 |
| | | | | | | 0.6078 | 0.6078 |
| $Degree^{Train}$ | Precision | 0.2103 | 0.2728 | 0.3620 | 0.4508 | 0.3240 | 0.3240 |
| | F1 | 0.1387 | 0.2253 | 0.3391 | 0.4508 | 0.2885 | 0.2885 |
| | NDCG | 0.0180 | 0.0352 | 0.0760 | 0.1222 | 0.0629 | 0.0629 |
| | Recall | 0.0238 | 0.0479 | 0.1111 | 0.1993 | 0.0955 | 0.0955 |
| | $Hits^N$ | 0.1688 | 0.1977 | 0.2551 | 0.2989 | 0.2301 | 0.2301 |
| | MRR | | 0.0512 | | | 0.0512 | 0.0512 |
| | | | | | | 0.2002 | 0.2002 |
| $Degree^{Val}$ | Precision | 0.0312 | 0.0747 | 0.1182 | 0.1758 | 0.1000 | 0.1000 |
| | F1 | -0.0135 | 0.0414 | 0.0989 | 0.1685 | 0.0806 | 0.0738 |
| | NDCG | -0.0527 | -0.0455 | -0.0336 | -0.0153 | 0.0368 | -0.0368 |
| | Recall | -0.0670 | -0.0487 | -0.0309 | 0.0059 | 0.0381 | -0.0352 |
| | $Hits^N$ | 0.0077 | 0.0180 | 0.0368 | 0.0599 | 0.0306 | 0.0306 |
| | MRR | | -0.0215 | | | -0.0215 | -0.0215 |
| | | | | | | 0.0572 | 0.0265 |
| $Degree^{Test}$ | Precision | 0.3731 | 0.5111 | 0.6562 | 0.8126 | 0.5883 | 0.5883 |
| | F1 | 0.2040 | 0.3944 | 0.5926 | 0.7916 | 0.4957 | 0.4957 |
| | NDCG | 0.0004 | 0.0257 | 0.0722 | 0.1330 | 0.0578 | 0.0578 |
| | Recall | -0.0942 | -0.0697 | -0.0301 | 0.0419 | 0.0590 | -0.0380 |
| | $Hits^N$ | 0.2320 | 0.2604 | 0.2731 | 0.2529 | 0.2546 | 0.2546 |
| | MRR | | 0.1642 | | | 0.1642 | 0.1642 |
| | | | | | | 0.2911 | 0.2717 |
| Subgraph Density | Precision | 0.0744 | 0.0369 | -0.0119 | -0.0815 | 0.0512 | 0.0045 |
| | F1 | 0.1372 | 0.0860 | 0.0187 | -0.0689 | 0.0777 | 0.0433 |
| | NDCG | 0.2205 | 0.2341 | 0.2398 | 0.2398 | 0.2336 | 0.2336 |
| | Recall | 0.2178 | 0.2366 | 0.2495 | 0.2545 | 0.2396 | 0.2396 |
| | $Hits^N$ | 0.1206 | 0.1493 | 0.1688 | 0.2138 | 0.1631 | 0.1631 |
| | MRR | | 0.2026 | | | 0.2026 | 0.2026 |
| | | | | | | 0.1530 | 0.1368 |

Table 10: The correlation between $TC^{Train}/TC^{Val}/TC^{Test}/Degree^{Train}$ and the GCN's LP performance on **Reptile**. We note that the formal definitions of Absolute Avg. and Basic Avg. are provided in Section G.8 and they represent the average absolute and simple average correlation, respectively, across the range of @K for the given metric; these are also then calculated overall.

| | Metric | @5 | @10 | @20 | @50 | Absolute Avg. | Basic Avg. |
|---|---|---|---|---|---|---|---|
| $TC^{Train}$ | Precision | 0.5189 | 0.5084 | 0.4977 | 0.5009 | 0.5065 | 0.5065 |
| | F1 | 0.5420 | 0.5307 | 0.5146 | 0.5090 | 0.5241 | 0.5241 |
| | NDCG | 0.5298 | 0.5502 | 0.5636 | 0.5741 | 0.5544 | 0.5544 |
| | Recall | 0.5097 | 0.5176 | 0.5343 | 0.5475 | 0.5273 | 0.5273 |
| | $Hits^N$ | 0.5208 | 0.5278 | 0.5407 | 0.5502 | 0.5349 | 0.5349 |
| | MRR | | 0.5300 | | | 0.5300 | 0.5300 |
| | | | | | | 0.5294 | 0.5294 |
| $TC^{Val}$ | Precision | 0.3994 | 0.4316 | 0.4550 | 0.4647 | 0.4377 | 0.4377 |
| | F1 | 0.3753 | 0.4250 | 0.4573 | 0.4670 | 0.4312 | 0.4312 |
| | NDCG | 0.3183 | 0.3535 | 0.3790 | 0.3909 | 0.3604 | 0.3604 |
| | Recall | 0.2670 | 0.3085 | 0.3525 | 0.3744 | 0.3256 | 0.3256 |
| | $Hits^N$ | 0.3213 | 0.3483 | 0.3675 | 0.3840 | 0.3553 | 0.3553 |
| | MRR | | 0.3666 | | | 0.3666 | 0.3666 |
| | | | | | | 0.3820 | 0.3820 |
| $TC^{Test}$ | Precision | 0.7083 | 0.7000 | 0.6739 | 0.6506 | 0.6832 | 0.6832 |
| | F1 | 0.7898 | 0.7629 | 0.7138 | 0.6678 | 0.7336 | 0.7336 |
| | NDCG | 0.8475 | 0.8897 | 0.9029 | 0.9072 | 0.8868 | 0.8868 |
| | Recall | 0.8573 | 0.8858 | 0.8931 | 0.8759 | 0.8780 | 0.8780 |
| | $Hits^N$ | 0.8276 | 0.8566 | 0.8604 | 0.8495 | 0.8485 | 0.8485 |
| | MRR | | 0.8163 | | | 0.8163 | 0.8163 |
| | | | | | | 0.8060 | 0.8060 |
| $Degree^{Train}$ | Precision | 0.4998 | 0.5294 | 0.5664 | 0.5947 | 0.5476 | 0.5476 |
| | F1 | 0.5082 | 0.5411 | 0.5788 | 0.6017 | 0.5575 | 0.5575 |
| | NDCG | 0.4247 | 0.4572 | 0.4914 | 0.5120 | 0.4713 | 0.4713 |
| | Recall | 0.4338 | 0.4598 | 0.5201 | 0.5615 | 0.4938 | 0.4938 |
| | $Hits^N$ | 0.4998 | 0.5073 | 0.5391 | 0.5664 | 0.5282 | 0.5282 |
| | MRR | | 0.4369 | | | 0.4369 | 0.4369 |
| | | | | | | 0.5197 | 0.5197 |
| $Degree^{Val}$ | Precision | 0.3185 | 0.3577 | 0.3797 | 0.3924 | 0.3621 | 0.3621 |
| | F1 | 0.3022 | 0.3546 | 0.3840 | 0.3956 | 0.3591 | 0.3591 |
| | NDCG | 0.2285 | 0.2617 | 0.2858 | 0.2985 | 0.2686 | 0.2686 |
| | Recall | 0.1997 | 0.2384 | 0.2830 | 0.3093 | 0.2576 | 0.2576 |
| | $Hits^N$ | 0.2729 | 0.2938 | 0.3165 | 0.3339 | 0.3043 | 0.3043 |
| | MRR | | 0.2677 | | | 0.2677 | 0.2677 |
| | | | | | | 0.3103 | 0.3103 |
| $Degree^{Test}$ | Precision | 0.6833 | 0.7492 | 0.7935 | 0.8118 | 0.7595 | 0.7595 |
| | F1 | 0.5477 | 0.6726 | 0.7556 | 0.7968 | 0.6932 | 0.6932 |
| | NDCG | 0.3062 | 0.3404 | 0.3676 | 0.3790 | 0.3483 | 0.3483 |
| | Recall | 0.1840 | 0.2103 | 0.2429 | 0.2532 | 0.2226 | 0.2226 |
| | $Hits^N$ | 0.3940 | 0.3555 | 0.3381 | 0.3283 | 0.3540 | 0.3540 |
| | MRR | | 0.4468 | | | 0.4468 | 0.4468 |
| | | | | | | 0.4755 | 0.4755 |
| Subgraph Density | Precision | 0.2482 | 0.2491 | 0.2211 | 0.2022 | 0.2302 | 0.2302 |
| | F1 | 0.2943 | 0.2849 | 0.2420 | 0.2108 | 0.2580 | 0.2580 |
| | NDCG | 0.3560 | 0.3819 | 0.3792 | 0.3765 | 0.3734 | 0.3734 |
| | Recall | 0.3588 | 0.3928 | 0.3777 | 0.3607 | 0.3725 | 0.3725 |
| | $Hits^N$ | 0.3440 | 0.3891 | 0.3837 | 0.3745 | 0.3728 | 0.3728 |
| | MRR | | 0.3510 | | | 0.3510 | 0.3510 |
| | | | | | | 0.3214 | 0.3214 |

## H    EDGE REWEIGHTING ALGORITHM

Here we present our edge reweigting algorithm to enhance the link prediction performance by modifying the graph adjacency matrix in message-passing. We normalize the adjacency matrix to get $\widetilde{\mathbf{A}}$ and $\widehat{\mathbf{A}}$ as defined in the algorithm below.

---

**Algorithm 1:** Edge Reweighting to Boost LP performance

---

**Input:** The input training graph $(\mathbf{A}, \mathbf{X}, \mathcal{E}^{\text{Train}}, \mathbf{D})$, graph encoder $f_{\mathbf{\Theta}_f}$, link predictor $g_{\mathbf{\Theta}_g}$, update interval $\Delta$, training epochs $T$, warm up epochs $T^{\text{warm}}$ and weights $\gamma$ for combining the original adjacency matrix and the updated adjacency matrix. The validation adjacency/degree matrix $\mathbf{A}^{\text{Val}}/\mathbf{D}^{\text{Val}}$ that only includes edges in the validation set.

1  Compute the normalized adjacency matrices $\widehat{\mathbf{A}} = \mathbf{D}^{-0.5}\mathbf{A}\mathbf{D}^{-0.5}, \widetilde{\mathbf{A}} = \mathbf{D}^{-1}\mathbf{A}, \widetilde{\mathbf{A}}^{\text{Val}} = \mathbf{D}^{\text{Val}-1}\mathbf{A}^{\text{Val}}$

2  $\widetilde{\mathbf{A}}^0 = \widehat{\mathbf{A}}$

3  **for** $\tau = 1, \ldots, T$ **do**

4  $\quad$ **if** $\tau\%\Delta \neq 0 \; or \; \tau \leq T^{\text{warm}}$ **then**

5  $\quad\quad$ $\widetilde{\mathbf{A}}^\tau = \widetilde{\mathbf{A}}^{\tau-1}$

$\quad$ /* Message-passing and LP to update model parameters $\qquad\qquad$ */

6  $\quad$ **for** *mini-batch of edges* $\mathcal{E}^b \subseteq \mathcal{E}^{\text{Train}}$ **do**

7  $\quad\quad$ Sample negative edges $\mathcal{E}^{b,-}$, s.t., $|\mathcal{E}^{b,-}| = |\mathcal{E}^b|$

8  $\quad\quad$ Compute node embeddings $\mathbf{H}^\tau = f_{\mathbf{\Theta}_f^{\tau-1}}(\widetilde{\mathbf{A}}^\tau, \mathbf{X})$

9  $\quad\quad$ Compute link prediction scores $\mathbf{E}_{ij}^\tau = g_{\mathbf{\Theta}_g^{\tau-1}}(\mathbf{H}_i^\tau, \mathbf{H}_j^\tau), \forall (i,j) \in \mathcal{E}^b \cup \mathcal{E}^{\text{Train}}$

10  $\quad\quad$ $\mathcal{L}^{b,\tau} = -\frac{1}{|\mathcal{E}^b|}(\sum_{e_{ij}\in\mathcal{E}^b}\log\mathbf{E}_{ij}^\tau + \sum_{e_{mn}\in\mathcal{E}^{b,-}}\log(1-\mathbf{E}_{mn}^\tau))$

11  $\quad\quad$ Update $\mathbf{\Theta}_g^\tau \leftarrow \mathbf{\Theta}_g^{\tau-1} - \nabla_{\mathbf{\Theta}_g^{\tau-1}}\mathcal{L}^{b,\tau}, \;\; \mathbf{\Theta}_f^\tau \leftarrow \mathbf{\Theta}_f^{\tau-1} - \nabla_{\mathbf{\Theta}_f^{\tau-1}}\mathcal{L}^{b,\tau-1}$

$\quad$ /* Update adjacency matrix to enhance weighted TC $\qquad\qquad\qquad$ */

12  $\quad$ **if** $\tau\%\Delta == 0 \; and \; \tau > T^{\text{warm}}$ **then**

13  $\quad\quad$ Compute node embeddings $\mathbf{H}^\tau = f_{\mathbf{\Theta}_f^{\tau-1}}(\widetilde{\mathbf{A}}^{\tau-1}, \mathbf{X})$;

14  $\quad\quad$ **if** *Using training neighbors to reweigh* **then**

15  $\quad\quad\quad$ Average pooling the neighborhood embeddings $\mathbf{N}^\tau = \widetilde{\mathbf{A}}\mathbf{H}^\tau$

16  $\quad\quad$ **if** *Using validation neighbors to reweigh* **then**

17  $\quad\quad\quad$ Average pooling the neighborhood embeddings $\mathbf{N}^\tau = \widetilde{\mathbf{A}}^{\text{Val}}\mathbf{H}^\tau$

18  $\quad\quad$ Compute the link prediction scores $\mathbf{S}_{ij}^\tau = \frac{\exp(g_{\mathbf{\Theta}_g^\tau}(\mathbf{N}_i^\tau, \mathbf{H}_j^\tau))}{\sum_{j=1}^n \exp(g_{\mathbf{\Theta}_g^\tau}(\mathbf{N}_i^\tau, \mathbf{H}_j^\tau))}$

19  $\quad\quad$ Update the adjacency matrix $\widetilde{\mathbf{A}}^\tau \leftarrow \widehat{\mathbf{A}} + \gamma\mathbf{S}^\tau$

20  **Return:** $\widetilde{\mathbf{A}}^\tau, f_{\mathbf{\Theta}_f^\tau}, g_{\mathbf{\Theta}_g^\tau}$

---

## I    REWEIGH EDGES FOR BASELINES WITHOUT MESSAGE-PASSING

As discussed in Section 4, we enhance node $\text{TC}^{\text{Train}}$ by reweighing the edges in message-passing. However, for some state-of-the-art baselines Chamberlain et al. (2022) that directly employ the neural transformation rather than message-passing to obtain node embeddings, we reweigh edges in computing the binary cross entropy loss in the training stage as follows:

$$\mathcal{L} = -\frac{1}{|\mathcal{E}^b|}\sum_{e_{ij}\in\mathcal{E}^b}(w_{ij}\sum_{e_{ij}\in\mathcal{E}^b}\log\mathbf{E}_{ij}^\tau + w_{mn}\sum_{e_{mn}\in\mathcal{E}^{b,-}}\log(1-\mathbf{E}_{mn}^\tau)), \qquad (30)$$

where $w_{ij} = \sigma(\phi(\mathbf{N}_i, \mathbf{N}_j))$ quantifies the edge weight between $v_i$ and $v_j$ with $\sigma$ being the Sigmoid function and $\phi$ being the cosine similarity. $\mathbf{N}_i$ is the node embedding of $v_i$ obtained in Eq. (2).

Table 11: Comparing the efficiency (s) between X and our proposed $X_{rw}$.

| Baseline | Cora | Citeseer | Pubmed | Collab | Reptile | Vole |
|---|---|---|---|---|---|---|
| GCN | 19,5 | 15.5 | 158.4 | 2906 | 18.3 | 53.8 |
| $GCN_{rw}$ | 21.1 | 17.2 | 158.5 | 2915 | 17.0 | 53.0 |
| SAGE | 22.3 | 17.0 | 189.8 | 2970 | 20.9 | 61.2 |
| $SAGE_{rw}$ | 23.8 | 20.5 | 192.4 | 2982 | 21.7 | 61.9 |
| BUDDY | 3.41 | 4.51 | 15.51 | 906.18 | 2.50 | 4.99 |
| $BUDDY_{rw}$ | 3.98 | 4.92 | 14.56 | 907.52 | 2.62 | 5.06 |

## J COMPARING THE EFFICIENCY BETWEEN BASELINE AND THEIR AUGMENTED VERSION BY TC

Here we compare the running time (s) of each baseline and their corresponding augmented version by uniformly testing them on the same machine in Table 11. We can see that equipping our proposed reweighting strategy could enhance the performance but only lead to marginal computational overhead. This is because firstly, we only change the weight of existing edges and hence the number of edge weights to be calculated is linear to the network size. Secondly, we leverage the pre-computed node embeddings to compute the edge weights. Thirdly, we only periodically update the edge weights.

## K REWEIGHTING TRAINING NEIGHBORS BASED ON THEIR CONNECTIONS TO TRAINING NEIGHBORS OR VALIDATION NEIGHBORS

As discussed in **Obs. 3**, due to the topological distribution shift, the newly joined neighbors of one node become less and less connective to the previous neighbors of that node. Therefore, the training neighbors of one node share fewer connections with the testing neighbors of that node than the validation neighbors. This motivates us to further improve our reweighting strategy based on validation neighbors rather than training neighbors. **The intuition is that when performing message-passing to aggregate training neighbors' information for each node, we want to incorporate those training neighbors with more connections to that node's validation neighbors instead of those training neighbors with more connections to that node's training neighbors.** Technically, we include additional steps 14-17 to consider two scenarios in Algorithm H: **(1)** reweighting based on the connections of training neighbors to training neighbors and **(2)** reweighting based on the connections of training neighbors to validation neighbors. We experiment on Collab to compare the performance of these two scenarios in Table 12. We can see the performance of reweighting based on validation neighbors is higher than reweighting based on training neighbors. This demonstrates that the validation neighbors are more connected to the testing neighbors, justifying the existence of the topological distribution shift.

Table 12: Comparing the link prediction performance on Collab between reweighting based on training neighbors and reweighting based on validation neighbors

| Performance | GCN | | | SAGE | | |
|---|---|---|---|---|---|---|
| | No | Train | Val | No | Train | Val |
| Hits@5 | 18.94±1.20 | 19.48±0.75 | **22.36±0.32** | 11.25±1.24 | 20.52±2.35 | **24.34±0.07** |
| Hits@10 | 31.24±3.44 | 32.69±1.00 | **35.15±2.42** | 26.41±1.88 | 31.23±3.52 | **37.15±2.44** |
| Hits@50 | 50.12±0.22 | 52.77±1.00 | **53.24±0.22** | 49.68±0.25 | 51.87±0.10 | **52.69±0.26** |
| Hits@100 | 54.44±0.49 | 56.89±0.17 | **57.28±0.10** | 54.69±0.18 | 56.59±0.19 | **57.27±0.25** |

## L    EXPLAINING WHY THE CURVE OF LINK PREDICTION PERFORMANCE HAS SEVERAL FAST DOWN IN FIGURE 7(A)

Here we delve deep into the reason why we encounter several fast-down performances in Figure 7(a). We ascribe it to the weight clip [3]. We hypothesize that the loss landscape has several local minimums and hence by weight clipping with higher upper bound constraints, our learning step would be also larger so that the model could jump out of its original local optimum and keep finding some other better local optimum, which corresponds to the fast downtrend (first jump away from one local minimum and then find another better local minimum). We further empirically verify our hypothesis by visualizing the performance curve for each training process with different clipping weights in Figure 23. We can clearly see that as the clipping threshold becomes lower (upper bound decreases), we observe less fast downtrend decreases.

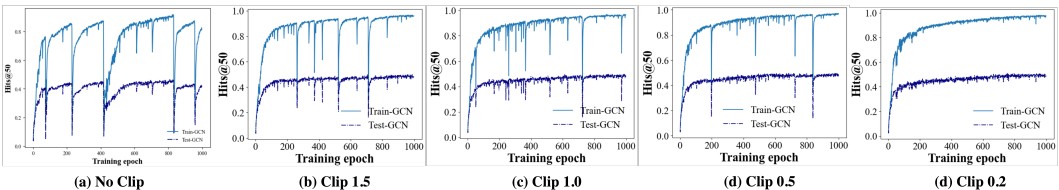

(a) No Clip          (b) Clip 1.5          (c) Clip 1.0          (d) Clip 0.5          (d) Clip 0.2

Figure 23: From left to right, we constrain GCN-based link predictor with fewer upper bounds by clipping using a lower threshold. We can see the number of performance fast downtrend decreases. We hypothesize that the loss landscape has several local minimums and hence by weight clipping with lower upper bounds, our learning step would be also smaller so that the model could not jump out of its origin local optimum and hence we end up with fewer fast downtrends.

---

[3]Following the publically available implementation on GCN/SAGE on Collab link prediction, we employ the weight clip every time after parameter update

