# OpenReview forum: "A Topological Perspective on Demystifying GNN-Based Link Prediction Performance"
_ICLR.cc/2024/Conference — ICLR 2024 poster_

### Official Review · Reviewer_QpHH · 2023-10-31

**Soundness:** 3 good
**Presentation:** 3 good
**Contribution:** 3 good
**Rating:** 6
**Confidence:** 2

**Summary:**

This paper proposes a novel metric called Topological Concentration (TC) to demythify the varying performance of Graph Neural Networks (GNNs) in link prediction across different nodes. The authors demonstrate that TC has a higher correlation with link prediction performance than other node-level topological metrics like degree and subgraph density. They also uncover a novel Topological Distribution Shift (TDS) issue and propose an Approximated Topological Concentration (ATC) to address the computational complexity of TC. The paper concludes by exploring the potential of boosting link prediction performance via enhancing TC.

**Strengths:**

- The paper introduces a new metric, Topological Concentration (TC), which provides a better characterization of node link prediction performance in GNNs than traditional metrics like degree and subgraph density.
- The authors uncover a novel Topological Distribution Shift (TDS) issue and demonstrate its negative impact on link prediction performance at the node and graph levels.
- The paper proposes an Approximated Topological Concentration (ATC) to address the computational complexity of TC, while maintaining high correlations to link prediction performance.
- The authors explore the potential of boosting link prediction performance by enhancing TC through edge reweighting in message-passing and discuss its efficacy and limitations.

**Weaknesses:**

- The paper could benefit from a more comprehensive evaluation of the proposed methods on a wider range of datasets and benchmark tasks. This would help to establish the generalizability and robustness of the proposed techniques across different domains and problem settings.

- The theoretical analysis of the relationship between TC and link prediction performance could be further strengthened with additional mathematical proofs or rigorous analysis. Providing a more solid theoretical foundation would increase the credibility and impact of the proposed methods.

- The paper could discuss the limitations and potential biases of the proposed evaluation metrics, such as TC and ATC. Addressing these concerns would help to ensure that the results are reliable and that the methods are not overly sensitive to specific aspects of the data.

- The paper could provide a more in-depth analysis of the cold-start problem and its relationship with the proposed metrics, as well as discuss potential strategies for addressing this issue. This would help to better understand the practical implications of the proposed methods and their potential to improve link prediction performance in real-world scenarios.

**Questions:**

- Could you please provide more details on the implementation of the Approximated Topological Concentration (ATC) and its theoretical justification for approximating TC? This would help to better understand the rationale behind the proposed method and its potential advantages over other approaches.

- How do the proposed methods compare to other state-of-the-art link prediction techniques in terms of performance and computational efficiency?

- What does the "cold-start" nodes mean? Please clarify the definition and implications of cold-start nodes in the context of link prediction. This would help to better understand the relevance and importance of addressing the cold-start problem in the proposed methods.

---

> ### Author Response · Authors · 2023-11-16
> **Response to Reviewer QpHH, Question 1, how to implement ATC.**
>
> **=====Question 1=====**
>
> Thank you for the question. As listed in Eq. (2), we calculate ATC in the following steps:
> - Firstly, we initialize node embeddings by sampling from a d-dimensional multivariate normal distribution with zero mean
>
> - Secondly, we perform message-passing K times, summarize the adjacency matrices from the power of 1 to K, and obtain the propagated embeddings for each node.
>
> - Thirdly, we compute each node's embedding similarity to each neighbor and average them to calculate the ATC.
>
> **Our paper has justified the theoretical connection between ATC and TC by proving the linear relationship between TC and the mean value of ATC in Theorem 1 in Section 3.4**
>
> **=====================**

---

> ### Author Response · Authors · 2023-11-22
> **Response to Reviewer QpHH, Question 3, the definition about the cold-start nodes in our paper**
>
> **=====Question 3=====**
>
> We really appreciate the reviewer's observation about the definition of cold-start nodes in our paper. Following the suggestion, we conduct a systematic literature review about the definition of cold-start nodes in the context of link prediction. We first present the summary of the definition of cold-start nodes in existing literature as follows:
>
> | Literature | Definition  | Solution |
> |----------|----------------|-------------------|
> |  [1]    |  Little or no topological information (isolated user)    |   Augment the cold-start nodes by auxiliary information (user profile/rich text information)      |
> |  [2]   |   New node, new user   |   Augment the link prediction for new users using their social features      |
> |  [3]   |   Users with no interactions | Augment the link prediction for new users using their interactions in another platform based on the bidirectional relation.
> |  [4]   |   Vertices without edges | Consider the node community information in metric learning|
> |  [5]   |   No network information is observed | Leverage user group membership information to bootstrap probabilistic graph and then refine the graph probability by considering the transitivity of the contract relationship.|
> |  [6]   |   Pseudo cold start link prediction (predicting the structure of a social network when only a small subgraph is observed) | Augment the link prediction by features from multiple auxiliary networks with feature selection and regularization|
> |  [7]/[8]   |   Low activity users/Disadvantaged users | Reranking to reduce the performance gap between low-activity users and high-activity users|
> |  [9]   |   Fresh users with limited interactions | Meta-optimization by learning to learn the user preference based on his/her only a few past interactions|
> |  [10]  |  When no time is given, cold-start items are defined based on items with fewer interactions | Learning the local parameters for each user in meta-optimization|
> |  [11]  |  Users with sparse interactions | Training GNNs to adapt to cold-start scenarios by mimicking the cold-start scenario for warm users |
> | [12]  | Users with sparse interactions | Use knowledge graph embedding task to assist with recommendation task |
>
>
> From the above summary, we can see that:
>
>  **(1) conventional link prediction papers work on cold-start problems at the link level and define the cold-start link as the newly joined one with little or no topological information available.**
>
> **(2) some node-level works such as recommender systems define the cold-start nodes/users as the newly joined ones with sparse interactions.**
>
> [1] Wang, Zhiqiang, et al. "An approach to cold-start link prediction: Establishing connections between non-topological and topological information." IEEE Transactions on Knowledge and Data Engineering 28.11 (2016): 2857-2870.
>
> [2] Han, Xiao, et al. "Link prediction for new users in social networks." 2015 IEEE International Conference on Communications (ICC). IEEE, 2015.
>
> [3] Yan, Ming, et al. "Friend transfer: Cold-start friend recommendation with cross-platform transfer learning of social knowledge." 2013 IEEE International Conference on Multimedia and Expo (ICME). IEEE, 2013.
>
> [4] Xu, Linchuan, et al. "On learning mixed community-specific similarity metrics for cold-start link prediction." Proceedings of the 26th International Conference on World Wide Web Companion. 2017.
>
> [5] Leroy, Vincent, B. Barla Cambazoglu, and Francesco Bonchi. "Cold start link prediction." Proceedings of the 16th ACM SIGKDD international conference on Knowledge discovery and data mining. 2010.
>
> [6] Ge, Liang, and Aidong Zhang. "Pseudo cold start link prediction with multiple sources in social networks." Proceedings of the 2012 SIAM International Conference on Data Mining. Society for Industrial and Applied Mathematics, 2012.
>
> [7] Li, Yunqi, et al. "User-oriented fairness in recommendation." Proceedings of the Web Conference 2021. 2021.
>
> [8] Rahmani, Hossein A., et al. "Experiments on generalizability of user-oriented fairness in recommender systems." Proceedings of the
> 45th International ACM SIGIR Conference on Research and Development in Information Retrieval. 2022.
>
> [9] Wei, Tianxin, and Jingrui He. "Comprehensive fair meta-learned recommender system." Proceedings of the 28th ACM SIGKDD Conference on Knowledge Discovery and Data Mining. 2022.
>
> [10] Dong, Manqing, et al. "Mamo: Memory-augmented meta-optimization for cold-start recommendation." Proceedings of the 26th ACM SIGKDD international conference on knowledge discovery & data mining. 2020.
>
> [11] Hao, Bowen, et al. "Pre-training graph neural networks for cold-start users and items representation." Proceedings of the 14th ACM International Conference on Web Search and Data Mining. 2021.
>
> [12] Wang, Hongwei, et al. "Multi-task feature learning for knowledge graph enhanced recommendation." The world wide web conference. 2019.
>
> **=====Question 3=====**

---

> ### Author Response · Authors · 2023-11-22
> **Response to Reviewer QpHH, Question 3, the definition about the cold-start nodes in our paper**
>
> **=====Question 3=====**
>
> From the above summary, we can see that:
>
> (1) conventional link prediction papers work on cold-start problems at the link level and define the cold-start link as the newly joined one with little or no topological information available.
>
> (2) some node-level works such as recommender systems define the cold-start nodes/users as the newly joined ones with sparse interactions.
>
>
> Based on the above literature review, we answer the question 3 as follows:
>
> **Following the above second paradigm, we study the cold-start link prediction at the node level since our paper targets demystifying the varying link prediction performance**. Therefore, we follow some conventional literature [1-3] in that paradigm and generally deem the nodes with few degrees are cold-start ones. Concretely, in Figure 4(b)-(e), we change the degree threshold from 1 to 10, divide nodes into two groups at each degree threshold, and further visualize the average performance for each group. We can see that nodes in the low-degree groups generally have higher performance than nodes in the high-degree groups.
>
> The above observation has two promising insights compared with conventional literature:
>
> **(1) Many existing recommendation-based papers define cold-start users/nodes as the ones with few/little interactions/topological signals. However, our paper empirically justifies such a group of nodes even exhibiting higher performance in LP**
>
> We justify the above observation by relating to real-world scenarios where users with high degrees usually tend to possess diverse interests (nodes with higher degrees may tend to belong to diverse communities) and therefore, using the equal capacity of embedding cannot equally characterize all of their interests [4].
>
> **(2) Many existing node classification papers [5-7] find nodes with low degrees have lower performance. However, our work sheds new insights on the link prediction where nodes with lower degrees actually possess higher performance.**
>
> We justify the above observation by relating to the inherent difference between the mechanism of node classification and the mechanism of link prediction. For node classification, high-degree nodes are more likely to obtain the supervised signals from labeled nodes in the same class [8]. For link prediction, the ground-truth class for each node is actually its testing neighbors and hence there is no implicit biasing assumption that high-degree nodes would obtain more supervised signals.
>
>
> We include the above systematical literature review with a discussion about our observation on the cold-start problem in the Appendix B.
>
> **Note that the above analysis makes another significant contribution by summarizing the cold-start definition and sheds new insights on future works addressing cold-start/sparsity issues in link prediction.**
>
>
> [1] Dong, Manqing, et al. "Mamo: Memory-augmented meta-optimization for cold-start recommendation." Proceedings of the 26th ACM SIGKDD international conference on knowledge discovery & data mining. 2020.
>
> [2] Li, Yunqi, et al. "User-oriented fairness in recommendation." Proceedings of the Web Conference 2021. 2021.
>
> [3] Hao, Bowen, et al. "Pre-training graph neural networks for cold-start users and items representation." Proceedings of the 14th ACM International Conference on Web Search and Data Mining. 2021.
>
> [4] Zhaok, Xiangyu, et al. "Autoemb: Automated embedding dimensionality search in streaming recommendations." 2021 IEEE International Conference on Data Mining (ICDM). IEEE, 2021.
>
> [5] Tang, Xianfeng, et al. "Investigating and mitigating degree-related biases in graph convoltuional networks." Proceedings of the 29th ACM International Conference on Information & Knowledge Management. 2020.
>
> [6] Liu, Zemin, Trung-Kien Nguyen, and Yuan Fang. "Tail-gnn: Tail-node graph neural networks." Proceedings of the 27th ACM SIGKDD Conference on Knowledge Discovery & Data Mining. 2021.
>
> [7] Wang, Ruijia, et al. "Uncovering the Structural Fairness in Graph Contrastive Learning." Advances in Neural Information Processing Systems 35 (2022): 32465-32473.
>
> [8] Chen, Deli, et al. "Topology-imbalance learning for semi-supervised node classification." Advances in Neural Information Processing Systems 34 (2021): 29885-29897.
>
> **=====Question 3=====**

---

> ### Author Response · Authors · 2023-11-22
> **Response to Reviewer QpHH, Question 2, how is the performance and efficiency of the proposed method to other SOTA LP**
>
> We really thank the reviewer for asking about the effectiveness and efficiency of our proposed method. Our proposed strategy is a model-agnostic strategy that can be applied to existing LP models. Therefore, we compare the link prediction performance of many existing baselines (including state-of-the-art ones) with their augmented versions by our method. We visualize their performance as follows:
>
> | Baseline | Cora Hits@100  | Citeseer Hits@100 | Pubmed Hits@100 | Collab Hits@50 | Citation2 MRR | Reptile Hits@100 | Vole Hits@100 |
> |----------|----------------|-------------------|-----------------|----------------|---------------|------------------|---------------|
> | GCN      | 70.63±0.67     | 65.96±2.12        | 69.35±1.02      | 49.52±0.52     | 84.42±0.05    | 65.52±2.73       | 73.84±0.98    |
> | GCN$_{rw}$    | 75.98±1.28     | 74.40±1.13        | 68.87±0.99      | 52.85±0.14     | 83.54±0.30    | 70.79±2.00       | 74.50±0.84    |
> | SAGE     | 74.27±2.08     | 61.57±3.28        | 66.25±1.08      | 50.01±0.50     | 80.44±0.10    | 72.59±3.19       | 80.55±1.59    |
> | SAGE$_{rw}$   | 74.62±2.30     | 69.89±1.66        | 66.77±0.69      | 52.96±0.37     | 80.61±0.10    | 74.35±3.20       | 81.27±0.96    |
> | NCN      | 87.73±1.41     | 90.93±0.83        | 76.20±1.55      | 54.43±0.17     | 88.64±0.14    | 68.37±3.57       | 66.10±1.13    |
> | NCN$_{rw}$    | 87.95±1.30     | 91.86±0.82        | 76.51±1.41      | 54.16±0.24     | OOM             | 73.81±4.71       | 67.32±0.81    |
> | BUDDY    | 84.82±1.93     | 91.44±1.14        | 74.46±3.32      | 55.73±0.16     | More than two days    | 74.69±7.55                | 83.37±1.05
> | BUDDY$_{rw}$    | 85.06±1.81     | 91.86±1.00        | 75.74±2.03      | 56.17±0.42     | More than two days    | 77.20±4.10                | 84.64±1.29             |
>
> We can see the performance after equipping our reweighting strategy increases for most cases.
>
>
>
> Moreover, we compare the efficiency (in running time (s)) of the baselines before/after equipping our strategies as follows:
>
>
> | Baseline | Cora    | Citeseer | Pubmed | Collab | Reptile | Vole   |
> |----------|---------|----------|--------|--------|---------|--------|
> | GCN      | 19.5    | 15.5     | 158.4  | 2906   | 18.3    | 53.8   |
> | GCN$_{rw}$    | 21.1    | 17.2     | 158.5  | 2915   | 17.0    | 53.0   |
> | SAGE     | 22.3    | 17.0     | 189.8  | 2970   | 20.9    | 61.2   |
> | SAGE$_{rw}$  | 23.8    | 20.5     | 192.4  | 2982   | 21.7    | 61.9   |
> | BUDDY    | 3.41    | 4.51     | 15.51  | 906.18 | 2.50    | 4.99   |
> | BUDDY$_{rw}$  | 3.98    | 4.92     | 14.56  | 907.52 | 2.62    | 5.06   |
>
> We can find that equipping our strategy would only marginally increase the running time. This is because we only periodically update the edge weight and when calculating the weight using the node embeddings, we directly use the cached one obtained from the message-passing during the training stage.
>
> **=====================**

---

> > ### Comment · Reviewer_QpHH · 2023-11-23
> >
> > I appreciate the authors' responses and concur with the concerns raised by Reviewer 1hhc. The organization of the paper is somewhat confusing, particularly in the differentiation between 'cold-start nodes' and 'low-degree nodes'. The term 'low-degree nodes' is only mentioned twice in the paper, yet it is used during the discussion to explain 'cold-start nodes'.
> >
> > Despite this concern not being fully addressed, I am willing to increase my score. However, I strongly recommend that the authors revise the sections pertaining to the experiments and discussion of 'cold-start nodes', and restructure the paper for better clarity and understanding.

---

> > > ### Author Response · Authors · 2023-11-23
> > > **Response to Reviewer QpHH**
> > >
> > > We really thank the reviewer QpHH for the prompt response and appreciate the sharp observation on the low-degree and cold-start. We will update our discussion in the revised paper about the cold-start part and differentiate it from the low-degree one.

---

> ### Author Response · Authors · 2023-11-23
> **A kind reminder**
>
> Dear reviewer QpHH, as today is the final day of our discussion, we appreciate the opportunity to engage with you. If you have any remaining questions or concerns, please don't hesitate to share them with us; we will be happy to respond. Thank you.

---

> ### Author Response · Authors · 2023-11-23
> **Response to Reviewer QpHH, updating the cold-start content in the final paper**
>
> We have uploaded the final revised paper based on suggestions from all other reviewers and have reworded the content involving the cold-start part.
>
> Basically, we clarify that we are mainly working on degree-related issues in link prediction, not strictly on cold-start problems. We really appreciate reviewer's sharp observation on this.

---

### Official Review · Reviewer_f1PG · 2023-10-31

**Soundness:** 2 fair
**Presentation:** 3 good
**Contribution:** 2 fair
**Rating:** 6
**Confidence:** 4

**Summary:**

This paper proposes a metric "Topological Concentration" (TC) as a new measure for Link Prediction (LP) in GNNs.
TP is basically the intersection of the subgraphs of extremal nodes $i$ ans $j$ of an edge $(i,j)$. A concept very similar
to the common-neighbors heuristic. The authors claim "With TC, newly joined neighbors of a node tend to become less
interactive with that node’s existing neighbors, compromising the generalizability of node embeddings for LP at the testing time".

There is empirical evidence that GNNs perform better on high-degree nodes than on low-degree nodes. However, the authors
observe that this is not the case in LP. TC seems to find better correlations with LP performance (again, as common neighbors).
TC also inspires a new message-passing strategy considering well-connected neighbors.

The authors highlight the good properties of their measure and propose how to compute it efficiently. ATC (approximated topological
concentration) relies on powers-hops of the transition matrix. This is $O(Kd(|E| + |V|))$, where $K$ is the maximum hop.

The experiments show that TC boosts the performance of LP in several basic baselines. No comparisons with subgraph-based baselines.

**Strengths:**

* Formalization of an intuition (common neighbors and subgraph-based methods for LP).
* Nice empirical study on the properties of TC.
* ATC is efficient.

**Weaknesses:**

* No comparison with state-of-the-art subgraph-based methods (e.g. subgraph sketching).

**Questions:**

* The ATC heavily depends on the powers of the transition matrix. In this regard, how do you fix $K$? Large values of $K$ lead to a matrix with constant rows (ergodicity theorem).
* What are the computational advantages wrt subgraph-sketching, if any?
* What is the expected gain in performance wrt to state-of-the-art LP methods beyond the common-neighbors heuristic?

---

> ### Author Response · Authors · 2023-11-16
> **Response to Reviewer 6TYH, Weakness & Question 3, no comparison with SOTA subgraph sketching method and performance gain wrt to SOTA LP methods.**
>
> **=====Weakness=====**
>
> Thanks for sharing another SOTA baseline Subgraph sketch: ELPH and BUDDY, we have run their code, and here are the updated results for Table 1:
>
> | Baseline | Cora Hits@100  | Citeseer Hits@100 | Pubmed Hits@100 | Collab Hits@50 | Citation2 MRR | Reptile Hits@100 | Vole Hits@100 |
> |----------|----------------|-------------------|-----------------|----------------|---------------|------------------|---------------|
> | GCN      | 70.63±0.67     | 65.96±2.12        | 69.35±1.02      | 49.52±0.52     | 84.42±0.05    | 65.52±2.73       | 73.84±0.98    |
> | GCN$_{rw}$    | 75.98±1.28     | 74.40±1.13        | 68.87±0.99      | 52.85±0.14     | 83.54±0.30    | 70.79±2.00       | 74.50±0.84    |
> | SAGE     | 74.27±2.08     | 61.57±3.28        | 66.25±1.08      | 50.01±0.50     | 80.44±0.10    | 72.59±3.19       | 80.55±1.59    |
> | SAGE$_{rw}$   | 74.62±2.30     | 69.89±1.66        | 66.77±0.69      | 52.96±0.37     | 80.61±0.10    | 74.35±3.20       | 81.27±0.96    |
> | NCN      | 87.73±1.41     | 90.93±0.83        | 76.20±1.55      | 54.43±0.17     | 88.64±0.14    | 68.37±3.57       | 66.10±1.13    |
> | NCN$_{rw}$    | 87.95±1.30     | 91.86±0.82        | 76.51±1.41      | 54.16±0.24     | OOM             | 73.81±4.71       | 67.32±0.81    |
> | BUDDY    | 84.82±1.93     | 91.44±1.14        | 74.46±3.32      | 55.73±0.16     | More than two days    | 74.69±7.55                | 83.37±1.05
> | BUDDY$_{rw}$    | 85.06±1.81     | 91.86±1.00        | 75.74±2.03      | 56.17±0.42     | More than two days    | 77.20±4.10                | 84.64±1.29             |
>
> We download the most latest version of the BUDDY code and also modify it based on our reweighting strategy. We can see that equipping BUDDY with the reweighting could still enhance its link prediction performance. As running BUDDY on Citation2 cannot fit on our machine with 128 G RAM, we are currently working on getting another machine to run it. We will update the results once we have on Citation2.
>
> As you can see, equipping our strategy achieves higher performance than SOTA LP methods such as BUDDY.
>
> **=====================**

---

> > ### Author Response · Authors · 2023-11-16
> > **Response to Reviewer 6TYH, Question 1, how do we fix K and large K leads to constant rows.**
> >
> > **=====Question 1=====**
> >
> > We first acknowledge that large values of K would make the adjacency matrix (stochastic one) converge to a stationary point. However, **our approximated ATC leverages the accumulative summation of adjacency matrices from power 1 to power K, which would not lead to constant rows.**
> >
> > To determine K, based on the experiment shown in Figure 6, as K keeps increasing, the performance correlation curves become more and more stable. Therefore, we can take a modestly large K to guarantee a better correlation, as shown in Figure 6.
> >
> > **=====================**

---

> > > ### Comment · Reviewer_f1PG · 2023-11-21
> > > **Response to authors.**
> > >
> > > Thank you to the authors for including BUDDY (sota LP). I increased the rate.

---

### Official Review · Reviewer_6TYH · 2023-10-31

**Soundness:** 3 good
**Presentation:** 3 good
**Contribution:** 3 good
**Rating:** 6
**Confidence:** 3

**Summary:**

The authors identify a gap in understanding how different nodes in a graph achieve varying LP performance. They propose the Topological Concentration (TC) and its scalable version, Approximated Topological Concentration (ATC) metrics, which offer a more accurate measurement of the correlation between local subgraph structure and LP performance compared to traditional node degrees. Surprisingly, the paper reveals a counterintuitive observation that LP performance does not consistently increase with node degree, challenging conventional wisdom. Additionally, the paper uncovers a Topological Distribution Shift (TDS) issue, which impacts LP generalizability and highlights the importance of TC in understanding node-level LP performance. The authors propose TC-inspired message-passing techniques to enhance LP performance by focusing on well-connected neighbors within a node's computational tree. Overall, this research contributes valuable insights into LP performance variation and cold-start issues in GNNs, with potential implications for improving network dynamics and LP strategies.

**Strengths:**

* The paper's introduction of the Topological Concentration (TC) and Approximated Topological Concentration (ATC) metrics provides a new and innovative approach to addressing the variation in Link Prediction (LP) performance within Graph Neural Networks (GNNs).

* The paper's observation that LP performance does not consistently increase with node degree challenges existing assumptions in the field. This counterintuitive finding sparks curiosity and adds an element of novelty to the research.

* The paper proposed a new method to improve model's link prediction performances based on their TC and show empirical results.

**Weaknesses:**

* Discussion of the time complexity is not so good. The adjacent matrix are based on each layer's embedding, so the time consumption will be larger. Experiments on this are needed.
* Subgraph-based methods as baselines are not so complete.

**Questions:**

* Can you measure the time complexity through experiments? I guess the time consumption will not be so close to the original method.

* NCN already explicitly accounts for this common neighbor signal in its decoder, the performance gains from our strategy are relatively modest. So why don't you choose another subgraph-based method because I'm curious what's your strategy's effect on subgraph-based methods that don't explicitly use common neighbor signals.

* How to get the performances of link prediction of nodes with different TC? I mean one link has two nodes. For example for the results for nodes'TC in \[0,0.2\), does every link includes two nodes in that domain or just one node is enough?


* Have you tried to make TC^{Tr} closer to TC^{VAL}, if wanting to make model more generalizing to prediciting test links. Because we may assume validation set has the same distribution of test set.
* In Figure 7(a), why the curve of training original graph has several fast down of performances?

---

> ### Author Response · Authors · 2023-11-16
> **Response to Reviewer 6TYH, Question 3, how to get the link prediction performance of each node?**
>
> **=====Question 3=====**
>
> Thanks for the question. We measure the link prediction performance of each node following existing works in recommender systems [1]. The high-level idea is for each node, we compute its pairwise link prediction score with all other nodes and rank them. Then we check the overlap of the top-K of the ranked other nodes with the top-K of the ground-truth neighbors of this node. In this way, we define Recall/Precision/F1/NDCG/MRR/Hits@K for each node and group this performance based on nodes in different TC groups. **More details can be seen in Appendix B.**
>
> [1] He, Xiangnan, et al. "Lightgcn: Simplifying and powering graph convolution network for recommendation." Proceedings of the 43rd International ACM SIGIR conference on research and development in Information Retrieval. 2020.
>
> As another benefit, evaluating in this way avoids any sampling bias. Previously in evaluating each link prediction performance, we randomly sampled negative node pairs and compared their link prediction score with the ground one, which might suffer from bias by only sampling a tiny portion from all negative pairs, the number of which is quadratic to the total number of nodes in the network. However, our node-centric evaluation metric avoids this issue since for each node, we rank its ground-truth neighbors against all other nodes. We provide this discussion in Appendix B.
>
> **=====================**

---

> ### Author Response · Authors · 2023-11-16
> **Response to Reviewer 6TYH, Question 1, analyzing efficiency of our strategy theoretically**
>
> **=====Question 1=====**
>
> We sincerely thank the reviewer for the sharp observation of the issue in our time complexity analysis. We first clarify this complexity analysis in two cases. The first one is the time complexity of calculating approximated topological concentration. The second one is the time complexity of applying TC-inspired reweight to the message-passing of GNNs. We discuss them in the following respectively:
>
> For the time complexity of calculating approximated topological concentration, we:
> - Firstly, we initialized each node embedding sampled from d-dimensional standard multi-normal Gaussian distribution, taking $O(|V|d)$ time
> - Secondly, we perform message-passing K times, taking $O(K(|E| + |V|)d)$ time if using sparse operator [1]
> - Thirdly, we summerize the propagated embeddings from $k = 1$ to $k = K$, taking $O(k|V|d)$.
> - Lastly, we calculate the ATC by calculating the embedding similarity of each node to each of its neighbors, taking $O(|V|d)$.
>
> It takes $O(K(|E| + |V|)d)$ time, which is far more efficient than the original TC computation. And we further verify it empirically in Figure 6.
>
>
> For the time complexity of the TC-based edge reweighting method, we have an additional step to periodically update the edge weight based on the node embeddings obtained during the training process. However, the node embedding used for updating the edge weight can be directly borrowed from the most recent epoch as long as we cache it. Then, the only additional computation is the group-based softmax-nomralization. However, as shown in Algorithm 1 in Appendix G, we only periodically updated the edge weight; hence, the extra computational is pretty affordable.
>
> **=====================**

---

> ### Author Response · Authors · 2023-11-17
> **Response to Reviewer 6TYH, Question 4, compare reweighting based on training and reweighting based on validation.**
>
> **=====Question 4=====**
>
> We thank the reviewer for the excellent suggestion and experiment by reweighting the neighbors based on their connection level to the training neighbors or the validation neighbors. We indeed find that the performance improves more if we reweight based on validation neighbors closer to the training neighbors. We put the results as follows:
>
> | Performance | GCN No | GCN Train | GCN Valid | SAGE No | SAGE Train | SAGE Valid |
> |-------------|--------|-----------|-----------|---------|------------|------------|
> | Hits@5      | 18.94±1.20 | 19.48±0.75 | 22.36±0.32 | 11.25±1.24 | 20.52±2.35 | 24.34±0.07 |
> | Hits@10     | 31.24±3.44 | 32.69±1.00 | 35.15±2.42 | 26.41±1.88 | 31.23±3.52 | 37.15±2.44 |
> | Hits@50     | 50.12±0.22 | 52.77±1.00 | 53.24±0.22 | 49.68±0.25 | 51.87±0.10 | 52.69±0.26 |
> | Hits@100    | 54.44±0.49 | 56.89±0.17 | 57.28±0.10 | 54.69±0.18 | 56.59±0.19 | 57.27±0.25 |
>
> In the above Table, we can see that reweighting by training edges and validation edges can both lead to higher performance. Even more, reweighting by validation edges leads to more considerable performance gain, which aligns with our expectation that the distribution of validation edges is closer to testing edges due to the shorter time interval.
>
> **=====================**

---

> ### Author Response · Authors · 2023-11-21
> **Response to Reviewer 6TYH, Question 2, effect on subgraph-based methods.**
>
> **=====Question2=====**
>
> Thanks for asking about the effect of our method on subgraph-based methods. Here we choose the SOTA subgraph-based one BUDDY and augment it with our proposed reweighting strategy in Appendix H. We present the updated results in Table 1:
>
> | Baseline | Cora Hits@100  | Citeseer Hits@100 | Pubmed Hits@100 | Collab Hits@50 | Citation2 MRR | Reptile Hits@100 | Vole Hits@100 |
> |----------|----------------|-------------------|-----------------|----------------|---------------|------------------|---------------|
> | GCN      | 70.63±0.67     | 65.96±2.12        | 69.35±1.02      | 49.52±0.52     | 84.42±0.05    | 65.52±2.73       | 73.84±0.98    |
> | GCN$_{rw}$    | 75.98±1.28     | 74.40±1.13        | 68.87±0.99      | 52.85±0.14     | 83.54±0.30    | 70.79±2.00       | 74.50±0.84    |
> | SAGE     | 74.27±2.08     | 61.57±3.28        | 66.25±1.08      | 50.01±0.50     | 80.44±0.10    | 72.59±3.19       | 80.55±1.59    |
> | SAGE$_{rw}$   | 74.62±2.30     | 69.89±1.66        | 66.77±0.69      | 52.96±0.37     | 80.61±0.10    | 74.35±3.20       | 81.27±0.96    |
> | NCN      | 87.73±1.41     | 90.93±0.83        | 76.20±1.55      | 54.43±0.17     | 88.64±0.14    | 68.37±3.57       | 66.10±1.13    |
> | NCN$_{rw}$    | 87.95±1.30     | 91.86±0.82        | 76.51±1.41      | 54.16±0.24     | OOM            | 73.81±4.71       | 67.32±0.81    |
> | BUDDY    | 84.82±1.93     | 91.44±1.14        | 74.46±3.32      | 55.73±0.16     | More than two days    | 74.69±7.55                | 83.37±1.05
> | BUDDY$_{rw}$    | 85.06±1.81     | 91.86±1.00        | 75.74±2.03      | 56.17±0.42     | More than two days    | 77.20±4.10                | 84.64±1.29             |
>
> We download the most latest version of the BUDDY code and also modify it based on our reweighting strategy. We can see that equipping BUDDY with the reweighting could still enhance its link prediction performance. As running BUDDY on Citation2 cannot fit on our machine with 128 G RAM, we are currently working on getting another machine to run it. We will update the results once we have them on Citation2. But based on the results on most of the datasets, we can still see the positive impact of our reweighting strategy on most of the datasets.
>
> **=====================**

---

> > ### Comment · Reviewer_6TYH · 2023-11-21
> > **Response to the Authors**
> >
> > I appreciate the authors' responses. Although Question 5 is not responded, I will raise my score. And the authors should update the experiments results in their paper.

---

> > > ### Author Response · Authors · 2023-11-21
> > > **Appreciate the feedback and will keep running for some last results for addressing the left questions**
> > >
> > > We really appreciate the reviewers' response and we will keep running for additional results to address reviewers' remaining concerns.
> > >
> > > Once we collect all results, will update the paper and upload the final version.
> > >
> > > Thanks again for your time!

---

> > > ### Author Response · Authors · 2023-11-22
> > > **Response to Reviewer 6TYH, Question 5, why multiple fast-down performance trends?**
> > >
> > > We really thank Reviewer 6TYH for taking care of our discussion. We have further addressed your last question about the multiple fast-down performance trends in our reported performance curve on Collab.
> > >
> > > Here we delve deep into the reason why we encounter several fast-down performances in Figure 7(a). We ascribe it to the weight clip technique (we use the default publically available implementation https://github.com/snap-stanford/ogb/tree/master/examples/linkproppred/collab on GCN/SAGE on Collab link prediction and here it uses the weight clip technique), we employ the weight clip every time after parameter update. We hypothesize that the loss landscape has several local minimums and hence by weight clipping with higher upper bound constraints, our model updating step would be also larger so that the model could jump out of its original local optimum and keep finding some other better local optimum, which corresponds to the fast downtrend (first jump away from one local minimum and then find another better local minimum). We further empirically verify our hypothesis by visualizing the performance curve for each training process with different clipping weights (Figure 11 Appendix K), we can clearly see that as the clipping threshold becomes lower (upper bound decreases), we observe less fast downtrend decreases.

---

> ### Author Response · Authors · 2023-11-22
> **Response to Reviewer 6TYH, Weakness 1, experiments are needed to verify the efficiency of the proposed method.**
>
> We really thank the reviewer for taking care of the efficiency of our method. We have addressed this partially from the analysis point of view in our reply to question 1 above. Here we further empirically verify the efficiency by conducting running time comparison (s) among multiple baselines as below:
>
> | Baseline | Cora    | Citeseer | Pubmed | Collab | Reptile | Vole   |
> |----------|---------|----------|--------|--------|---------|--------|
> | GCN      | 19.5    | 15.5     | 158.4  | 2906   | 18.3    | 53.8   |
> | GCN$_{rw}$    | 21.1    | 17.2     | 158.5  | 2915   | 17.0    | 53.0   |
> | SAGE     | 22.3    | 17.0     | 189.8  | 2970   | 20.9    | 61.2   |
> | SAGE$_{rw}$  | 23.8    | 20.5     | 192.4  | 2982   | 21.7    | 61.9   |
> | BUDDY    | 3.41    | 4.51     | 15.51  | 906.18 | 2.50    | 4.99   |
> | BUDDY$_{rw}$  | 3.98    | 4.92     | 14.56  | 907.52 | 2.62    | 5.06   |
>
> We can find that equipping our reweighting strategy only induces marginal time complexity compared with its corresponding baseline. This is because, as we justified in the response above,
>
> (1) we only update the weight for each existing edge, which is linear to the size of the graph
>
> (2) we only periodically update rather than every epoch.
>
> (3) we directly use the cached node embeddings rather than computing from scratch.

---

### Official Review · Reviewer_1hhc · 2023-11-09

**Soundness:** 3 good
**Presentation:** 2 fair
**Contribution:** 3 good
**Rating:** 5
**Confidence:** 4

**Summary:**

This paper proposes a new metric called Topological Concentration (TC) for GNN-based link prediction. The authors also discover a novel topological distribution shift issue and use TC to quantify this shift and its negative impact. They design a message-passing scheme that reweights the edges based on the contribution of the neighbors. They show that this scheme can enhance TC and boost link prediction performance to some extent.

**Strengths:**

- The paper proposes a novel metric, Topological Concentration (TC), to measure the varying link prediction performance across different nodes.
- It demonstrates the superiority of TC over other node topological properties, such as degree and subgraph density.
- It explores the potential of boosting GNNs’ LP performance by enhancing TC via re-weighting edges in message-passing.

**Weaknesses:**

- The writing is unclear and hard to follow. The notations are confusing. The paper uses similar symbols for some concepts, such as ${TC}^{TR}$, ${TC}^{Tr}$, ${Tc}^{Tr}$, which are hard to distinguish. The paper also uses uncommon abbreviations for training and testing sets, such as Tr and Te.
- The motivation of probing the node characteristic for LP is questionable. The paper does not explain why this is a meaningful problem, given that [1] has proven that LP cannot be reduced to two node problems.
- The definition of Topological Concentration is complicated and the rationality is not obvious. The paper uses a complex formula to find the intersection of subgraphs at different hops, but does not justify its choice, such as the exponential decaying coefficients. Why don't you define $\mathcal{H}_i^k$ as $k$-hop neighbors, which is more clear and straightforward?
- The organization is confusing. The paper switches between different topics without clear transitions. For example, it introduces TC in section 3.2, then discusses cold-start nodes and distribution shift in section 3.3, and then returns to TC in section 3.4.
- Obs.2 and Obs. 3 in section 3.3 seem irrelevant for the proposed model and discussion. The paper does not explain how these observations inform the design or evaluation of the edge reweighting strategy.
- The technical novelty is limited. The paper only proposes edge reweighting as a strategy to enhance LP performance, which is a common technique in LP [2][3]. The paper does not compare or contrast its strategy with existing methods.
- The experiment in this paper is weak. You only show the relative gain of GCN/SAGE/NCN with reweighting, but you do not compare with the SOTA LP methods like BUDDY [4]. Your result is not competitive, as you only achieve 54% Hits@50 on Collab. Why do you not include experiments on other OGB datasets like ogbl-ddi and ogbl-ppa?
- The paper lacks ablation studies or case studies to demonstrate the effectiveness of the reweighting strategy. You could also provide some qualitative analysis or visualization to show how the reweighting strategy affects the prediction results.
- The font in every figure is too small to read, and the figures are not well-designed.

[1] Zhang, M., Li, P., Xia, Y., Wang, K., & Jin, L. (2021). Labeling trick: A theory of using graph neural networks for multi-node representation learning. In NeurIPS.

[2] Huang, Z., Kosan, M., Silva, A., & Singh, A. (2023). Link Prediction without Graph Neural Networks. arXiv preprint arXiv:2305.13656.

[3] Pan, L., Shi, C., & Dokmanić, I. (2022). Neural link prediction with walk pooling. In ICLR.

[4] Chamberlain, B. P., Shirobokov, S., Rossi, E., Frasca, F., Markovich, T., Hammerla, N., … & Hansmire, M. (2023). Graph neural networks for link prediction with subgraph sketching. In ICLR.

**Questions:**

See Weaknesses.

---

> ### Author Response · Authors · 2023-11-16
> **Response to Reviewer 1hhc, Weakness 1 and 2**
>
> **=====Weakness 1=====**
>
> We really appreciate the reviewer for pointing out the potential notation confusion. Since this work evaluates the link prediction performance on training/validation/testing links and also studies the distribution shift issue in node-link prediction, we need to separately denote them as Training (TR), Validation (Val), Testing (Te). These notations have also been frequently used in other works such as the following ones:
>
> [5] Jin, Wei, et al. "Empowering Graph Representation Learning with Test-Time Graph Transformation." The Eleventh International Conference on Learning Representations. 2022.
>
> [6] Bai Y, Chen M, Zhou P, et al. How important is the train-validation split in meta-learning?[C]//International Conference on Machine Learning. PMLR, 2021: 543-553.
>
> **=====================**
>
> **=====Weakness 2=====**
>
> Thanks for sharing the insight! We do admit that LP performance cannot be reduced to two node problems and our work does not study the relation between the node characteristics and the performance of each link. Instead, our work studies the link prediction performance at the node level (see Appendix B for the difference in evaluation link prediction performance between the link level and the node level). **We aim to study the relation between the local topology of each node and the link prediction performance of that node**. Moreover, when we study the local topology of each node, **we never reduce it to just that node nor the two nodes but instead, we study the interaction between the local topology of that node and the local topology of the neighbors of that node.**
>
> We motivate this study as follows:
>
> **(1) Real-world applications**: In social networks such as Facebook and e-commerce platforms such as Amazon, we care more about the user experience of each individual user than each specific social interaction or transaction. To maintain or even improve the user experience (in link prediction problem, it is to maintain a high link prediction performance for each node), the very first thing to do is to understand which nodes might have lower LP performance (which users might have worse user experience) and further identify its rooting reason on why these nodes have lower LP performance. Understanding the latent factors influencing the node LP performance would help these platforms detect early those poor-performing users and correspondingly develop user-specific strategies to enhance their experience (LP performance). This also corresponds exactly to what we did in this paper. As shown in Figure 3, we find that node LP performance is highly correlated to its TC and our TC-based reweighting strategy could enhance nodes’ LP performance to a certain extent.
>
> **(2) Our study is also important from the perspective of fairness.** Previous works [7,8] study the degree-related bias in node classification and knowledge graph completion. Only a few works [9] observe the degree-related bias in link prediction. However, they did not systematically study the underlying reason for this issue.
>
> [7] Tang, Xianfeng, et al. "Investigating and mitigating degree-related biases in graph convolutional networks." Proceedings of the 29th ACM International Conference on Information & Knowledge Management. 2020.
>
> [8] Shomer, Harry, et al. "Toward Degree Bias in Embedding-Based Knowledge Graph Completion." Proceedings of the ACM Web Conference 2023. 2023.
>
> [9] Wang, Yu, and Tyler Derr. "Degree-Related Bias in Link Prediction." 2022 IEEE International Conference on Data Mining Workshops (ICDMW). IEEE, 2022.
>
> **(3) In addition, characterizing node LP performance is very important in rethinking the cold-start issue.** Previously, the widely adopted notion in LP is that nodes with lower degrees have lower LP performance, and many existing works develop techniques to enhance LP performance for nodes with lower degrees in node classification. However, our work, for the very first time, breaks this conventional notion in Figure 1. The observation that low-degree nodes do not necessarily have lower link prediction performance would significantly influence future work studying cold-start issues and low-performing nodes in link prediction tasks.
>
> **=====================**

---

> ### Author Response · Authors · 2023-11-16
> **Response to Reviewer 1hhc, Weakness 2 discussing the connection between our TC and the labeling trick in [1]**
>
> **=====Weakness 2=====**
>
> Moreover, we appreciate the reviewer sharing this excellent literature [1]. We have thoroughly gone through it. The claimed “two-node problem” here refers to only aggregating two-node representations directly learned by basic GNNs like GAE. **We acknowledge that this would inform the problem of missing topological-relational information. However, we want to emphasize that, at any place in our work, we did not claim that we characterize the link prediction performance of each node only by itself and some other nodes.** Conversely, our proposed node-level topological metric, Topological Concentration (TC), indicates the intersection between the subgraph around that node and the subgraph around the neighbors of that node, which explicitly considers the topological-relational information.
>
> Furthermore, we want to highlight some connections between [1] and ours, which further justifies the motivation for using TC in characterizing node link prediction performance. Both works study  LP tasks and investigate the underlying reason for successfully performing LP. However, [1] focuses on the performance at each node pair (link level), while ours focuses on the performance at the node level. Therefore, the labeling trick proposed in [1] captures the topological characteristics of each node pair, while our TC essentially captures the topological attributes of each node.  Even more, a level of subgraph overlap between the node and its neighbors would equal the high TC of that node.
>
> **=====================**

---

> ### Author Response · Authors · 2023-11-16
> **Response to Reviewer 1hhc, Weakness 3, Definition of TC is complex and non-intuitive.**
>
> **=====Weakness 3=====**
>
> Sorry for the confusion. Previous studies [1]-[2] have found that the structural features of the enclosed subgraph of the node pair are essential in determining the link prediction between that node pair. Following this same logic, we hypothesize that each node's link prediction heavily depends on the enclosed subgraph between the node and its neighbors. This exactly corresponds to the formula we propose in Eq. (1). For each neighbor of a specific node $v_j \sim \mathcal{N}_i^t$, we consider the interaction between the computational tree $\mathcal{S}_i^K$ and $\mathcal{S}_j^K$. We discussed the intuition and rationality of TC at the beginning of Section 3.2.
>
>
> [1] Zhang, Muhan, and Yixin Chen. "Link prediction based on graph neural networks." Advances in neural information processing systems 31 (2018).
>
> [2] Chamberlain, Benjamin Paul, et al. "Graph neural networks for link prediction with subgraph sketching." arXiv preprint arXiv:2209.15486 (2022).
>
> **=====================**

---

> ### Author Response · Authors · 2023-11-16
> **Response to Reviewer 1hhc, Weakness 4, the organization is confusing.**
>
> **=====Weakness 4=====**
>
> Thank you for sharing your confusion. We want to highlight that **the main contribution of this work is not using TC to design a better link prediction method (although we did in Section 3.5 to demonstrate the usage of TC) but to systematically study the performance discrepancy of link prediction performance across different nodes, proposing a metric to quantify it, and demonstrate the real-world importance of using this metric in studying LP**. Following this line of story, we first introduce this metric, TC, in section 3.2. and then introduce the real-world importance of this metric in section 3.3 (we demonstrate its higher correlation than other metrics like degree, we demonstrate it can better identify low-performing nodes and we discover a new topological distribution shift issue using TC, all of which would inspire future research deepening our understanding of LP).
>
> **=====================**

---

> > ### Author Response · Authors · 2023-11-16
> > **Response to Reviewer 1hhc, Weakness 5, Obs.2 and Obs. 3 seem irrelevant for the proposed model and discussion.**
> >
> > **=====Weakness 5=====**
> >
> > Sorry for causing this confusion. Instead of contributing by solely proposing an edge reweighing strategy to augment GNN and enhance its LP performance, our work is to delve deeper into the relation between the node local topology and its link prediction performance and demonstrate the importance of studying this issue. To quantify such a relation, we propose topological concentration and verify its higher correlation to node LP performance in Obs 1. section 3.3. Then, we demonstrate other real-world applications of TC by Obs. 2 (better identifying low-performing nodes compared to a degree in cold-start issues) and Obs. 3 (discovering a new topological distribution shift issue by TC). Therefore, as our main story focuses on TC and its real-world application but not on proposing a specific method for enhancing LP, Obs.2 and Obs.3 are still crucial in our story.
> >
> > **=====================**

---

> > > ### Author Response · Authors · 2023-11-16
> > > **Response to Reviewer 1hhc, Weakness 8, lacks ablation studies or case studies on analyzing the reweight method**
> > >
> > > **=====Weakness 8=====**
> > >
> > > Based on our observation that TC consistently exhibits a stronger correlation with GNNs’ LP performance, we aim to boost TC by aggregating more information from neighbors who are more connected to the whole neighborhood and demonstrating the performance improvement in Table 1. We indeed provided case studies to illustrate the effectiveness of the proposed reweighting strategy. As shown in Figure 7(b), we visualize the relationship between the improvement of Hits@10 before and after equipping with our proposed TC-based reweighting strategy. We can see nodes with a more significant gain in their TC would also have a larger improvement in their link prediction performance.
> > >
> > > **=====================**

---

> ### Author Response · Authors · 2023-11-16
> **Response to Reviewer 1hhc, Weakness 7, do not compare with the SOTA LP methods like BUDDY**
>
> **=====Weakness 7=====**
>
> Thanks for sharing another SOTA baseline ELPH and BUDDY, we have run their code, and here are the updated results for Table 1:
>
> | Baseline | Cora Hits@100  | Citeseer Hits@100 | Pubmed Hits@100 | Collab Hits@50 | Citation2 MRR | Reptile Hits@100 | Vole Hits@100 |
> |----------|----------------|-------------------|-----------------|----------------|---------------|------------------|---------------|
> | GCN      | 70.63±0.67     | 65.96±2.12        | 69.35±1.02      | 49.52±0.52     | 84.42±0.05    | 65.52±2.73       | 73.84±0.98    |
> | GCN$_{rw}$    | 75.98±1.28     | 74.40±1.13        | 68.87±0.99      | 52.85±0.14     | 83.54±0.30    | 70.79±2.00       | 74.50±0.84    |
> | SAGE     | 74.27±2.08     | 61.57±3.28        | 66.25±1.08      | 50.01±0.50     | 80.44±0.10    | 72.59±3.19       | 80.55±1.59    |
> | SAGE$_{rw}$   | 74.62±2.30     | 69.89±1.66        | 66.77±0.69      | 52.96±0.37     | 80.61±0.10    | 74.35±3.20       | 81.27±0.96    |
> | NCN      | 87.73±1.41     | 90.93±0.83        | 76.20±1.55      | 54.43±0.17     | 88.64±0.14    | 68.37±3.57       | 66.10±1.13    |
> | NCN$_{rw}$    | 87.95±1.30     | 91.86±0.82        | 76.51±1.41      | 54.16±0.24     | OOM             | 73.81±4.71       | 67.32±0.81    |
> | BUDDY    | 84.82±1.93     | 91.44±1.14        | 74.46±3.32      | 55.73±0.16     | More than two days    | 74.69±7.55                | 83.37±1.05
> | BUDDY$_{rw}$    | 85.06±1.81     | 91.86±1.00        | 75.74±2.03      | 56.17±0.42     | More than two days    | 77.20±4.10                | 84.64±1.29             |
>
> We download the most latest version of the BUDDY code and also modify it based on our reweighting strategy. We can see that equipping BUDDY with the reweighting could still enhance its link prediction performance. As running BUDDY on Citation2 cannot fit on our machine with 128 G RAM, we are currently working on getting another machine to run it. We will update the results once we have them on Citation2
>
> Moreover, **our result on Collab is still competitive** since different from subgraph sketching allowing validation edges used in message-passing when evaluating edges in the testing set, we strictly follow the conventional setting[1] avoiding using validation edges during testing time.
>
> [1] Yun, Seongjun, et al. "Neo-gnns: Neighborhood overlap-aware graph neural networks for link prediction." Advances in Neural Information Processing Systems 34 (2021): 13683-13694.
> **=====================**

---

> > ### Comment · Reviewer_1hhc · 2023-11-21
> > **Response to authors**
> >
> > Thank you for your responses. While they address some of my concerns, a few points still remain:
> > - My primary concern is the focus on the node level in tackling the LP problem, which is an uncommon research direction in LP tasks. Established LP algorithms, such as SEAL, NCN, and BUDDY, explicitly consider link topology, involving subgraphs or subgraph sketches. In your paper, you posit that nodes with higher TC are more likely to form links with others, essentially reducing link problems to node problems. This approach is statistical, only leading to the conclusion that concentrated nodes are more inclined to form links. However, the contribution and insights it provides to the link prediction community appear to be somewhat constrained.
> > - Although your results are compared with BUDDY, the datasets are not large-scale, where simpler models could perform well, and the observed improvement is marginal. It would be more insightful to include results on larger datasets like ogbl-ddi, ogbl-citation2, and ogbl-ppa.
> > - BUDDY attains a 66% Hits@50 in ogbl-collab by using of validation edges during **training**, not **testing** as you claimed. This practice could also be adopted in your approach, rather than attributing weaker results to the absence of this technique.
> > - The writing could benefit from reorganization and refinement of notation to enhance clarity.
> >
> > Despite these concerns not being addressed, I will maintain my current score.

---

> ### Author Response · Authors · 2023-11-23
> **Response to Reviewer 1hhc, Remaining Concern 3, not using validation edges in training**
>
> We extremely appreciate reviewer 1hhc for the persistent interest in helping polish and improve our work!
>
> We have systematically read previous papers [1]-[2] and indeed admit that they employ a setting on Collab by also using validation edges and gaining even higher performance. Following this awesome suggestion, we also modify our framework by including validation edges during the message-passing (the column Collab* in the table below) and we update the results as follows:
>
> | Baseline | Cora Hits@100  | Citeseer Hits@100 | Pubmed Hits@100 | **Collab Hits@50** | Collab* Hits@50 | Citation2 MRR | Reptile Hits@100 | Vole Hits@100 |
> |----------|----------------|-------------------|-----------------|----------------|---------------|---------------|------------------|---------------|
> | GCN      | 70.63±0.67     | 65.96±2.12        | 69.35±1.02      | 49.52±0.52     | **58.00±0.27**| 84.42±0.05    | 65.52±2.73       | 73.84±0.98    |
> | GCN$_{rw}$    | 75.98±1.28     | 74.40±1.13        | 68.87±0.99      | 52.85±0.14     | **60.57±0.38** | 83.54±0.30    | 70.79±2.00       | 74.50±0.84    |
> | SAGE     | 74.27±2.08     | 61.57±3.28        | 66.25±1.08      | 50.01±0.50     | **57.06±0.06** | 80.44±0.10    | 72.59±3.19       | 80.55±1.59    |
> | SAGE$_{rw}$   | 74.62±2.30     | 69.89±1.66        | 66.77±0.69      | 52.96±0.37     | **59.26±0.08** | 80.61±0.10    | 74.35±3.20       | 81.27±0.96    |
> | NCN      | 87.73±1.41     | 90.93±0.83        | 76.20±1.55      | 54.43±0.17     | **65.34±0.03** | 88.64±0.14    | 68.37±3.57       | 66.10±1.13    |
> | NCN$_{rw}$    | 87.95±1.30     | 91.86±0.82        | 76.51±1.41      | 54.16±0.24     | **65.41±0.03** | OOM             | 73.81±4.71       | 67.32±0.81    |
> | BUDDY    | 84.82±1.93     | 91.44±1.14        | 74.46±3.32      | 55.73±0.16     | **66.25±0.28** | More than two days    | 74.69±7.55                | 83.37±1.05
> | BUDDY$_{rw}$    | 85.06±1.81     | 91.86±1.00        | 75.74±2.03      | 56.17±0.42     | **66.31±0.50** | More than two days    | 77.20±4.10                | 84.64±1.29             |
>
>
> From the above Table column Collab*, we can see even including validation edges, our reweighting method still works in boosting the performance since here we still reweight the trailing edges in the message-passing and hence capture more useful neighbors that contribute more to the TC. However, we do admit that the performance improvement caused by our reweight method is weaker when equipping on the advanced LP baselines like NCN and BUDDY. This is reasonable since our reweighting essentially captures more network topology signals by aggregating more information from training neighbors who have more topology signals (common neighbors with the whole neighbors) and these captured signals can also be captured by BUDDY or NCN since they explicitly use network structure features (sketches) in their decoder.
>
> We really appreciate the reviewer for bringing this experiment up to further boost our work!
>
> [1] Chamberlain, Benjamin Paul, et al. "Graph neural networks for link prediction with subgraph sketching." arXiv preprint arXiv:2209.15486 (2022).
>
> [2] Zhang, Muhan, et al. "Labeling trick: A theory of using graph neural networks for multi-node representation learning." Advances in Neural Information Processing Systems 34 (2021): 9061-9073.
>
> We will update the above results in our final revised paper.

---

> ### Author Response · Authors · 2023-11-23
> **Response to Reviewer 1hhc, Remaining Concern 4, refine notations and paper reorganization**
>
> We really thank reviewer 1hhc for taking care of the readability of our work and we have constructed a comprehensive update to our paper throughout the discussion phase based on the suggestions from other reviewers. Now that all of them have joined the discussion, we have provided the updated version of our revised pdf file including all modifications. We apologize for not uploading a new PDF earlier, but did not want to cause confusion with multiple updates and just maintained all the changes in the comments until the end of this phase with the single updated PDF.
>
> **For the notations, we indeed admit that TC$^{\text{Tr}}$, TC$^{\text{Te}}$ are very confusion and have updated them to TC$^{\text{Train}}$ and TC$^{\text{Test}}$.**
>
> **Moreover, we have increased the font size from 15 to 25 for all figure x/y tick labels**. We really hope with such an larger font size, our work will be user-friendly for others to read.

---

> ### Author Response · Authors · 2023-11-23
> **Response to Reviewer 1hhc, Remaining Concern 2, simpler models could perform well, no ppa/ddi results**
>
> Actually, we observe that simple models here such as GCN and GCN$_{\text{rw}}$ are quite poor in performance compared to NCN or BUDDY. Furthermore, their respective augmented versions nearly always provide a consistent increase in performance.
>
> We didn't include results on ogbl-ddi/ppa because studying node-centric LP performance on these two datasets may not quite make sense since, in real-world applications, we mostly care about user experience (node-level performance) for social networks. However, for biology such as protein-protein interaction networks, we actually want to recover every potential interaction. Therefore, it is not meaningful to study the node-centric perspective as we did in our paper on these datasets.

---

> ### Author Response · Authors · 2023-11-23
> **Response to Reviewer 1hhc, Remaining Concern 1, studying node-level LP problem is rare**
>
> Thank you for worrying about our motivation and the significance of studying the node-level link prediction problem.
>
> **We first justify why it is important to study the link prediction performance for each node as follows:**
>
> (1) Nowadays, the most important real-world application is to use link prediction to enhance user experience, e.g., social recommendation(Instagram, Twitter), video recommendation (TikTok), and e-commerce recommendation (amazon). Therefore, compared with only understanding the link prediction performance for each link, we actually care more about how different users/items (i.e., nodes) perform in the real world. Studying this problem would help us better understand what factors would affect the user experience of certain groups of users, help us early detect those potentially disadvantages groups and so can devise corresponding actions to mitigate (e.g., collect more data for those groups of users). This is even more important from the fairness perspective, e.g., how to ensure users with different interaction levels end up with equal recommendation performance.
>
> (2) More importantly, we want to highlight that we don't try to reduce the link prediction from link problems to node problems, we are only providing a new perspective on analyzing the link prediction performance from the node perspective. For our methods, we still follow the architecture of many baselines.
>
> (3) **Also, at any time, we do not claim that nodes with higher TC would tend to form more links with other nodes. What we can derive based on the definition of TC is nodes with higher TC tend to have neighbors that are more connective with themselves** (a real-world case would be nodes with higher TC may correspond to certain users who share very consistent social intentions and hence usually form relations with others who also share relations (the neighbors of the higher TC user would connection with each other by themselves). This is very intuitive in reflecting the link prediction performance of each node since if the node's neighbors have very diverse community distributions, how can we leverage its training neighbors to predict testing neighbors who may have no connections with these training neighbors?
>
> (4) **At a very high level, we want to highlight that the most contribution of our paper is not to contribute a new SOTA method (as there are already quite a lot of well-established baselines such as BUDDY or NCNC or SEAL or NeoGNN), but the analysis perspective at the node level. In addition, we break the previous conventional notion [1-2] that low-degree nodes may perform worse than high-degree nodes and also observe the topological distribution shift issue that compromises the link prediction performance (which has also been concurrently observed in [3]). We strongly believe that all of these observations would enrich future research opportunities to further boost the LP community by analyzing the data rather than designing another model (essentially the data-centric AI).**
>
> [1] Liu, Zemin, Trung-Kien Nguyen, and Yuan Fang. "On Generalized Degree Fairness in Graph Neural Networks." arXiv preprint arXiv:2302.03881 (2023).
>
> [2] Li, Yunqi, et al. "User-oriented fairness in recommendation." Proceedings of the Web Conference 2021. 2021.
>
> [3] Wang, Xiyuan, Haotong Yang, and Muhan Zhang. "Neural Common Neighbor with Completion for Link Prediction." arXiv preprint arXiv:2302.00890 (2023).
>
>
>
> **We sincerely hope the reviewer to reconsider the motivation, the analysis, and the contribution of our work from the analysis perspective**

---

### Author Response · Authors · 2023-11-23

Dear Review Committee,

We greatly appreciate your insightful feedback and constructive suggestions throughout this review process. As today, November 23rd, marks the end of the author/reviewer discussion period, we hope that our responses and the additional experiments we conducted have effectively addressed your concerns and enhanced the overall quality of our manuscript.

Should you have any more thoughts or recommendations, we encourage you to share them with us. In the meantime, we wish you a wonderful Thanksgiving with your loved ones.

Warm regards,

Authors

---

### Meta-Review · Area_Chair_P8Jg · 2023-12-07

**Metareview:**

The study explores the varying performance of Graph Neural Networks (GNNs) in link prediction (LP) across different nodes, proposing a metric called Topological Concentration (TC) that correlates with LP performance and identifies a novel topological distribution shift issue, thereby enhancing LP performance by re-weighting edges in the message-passing.

The main concerns raised by reviewers are: Motivation about node-centric LP / Results of using validation edges / Paper writing. The questions have been generally addressed during the discussion period.

**Justification For Why Not Higher Score:**

The paper proposes a new method for an old problem, the method itself does not fundamentally change the landscape of the link prediction problem.

**Justification For Why Not Lower Score:**

The method proposed by the authors make sense, and main concerns have been addressed during the discussion period.

---

### Decision · Program_Chairs · 2024-01-16

Accept (poster)